# *Pendraig milnerae*, a new small-sized coelophysoid theropod from the Late Triassic of Wales

Stephan N. F. Spiekman[1,2], Martín D. Ezcurra[2,3], Richard J. Butler[2], Nicholas C. Fraser[4] and Susannah C. R. Maidment[1,2]

[1]Department of Earth Sciences, Natural History Museum, Cromwell Road, London SW7 5BD, UK
[2]School of Geography, Earth and Environmental Sciences, University of Birmingham, Edgbaston, Birmingham B15 2TT, UK
[3]Sección Paleontología de Vertebrados, CONICET-Museo Argentino de Ciencias Naturales, Ángel Gallardo 470, C1405DJR Buenos Aires, Argentina
[4]National Museums Scotland, Chambers Street, Edinburgh EH1 1JF, UK

SNFS, 0000-0002-3197-8752; MDE, 0000-0002-6000-6450; RJB, 0000-0003-2136-7541; NCF, 0000-0003-4408-4640; SCRM, 0000-0002-7741-2500

**Subject Areas:**
palaeontology/evolution/taxonomy and systematics

**Keywords:**
*Pendraig*, Coelophysoidea, Theropoda, Triassic, body size evolution, osteology

**Author for correspondence:**
Stephan N. F. Spiekman
e-mail: stephanspiekman@gmail.com

We describe a new small-bodied coelophysoid theropod dinosaur, *Pendraig milnerae* gen. et sp. nov, from the Late Triassic fissure fill deposits of Pant-y-ffynnon in southern Wales. The species is represented by the holotype, consisting of an articulated pelvic girdle, sacrum and posterior dorsal vertebrae, and an associated left femur, and by two referred specimens, comprising an isolated dorsal vertebra and a partial left ischium. Our phylogenetic analysis recovers *P. milnerae* as a non-coelophysid coelophysoid theropod, representing the first-named unambiguous theropod from the Triassic of the UK. Recently, it has been suggested that Pant-y-ffynnon and other nearby Late Triassic to Early Jurassic fissure fill faunas might have been subjected to insular dwarfism. To test this hypothesis for *P. milnerae*, we performed an ancestral state reconstruction analysis of body size in early neotheropods. Although our results indicate that a reduced body size is autapomorphic for *P. milnerae*, some other coelophysoid taxa show a similar size reduction, and there is, therefore, ambiguous evidence to indicate that this species was subjected to dwarfism. Our analyses further indicate that, in contrast with averostran-line neotheropods, which increased in body size during the Triassic, coelophysoids underwent a small body size decrease early in their evolution.

# 1. Introduction

The Late Triassic and Early Jurassic fissure fill deposits of southwestern England and southern Wales were formed as Mississippian Carboniferous Limestone that were exposed at the land surface and which underwent karstic weathering processes and faulting [1,2]. Previously, the putative Triassic age of the sediments was largely assessed as such on their basic lack of mammaliaform remains and their occurrence in generally wider, distinctly solution-etched karstic features; they were considered to be representative of an upland environment (e.g. [2]). Other fissures containing generally less diverse vertebrate assemblages, but including extensive mammaliaform remains and occurring in more restricted slot-like openings in the limestone, were regarded as representative of island communities—most specifically Early Jurassic as corroborated by palynomorphs [3,4]. Since then, many new discoveries have been made and evidence has been presented to suggest that at least some of the putative Triassic fissure fills were subject to the influences of marine transgressions and they are now considered to have been distributed among various near-shore islands [5–9]. Dating the fissure fills has continued to prove difficult due to their depositional setting, but some recent revisions, including faunal comparisons and palynological analyses, now argue that they are between early Rhaetian and early Sinemurian in age [1,6,10]. Yet, doubt still remains and others [11,12] continue to contend that some of the assemblages date back to the Norian and were filled at a time when the area was probably more akin to a broad sabkha-type environment.

Due to their unique preservational setting, a rich assemblage of mostly small-bodied vertebrates has been recovered from the fissure fills, which has provided important insights into the early evolution of major tetrapod groups such as mammaliaforms, rhynchocephalians, crocodylomorphs and dinosaurs, as well as enigmatic diapsid groups such as kuehneosaurids (e.g. [13–18]). It has been suggested that taxa known from the fissure fills are smaller than their close relatives [19–21]. Although this has not been tested quantitatively, it has tentatively been attributed to insular dwarfism, which is a manifestation of the biogeographic concept known as the 'Island Rule' [22,23]. However, most of the specimens are preserved as isolated fragments, meaning that the taxonomic status and phylogenetic position of much of the material has remained controversial.

Among the fissure fills faunas, archosaurs are represented by aetosaurs, phytosaurs, an enigmatic pseudosuchian, crocodylomorphs, sauropodomorph dinosaurs and theropod dinosaurs, many of which have not been formally described (e.g. [1,6,24]). *Thecodontosaurus antiquus* and *Pantydraco caducus*, two sauropodomorph dinosaurs, represent the best studied archosaurs from the fissure fills. *Thecodontosaurus antiquus* was the fourth British dinosaur to be described [25,26] and is represented by mostly isolated remains from Durdham Down Quarry in Bristol and Tytherington Quarry (?Rhaetian) [20,27]. *Pantydraco caducus* is known from the putatively coeval Pant-y-ffynnon Quarry and is represented by a partial articulated skeleton including most of the skull, as well as a few partial skeletons and isolated elements [28,29]. The material of *Pa. caducus* is considerably smaller than that described for *Th. antiquus*, and it was originally described and identified as *Thecodontosaurus* sp. [30]. The Pant-y-ffynnon specimens were subsequently referred to *Thecodontosaurus caducus* [29] and later *Pa. caducus* [31] and considered to differ from *Th. antiquus* in the morphology of the cervical vertebrae, ilium and humerus. Although *Pa. caducus* has been described in detail [28,29], its taxonomic status is contentious, since the observed morphological differences with *Th. antiquus* are minor and could be attributable to taphonomic deformation or ontogenetic variation, and *Pa. caducus* might represent an early ontogenetic stage of *Th. antiquus* [20].

Crocodylomorphs are currently represented by a single taxon, *Terrestrisuchus gracilis*, a gracile non-crocodyliform crocodylomorph known from several partially articulated and many disarticulated specimens from Pant-y-ffynnon Quarry [18]. However, extensive crocodylomorph material is also known from Cromhall Quarry and might represent two separate species, both different from *T. gracilis* [32], but this material has not yet been studied in detail. Cromhall Quarry has also yielded isolated material, comprising an ilium, maxilla, humerus, two astragali and a tooth that were interpreted to represent a single dinosauriform taxon, 'Agnosphytis cromhallensis' [32]. The 'Agnosphytis cromhallensis' specimens were inferred to represent a single taxon because of the absence of additional archosaurian material among the Cromhall Quarry specimens, except for confidently identified crocodylomorph and 'rauisuchian' remains. However, the astragalus possessed several dinosaurian traits, the holotype specimen, an ilium, appeared similar to sauropodomorphs, and the maxilla had features present in non-dinosaurian dinosauriforms; consequently, it was suggested that 'Agnosphytis cromhallensis' might represent a chimera and a *nomen dubium* [33–37]. The original hypodigm of 'Agnosphytis cromhallensis' was recovered as a silesaurid in the analysis of Baron *et al.* [33] and in subsequent revisions of this analysis in a large polytomy at the base of Saurischia [38] or as the sister taxon to Herrerasauridae and Dinosauria [39]. More recently, an enigmatic pseudosuchian archosaur of equivocal phylogenetic affinity,

*Aenigmaspina pantyffynnonensis*, has been described from Pant-y-ffynnon based on two complementary blocks comprising semi-articulated vertebrae, ribs, osteoderms, a scapula and a number of fragments including skull bones, in addition to several fragmentary remains [24]. Additional fragmentary archosaur remains have also been identified from various fissure fills localities. These include a single aetosaur scute from Cromhall Quarry [40] and a phytosaur tooth and a possible humerus from Durdham Down Quarry assigned to '*Paleosaurus platyodon*', which is now considered to represent a *nomen dubium* [27].

Finally, a theropod has been reported from Pant-y-ffynnon Quarry. It was originally described in the unpublished PhD thesis of Warrener [41] and interpreted as a coelurosaur. In it, an articulated pelvic girdle, associated with the two posteriormost dorsal vertebrae and four sacral vertebrae, a largely complete femur, an isolated dorsal vertebra, and a partial ischium were referred to the unnamed theropod. Additionally, it was suggested that three phalanges, including one ungual, and a metapodial that were associated with the other remains could possibly be referred to the same taxon. Subsequently, the Pant-y-ffynnon theropod was briefly described and interpretative drawings of the pelvic girdle and femur were presented by Rauhut & Hungerbühler [42]. Therein, the material was assigned to *Syntarsus* sp. and considered to be closely related to the coelophysoid theropods '*Syntarsus*' *rhodesiensis* (now *Megapnosaurus rhodesiensis*) and *Procompsognathus triassicus*. The Pant-y-ffynnon theropod was also mentioned by Galton & Kermack [28] and therein referred to as *Coelophysis* without providing further justification for this assignment. Recently, it was reported that the material could not be located within the NHMUK collections [43]. However, the articulated partial pelvic girdle and vertebrae, as well as the femur and complete isolated dorsal vertebra referred to the theropod by Warrener [41] have now been relocated. Here, we provide the first detailed comparative description of this new taxon. We incorporate it in a phylogenetic analysis and compare its relative body size to other early theropod dinosaurs to examine whether the small size of the taxon might be a consequence of island dwarfism, as has been suggested for other elements of the fissure fills faunas [19–21].

## 1.1. Institutional abbreviations

AMNH, American Museum of Natural History, New York, NY, USA; CMNH, Carnegie Museum of Natural History, Pittsburgh, PA, USA; FMNH, Field Museum of Natural History, Chicago, IL, USA; HMN, Museum für Naturkunde der Humboldt Universität, Berlin, Germany; MCZ, Museum of Comparative Zoology, Cambridge, MA, USA; MNA, Museum of Northern Arizona, Flagstaff, AZ, USA; NHMZ, Natural History Museum Zimbabwe, Bulawayo, Zimbabwe; NHMUK, The Natural History Museum, London, UK; NMV, Museum Victoria, Melbourne, Australia; NMW, National Museum of Wales, Cardiff, UK; PEFO, Petrified Forest National Park, AZ, USA; PVL, Paleontología de Vertebrados, Instituto 'Miguel Lillo', San Miguel de Tucumán, Tucumán, Argentina; PVSJ, División de Paleontología de Vertebrados del Museo de Ciencias Naturales y Universidad Nacional de San Juan, San Juan, Argentina; SMF, Sauriermuseum Frick, Frick, Switzerland; SMNS, Staatliches Museum für Naturkunde Stuttgart, Stuttgart, Germany; TMM, Texas Memorial Museum, Austin, TX, USA; UCMP, University of California Museum of Paleontology, Berkeley, CA, USA; USNM, National Museum of Natural History (formerly United States National Museum), Smithsonian Institution, Washington, DC, USA; WARMS, Warwickshire Museum, Warwick, UK.

# 2. Systematic palaeontology

Archosauria Cope, 1869–1870 [44] [Gauthier & Padian, 2020] [45]

Dinosauria Owen, 1842 [46] [Langer, Novas, Bittencourt, Ezcurra & Gauthier, 2020] [47]

Theropoda Marsh, 1881 [48] [Naish, Cau, Holtz, Fabbri & Gauthier, 2020] [49]

Neotheropoda Bakker, 1986 [50], *sensu* Sereno, 1998 [51]

Coelophysoidea Nopcsa, 1928 [52], *sensu* Sereno, 1998 [51]

*Pendraig milnerae* gen. et sp. nov.

**Nomenclatural acts.** This publication and its nomenclatural acts are registered at ZooBank. The publication is registered under LSID urn:lsid:zoobank.org:pub:E9F56CD2-AD1A-4E93-91A1-18618342C838, the new genus *Pendraig* under LSID urn:lsid:zoobank.org:act:724C0C8A-491D-4028-B692-FA921FB1273F, and the specific name *milnerae* under LSID urn:lsid:zoobank.org:act:CDEC343D-15F3-4A5B-A4F3-637C0D2A52E8.

**Etymology.** *Pendraig* from the Welsh *Pen* (head, chief or top) and *Draig* (dragon), literally meaning 'chief dragon' but used in a figurative sense in Medieval Welsh to mean 'chief warrior'. The anglicized form, Pendragon, was the epithet of Uther, father of King Arthur in medieval legend. *Milnerae*, in honour of Dr Angela C. Milner (1947–2021), in recognition of her major contributions to vertebrate palaeontology, including as one of the leading experts on British theropod dinosaur fossils [53], and to the Natural History Museum, London, where the type specimen is held.

**Diagnosis.** *Pendraig milnerae* is a small-sized non-averostran theropod (estimated femoral length: 10.21 cm; lower 95% CI: 8.60 cm; upper 95% CI: 12.08 cm; see below) that differs from other dinosaurs in the following unique combination of character states present in the holotype (autapomorphies indicated with an asterisk): posteriormost dorsal vertebrae with a strongly elongated centrum (centrum length *ca* 2.6 times its anterior height), ilium with a distinctly anteroventrally slanting dorsal margin of the preacetabular process, and posterodorsal margin of the postacetabular process curving abruptly posteroventrally and, as a result, the posteroventral end of the process is formed by an acute angle of approximately 65° in lateral view*; pubis with pubic fenestra; ischium with well-developed obturator plate but without posteroventral projection forming a deep U-shaped or V-shaped notch with the shaft; and femur with fourth trochanter posteriorly developed to a height similar to the mid-depth of the shaft at that level. In addition, the referred middle–posterior dorsal vertebra differs from other early neotheropods in the absence of an accessory hyposphene–hypantrum articulation, and the presence of an anteriorly expanded neural spine*.

**Holotype.** NHMUK PV R 37591: An articulated vertebral series and pelvic girdle comprising the two posteriormost dorsal vertebrae missing most of the neural spines, the three anteriormost sacral vertebrae and a small fragment of the centrum of the fourth sacral, a complete left ilium, a largely complete left pubis missing the distal end, a left ischium missing most of the distal portion, a largely complete right pubis missing the distal end, and a right ischium missing most of the dorsal and distal portions (field number P77/1) (figures 1 and 2). Additionally, a left femur was found disarticulated from the left hemipelvis in the same block (field number P76/1). It has been completely freed from the matrix and is confidently referred to the same individual as the vertebral column and pelvis (figure 3).

**Referred material**. NHMUK PV R 37596 (field number P83/1): A complete middle to posterior dorsal vertebra completely freed from matrix (figure 4). NHMUK PV R 37597 (field number P65/66b): the proximal end of a left ischium preserving the articular facet with the left ilium (figure 5). A counterslab to this specimen, listed as field number P65/66a and comprising the base of the distal portion and part of the proximal expansion of the ischium, was described and figured in the unpublished PhD thesis of Warrener [41]. However, there is no record of this specimen in the collections at NHMUK and it is therefore not considered here.

**Locality and horizon.** Fissure fills of Pant-y-ffynnon Quarry, southern Wales; earliest occurrence possibly late Norian, latest occurrence possibly late Rhaetian; 214.7–201.3 Ma [1].

**Remarks.** Additional material from Pant-y-ffynnon, comprising two isolated non-ungual phalanges (field numbers P65/30 and P65/49), an isolated metapodial (field number P65/23) and an isolated ungual (field number P65/45), were considered to possibly be conspecific with *P. milnerae* by Warrener [41] and interpretative drawings of these elements were provided therein. These elements cannot be confidently attributed to any other known taxa known from Pant-y-ffynnon and were likely recovered from the same block as the partial left ischium of *P. milnerae* (field number P65/66b) as indicated by the shared number 65 in their field number. The location of these elements is currently unknown and there is no record of them in the NHMUK collections. Since these elements do not exhibit diagnostic theropod features, there is currently insufficient support for an unequivocal attribution of this material to *P. milnerae*.

**Ontogenetic assessment.** The limited material currently referable to *P. milnerae* limits interpretations of intraspecific variation related to ontogeny, and, therefore, a comprehensive assessment of maturity cannot be provided. However, we approximate the ontogenetic stage of the holotype NHMUK PV R 37591 by scoring it for the maturity assessment matrix that has been formulated for early theropods (*Coelophysis bauri* and *M. rhodesiensis*) and a non-dinosaurian dinosauriform (*Asilisaurus kongwe*) [54–56]. Based on the results of our phylogenetic analysis (see below), these taxa provide a narrow phylogenetic bracket for *P. milnerae*, and the characters are, therefore, likely suitable for approximating the maturity of this new taxon (see also [57]). However, it is important to consider that it cannot be confidently ascertained whether all characters that were scored are truly ontogenetically dependent for this specific taxon. Out of 32 characters of the maturity assessment matrix, 16 could be scored for NHMUK PV R 37591. Eight characters were scored as mature (1) and another eight were scored as immature (0). None of the

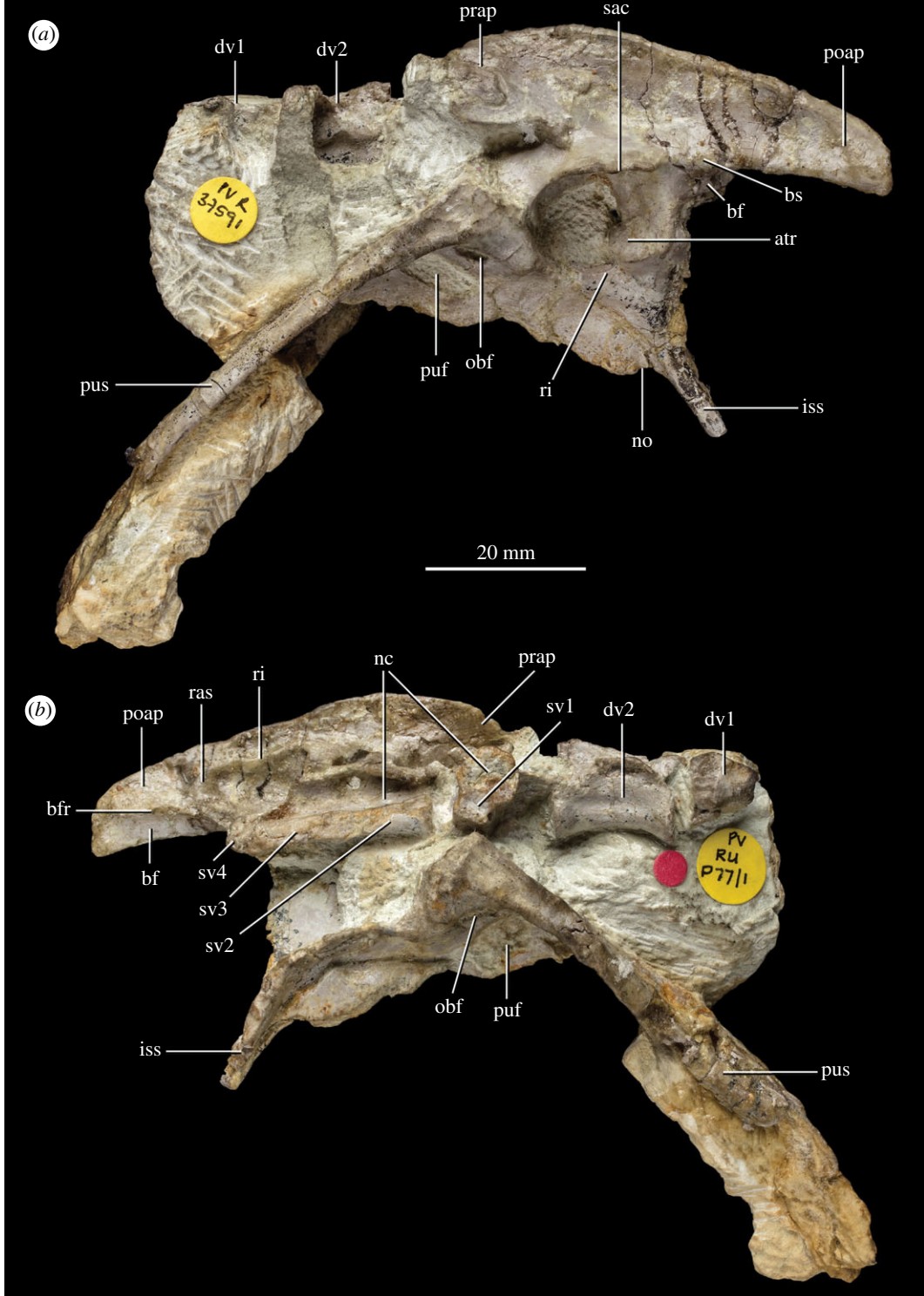

**Figure 1.** Holotype NHMUK PV R 37591 pelvis and vertebrae of *P. milnerae* gen. et sp. nov. in (*a*) left lateral view and (*b*) right lateral view. atr, antitrochanter; bf, brevis fossa; bfr, brevis fossa rim; bs, brevis shelf; dv, dorsal vertebra; iss, ischial shaft; nc, neural canal; no, notch; obf, obturator foramen; poap, postacetabular process; prap, preacetabular process; puf, pubic fenestra; pus, pubic shaft; ras, rib attachment scar; ri, rim; sac, supra-acetabular crest; sv, sacral vertebra.

characters considered highly informative of maturity (see fig. 22*a* in [54]) are scored as 1, nor are any of the characters considered highly informative of immaturity scored as 0. Of the sequence order of mature state attainment (between 1 and 29) [54,57], NHMUK PV R 37591 has a relative minimum maturity of 5 based on the presence of a mound-like dorsolateral trochanter on the femur (character 16, state 1 [16–1]), the presence of the *linea intermuscularis caudalis* on the femur (18–1), and the presence of an 'obturator ridge' on the femur (20–1), and a maximum maturity of 19 based on the co-ossification of less than four sacral

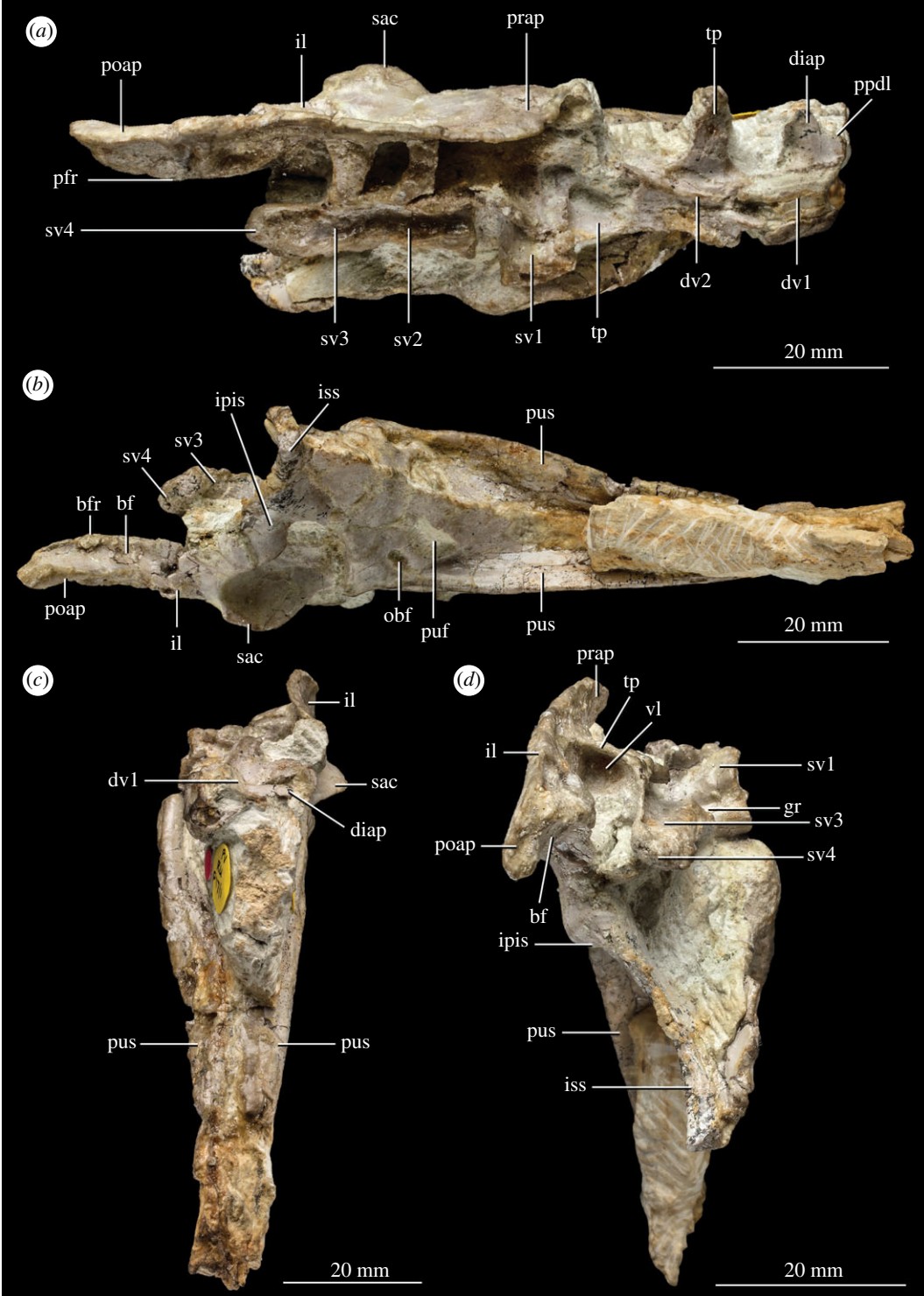

**Figure 2.** Holotype NHMUK PV R 37591 pelvis and vertebrae of *P. milnerae* gen. et sp. nov. in (*a*) dorsal view, (*b*) ventral view, (*c*) anterior view and (*d*) posterior view. bf, brevis fossa; bfr, brevis fossa rim; diap, diapophysis; dv, dorsal vertebra; gr, groove; il, ilium; ipis, iliac peduncle of the ischium; iss, ischiadic shaft; obf, obturator foramen; poap, postacetabular process; ppdl, paradiapophyseal lamina; prap, preacetabular process; puf, pubic fenestra; pus, pubic shaft; sac, supra-acetabular crest; sv, sacral vertebra; tp, transverse process; vl, ventral lamina.

vertebrae (2–0), the absence of co-ossification between the ilium and pubis (8–0), the absence of an 'anterolateral scar' on the femur (19–0), and the presence of a gracile and thin fourth trochanter of the femur (23–0). The relative maximum maturity indicates that the holotype of *P. milnerae* was likely not fully skeletally mature, but the relatively large number of characters scored as 1 indicates that it likely also did not represent an early ontogenetic stage. For example, the holotype of *P. milnerae* is interpreted as

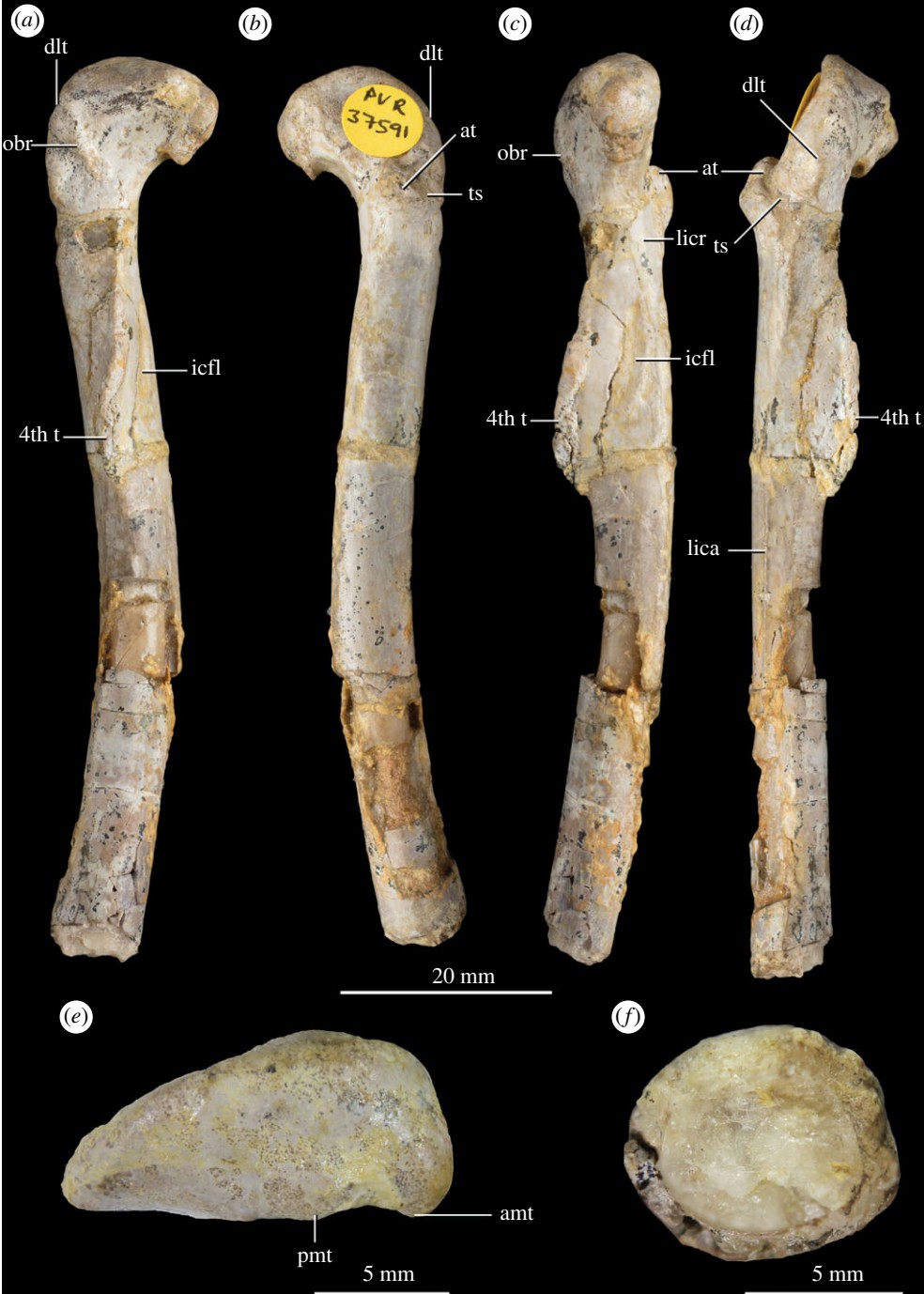

**Figure 3.** Holotype NHMUK PV R 37591 left femur of *P. milnerae* gen. et sp. nov. in (*a*) posteromedial, (*b*) anterolateral, (*c*) anteromedial, (*d*) posterolateral, (*e*) proximal and (*f*) distal view. amt, anteromedial tuber; at, anterior trochanter; icfl, depression associated with the insertion of the *M. caudofemoralis longus*; dlt, dorsolateral trochanter; lica, linea intermuscularis caudalis; lincr, linea intermuscularis cranialis; obr, 'obturator ridge'; pmt, posteromedial tuber; ts, trochanteric shelf; 4th t, fourth trochanter.

more skeletally mature than the type specimens of *Liliensternus liliensterni* and as mature as or potentially more mature than the holotype of *Gojirasaurus quayi* [57], which are considerably larger than the new taxon.

# 3. Morphological description and comparisons

Here, we describe in detail the holotype and referred specimens of *P. milnerae* and compare it exhaustively with other non-averostran neotheropods and the early theropod *Eodromaeus murphi*, which has been found as the sister taxon to Neotheropoda in recent analyses (e.g. [58,59]).

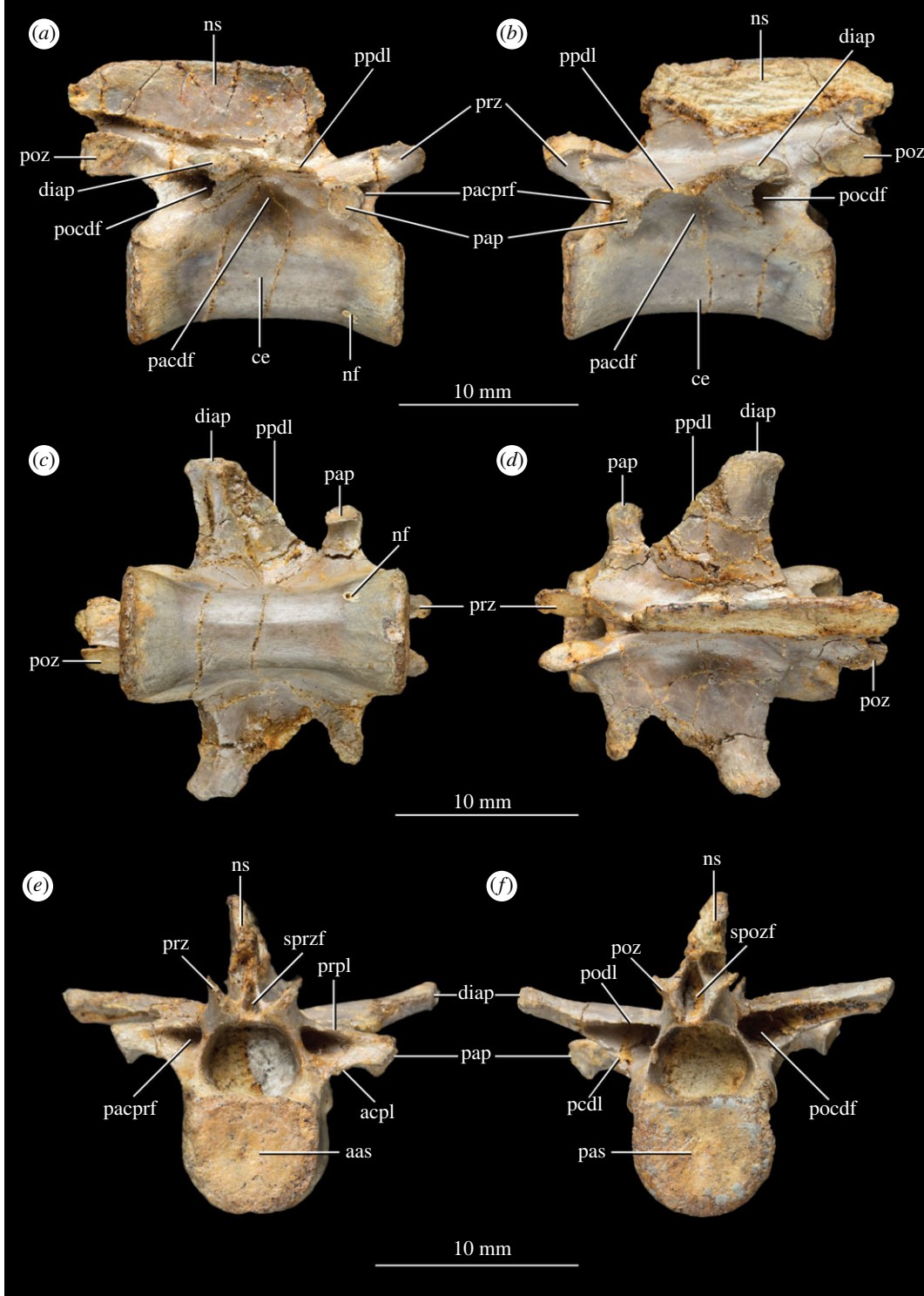

**Figure 4.** Isolated mid to posterior dorsal vertebra NHMUK PV 37596 of *P. milnerae* gen. et sp. nov. in (*a*) right lateral view, (*b*) left lateral view, (*c*) ventral view, (*d*) dorsal view, (*e*) anterior view and (*f*) posterior view. aas, anterior articular surface; acpl, anterior centroparapophyseal lamina; ce, centrum; diap, diapophysis; nf, nutrient foramen; ns, neural spine; pacdf, parapophyseal centrodiapophyseal fossa; pacprf, parapophyseal centroprezygapophyseal fossa; pap, parapophysis; pas, posterior articular surface; pcdl, posterior centrodiapophyseal; pocdf, postzygapophyseal centrodiapophyseal fossa; podl, postzygodiapophyseal lamina; poz, postzygapophysis; ppdl, paradiapophyseal lamina; prpl, prezygaparapohyseal lamina; prz, prezygapophysis; spozf, spinopostzygapophyseal fossa; sprzf, spinoprezygapophyseal fossa.

## 3.1. Axial skeleton

Four largely complete vertebrae are preserved in the holotype specimen NHMUK PV R 37591 (figures 1*b* and 2*a*). Additionally, a partial vertebra is preserved at the anterior end of the preserved vertebral column

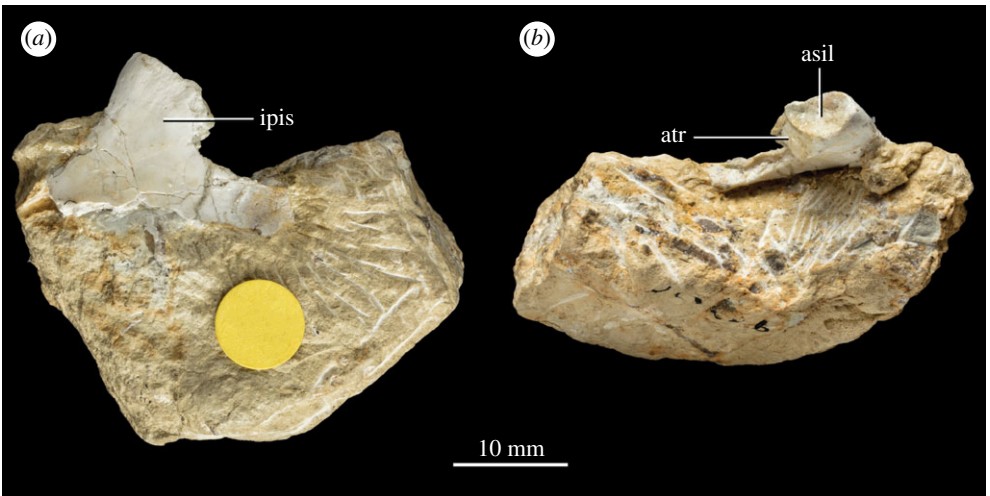

**Figure 5.** Isolated partial left ischium NHMUK PV R 37597 of *P. milnerae* gen. et sp. nov. in (*a*) medial and (*b*) dorsal view. asil, articulation surface with the ilium; atr, antitrochanter; ipis, iliac peduncle of the ischium.

**Table 1.** Vertebral measurements of NHMUK PV R 37591 and NHMUK PV 37596. Measurements were taken with a Sealey electronic vernier calliper. Values preceded by a tilde (∼) indicate an approximated value because a measurement was hampered, either by poor preservation or by the relevant structure being partially covered. dv, dorsal vertebra; sv, sacral vertebra.

| | dv1 | dv2 | sv1 | sv2 | sv3 | sv4 | NHMUK PV 37596 |
|---|---|---|---|---|---|---|---|
| centrum length | — | 14.7 mm | — | 13.3 mm | 11.1 mm | — | 14.6 mm |
| neural spine height | — | — | — | — | — | — | 5.0 mm |
| neural spine length | — | — | — | — | — | — | 14.3 mm |
| width of diapophysis/transverse processes (+ rib) | — | 9.9 mm | ∼5.39 mm | 6.2 mm | 6.5 mm | — | 7.4 mm |
| anterior articular surface centrum height | — | 6.4 mm | — | 5.2 mm | 4.7 mm | — | 5.7 mm |
| posterior articular surface centrum height | ∼7.0 mm | 6.4 mm | 3.94 mm | 4.7 mm | ∼4.2 mm | — | 5.8 mm |
| anteroposterior width transverse proces + rib distal end | n.a. | n.a. | — | 3.7 mm | 5.7 mm | — | n.a. |

and a tiny fragment of a sacral vertebra is preserved at the posterior end of the column. The anterior two vertebrae are identified as dorsal vertebrae and the other four as sacral vertebrae based on the presence of articulations with the ilia, as is explained in more detail below. The isolated dorsal vertebra NHMUK PV R 37596 (figure 4) was not associated with NHMUK PV R 37591 and thus cannot unequivocally be referred to the same individual. However, as is explained below, NHMUK PV R 37596 can be relatively confidently referred to *P. milnerae* based on a strong morphological overlap with NHMUK PV R 37591 and because both specimens were recovered from the same locality.

### 3.1.1. Dorsal vertebrae

The isolated dorsal vertebra NHMUK PV 37596 is virtually complete, undistorted and freed from the matrix (figure 4). The centrum is 14.6 mm long and 2.59 times longer than the height of its anterior articular surface (table 1). The elongation of this centrum matches or closely resembles that of the middle dorsal vertebrae of the coelophysids *C. bauri* (e.g. NMV P231382: *ca* 2.6, middle dorsal vertebra), *M. rhodesiensis* ([60]: 2.33–2.64, D6 and D7 of NHMZ QG 1) and *Pr. triassicus* (SMNS 12591: 2.70–2.91, D7 and D8), whereas in other early neotheropods, the middle–posterior dorsal vertebrae are

proportionally shorter (e.g. *L. liliensterni*, *G. quayi*; *Dilophosaurus wetherilli*, *Dracoraptor hanigani*, *Cryolophosaurus ellioti*; *Sarcosaurus woodi*) [58,61–64]. The ventral surface of NHMUK PV 37596 is concave in lateral view and there is no ventral keel. A single nutrient foramen can be observed close to the anterior end of the centrum on its right ventrolateral side. The anterior and posterior articular surfaces of the centrum are both very slightly concave and transversely broader than tall, resembling the condition in *G. quayi* [61]. By contrast, the centrum is taller than broad in the posterior dorsal vertebrae of *M. rhodesiensis* [60: table 6], *S. woodi* (only the anterior surface is preserved) [58], *Cr. ellioti* [63] and *D. wetherilli* [64], approximately as broad as tall in *L. liliensterni* (HMN MB.R.2175), and both conditions occur in *Eo. murphi* (PVSJ 562). The lateral surfaces of the centrum bear a shallow fossa directly ventral to the connection to the neural arch, as occurs in NHMUK PV R 37591 and other early neotheropods (see below). The neurocentral suture is closed along most of its extension, being only visible on the most posterior region of the neural arch peduncle on both sides of the bone.

The diapophysis is placed on a wide subtrapezoidal or wing-like transverse process (figure 4*d*). In dorsal view, the posterior margin of this process is mainly laterally oriented and slightly concave, whereas the anterior margin is anteromedially to posterolaterally oriented and somewhat sinusoidal. The anteroposteriorly long base of the transverse process and strong posterolateral slating of its anterior margin resemble the condition in the middle dorsal vertebrae of *Eo. murphi* (PVSJ 562) and the coelophysids *C. bauri* (AMNH 7224), *M. rhodesiensis* [60] and *Pr. triassicus* (SMNS 12591). The parapophysis is placed on a strongly developed, narrow and rod-like stalk, but it is considerably less extended laterally than the diapophysis, resembling the condition in the middle–posterior dorsal vertebrae of at least some other early neotheropods (e.g. *L. liliensterni*: HMN MB.R.2175; *M. rhodesiensis* [60]). Both processes are positioned fully on the neural arch and are connected through a thin paradiapophyseal lamina (*sensu* [65]). The diapophysis is located slightly dorsal to the parapophysis (figure 4*a,b*). The articular facet of the diapophysis is oval and anteroposteriorly elongated, whereas the facet of the parapophysis is subcircular. The parapophyseal centrodiapophyseal fossa ventral to the diapophysis is shallow, whereas the parapophyseal centroprezygapophyseal and postzygapophyseal centrodiapophyseal fossae are very deep and framed by pronounced and thin laminae (*sensu* [66]). The laminae framing the parapophyseal prezygapophyseal fossa are the prezygaparapophyseal lamina dorsally and the anterior centroparapophyseal lamina ventrally, whereas the postzygapophyseal centrodiapophyseal fossa is framed by the postzygodiapophyseal lamina dorsally and the posterior centrodiapophyseal lamina ventrally (*sensu* [65]). This pattern of laminae and fossae matches that of a posterior dorsal vertebra of *L. liliensterni* (HMN MB.R.2175 2.22). There are no pneumatic foramina within the fossae. The transition between the transverse process of the diapophysis and the neural spine forms an angle of approximately 90°, and there is no fossa present in this region. The postzygapophyses are closely placed together and their articulation facets face ventrolaterally. A hyposphene is absent between the postzygapophyses and, therefore, there is no accessory intervertebral articulation (figure 4*f*), contrasting with its presence in the middle–posterior dorsal vertebrae of *Eo. murphi* (PVSJ 562), *M. rhodesiensis* [60], *G. quayi* [61], *Cr. ellioti* [63], *S. woodi* [58] and *D. wetherilli* [64]. The articulation facets of the prezygapophyses face dorsomedially (figure 4*e*). Both the spinoprezygapophyseal and spinopostzygapophyseal fossae are very narrow slit-like openings between the pre- and postzygapophyses, respectively [66], and they do not extend onto the surface of the neural spine. The spinopostzygapophyseal fossa is considerably larger than the spinoprezygapophyseal fossa.

The neural spine is proportionally low, being 0.4 times taller than anteroposteriorly long at its base, resembling the condition in the middle and posterior—but not the posteriormost—dorsal vertebrae of *C. bauri* (AMNH 7224) and *M. rhodesiensis* [60]. By contrast, *Eo. murphi* and other early neotheropods (e.g. *D. wetherilli* [64]; *G. quayi* [61]; *Panguraptor lufengensis* [67]) have proportionally taller middle–posterior dorsal neural spines (ratio > 0.5). The anterior margin of the neural spine is located at the level of the parapophysis (figure 4*a,b*). It curves from its base into an anterodorsal direction and subsequently becomes posterodorsally oriented distally. The posterior margin of the neural spine is notched proximally as a result of strongly anterior bowing in lateral view. Distally, the posterior margin is posterodorsally directed and convex. The posterior end of the neural spine is slightly further extended posteriorly than the postzygapophyses. The presence of middle–posterior dorsal vertebrae with a curved posterior margin of the neural spine that overhangs the postzygapophysis also occurs in *C. bauri* (NMV P231382), *Pr. triassicus* (SMNS 12591) and *D. wetherilli* [64]. In some other early theropods, the neural spine also extends posteriorly close to or beyond the level of the postzygapophysis, but the posterior margin of the spine is straight (e.g. *Eo. murphi*: PVSJ 562; *G. quayi* [61]). The distal margin of the neural spine is not transversely expanded (figure 4*e,f*). Most of the distal margin is straight and slightly anteroventrally directed in lateral view. Posteriorly, the distal margin is convex, sloping down towards the posterior margin. The neural spine is 14.3 mm long and 5.0 mm tall (table 1). Its left lateral surface is largely missing (figure 4*b*).

Compared to NHMUK PV R 37596, only a few morphological features of the posterior dorsal vertebrae can be deduced from the articulated vertebral series of NHMUK PV R 37591. Nevertheless, the overall size as well as the relative proportions of the vertebral centrum of NHMUK PV 37596 correspond with that of the posteriormost dorsal vertebra of NHMUK PV R 37591 (e.g. centrum length versus anterior height ratio). Both vertebrae also lack a ventral keel and possess a similar curvature of the ventral margin of the centrum and are amphiplatyan. Furthermore, the diapophysis of NHMUK PV R 37596 has a similar wing shape and concave posterior margin as the penultimate and last dorsal vertebrae of NHMUK PV R 37591. Based on the position of the parapophysis, which is completely located on the neural arch (figure 4a,b), NHMUK PV 37596 can be interpreted as a middle to posterior dorsal vertebra [60,64,68].

The most anterior vertebra of the articulated vertebral series preserved in NHMUK PV R 37591 comprises a partial centrum and neural arch, which preserves the left diapophysis (figures 1b and 2a, c). The neural spine and anterior portion of the vertebra are completely missing. The right side of the vertebra is damaged and partly missing. The centrum is only exposed on its broken right lateral side. There is no visible suture between the centrum and neural arch. On the left side, the diapophysis is complete and has an anteroposteriorly elongate, oval articular facet (figure 1a). The process reaches less far laterally than the diapophysis of the succeeding vertebra (figure 2a). In dorsal view, the diapophysis is subtrapezoidal, with the posterior margin of the diapophysis being slightly concave and the anterior margin being anteriorly curved at its base. This anterior margin represents the paradiapophyseal lamina (sensu [65]) that connected the diapophysis to the parapophysis—which is not preserved—as occurs in NHMUK PV R 37596 and in the middle–posterior dorsal vertebrae of other early neotheropods (e.g. M. rhodesiensis [60]; L. liliensterni: HMN MB.R.2175; D. wetherilli [64]). The left postzygapophysis is poorly preserved and its morphology can therefore not be inferred, but it is still in articulation with the corresponding prezygapophysis of the succeeding dorsal vertebra.

The last dorsal vertebra of NHMUK PV R 37591 comprises a complete centrum and a neural arch missing the neural spine, the right diapophysis and the right postzygapophysis (figures 1b and 2a). Its centrum is 14.7 mm long, being 2.63 times the height of its anterior articular surface (table 1). This ratio matches that of at least some specimens of C. bauri (e.g. [69]: ratio = 1.98–2.87, D11–D13 of AMNH 7228 based on the posterior height of the centrum) and is slightly proportionally longer than those of the posterior dorsal vertebrae of Eo. murphi (PVSJ 562: ratio = 2.0–2.18), Pa. lufengensis ([67], figure 1: ratio = ca 2.0–2.11 in the last two dorsal vertebrae) and M. rhodesiensis ([60]: ratio = 2.10, D13 of NHMZ QG 1). By contrast, these vertebrae are considerably proportionally shorter in L. liliensterni (HMN MB.R.2175 2.22–2.24: ratio = 1.58–1.67, posterior dorsal vertebrae), Lucianovenator bonoi (PVSJ 906: ratio = 1.63, last dorsal vertebra), Lophostropheus airelensis ([70]: ratio = 1.32, last dorsal vertebra), Dr. hanigani (NMW 2015.5G.1–2015.5G.11: ratio = 1.63–1.75, middle–posterior dorsal vertebrae), Cr. ellioti (FMNH PR1821: ratio 1.07, posterior dorsal vertebra), D. wetherilli ([64]: ratio = 1.16–1.52, D10, D11 and D13 of UCMP 37302; ratio = 1.19 D14 of UCMP 77270) and S. woodi ([58]: ratio = ca 1.9, middle–posterior dorsal vertebra of WARMS G678). The ventral surface of the centrum is anteroposteriorly concave and lacks a ventral keel (figure 1b). The centrum is amphiplatyan with very slightly concave anterior and posterior articular surfaces. As in the preceding vertebra, no visible suture is present between the centrum and neural arch. The lateral surface of the centrum bears an anteroposteriorly elongate but shallow fossa just ventral to the articulation with the neural arch, which is a common condition in the middle–posterior dorsal vertebrae of early neotheropods (e.g. L. liliensterni: HMN MB.R.2175; Pr. triassicus: SMNS 12591; Lu. bonoi: PVSJ 906; Lo. airelensis [70]; S. woodi [58]). The last dorsal vertebra possesses only a single articular facet for the rib on each side, located at the end of a transversely wide, wing-like transverse process (figure 2a). In dorsal view, its posterior margin is concave and its anterior margin appears to be somewhat sinusoidal. There is no distinct fossa on the dorsal surface of the base of the transverse process. The articular surfaces of the prezygapophyses face dorsomedially. The articular surface of the left postzygapophysis is poorly preserved. The posterior part of the neural arch is too poorly preserved to corroborate the absence of the hyposphene articular surface seen in NHMUK PV 37596. The prezygapophyses diverge from each other in dorsal view and their tips are well-separated from the median line, contrasting with the subparallel prezygapophyses of S. woodi [58].

### 3.1.2. Sacral vertebrae

The first sacral vertebra of NHMUK PV R 37591 is disarticulated from the rest of the vertebral column and has rotated approximately 90° so that its posterior articular facet faces in a right lateral direction relative to the rest of the vertebral column (figure 1b). Because of this displacement it is apparent that the first sacral

centrum was not co-ossified with adjacent vertebrae. However, based on its position in the vertebral column and the width of its complete, wing-like, right transverse process/rib, it most likely would have attached to the preacetabular process of the ilium (figure 2a). This element is therefore interpreted as a dorsosacral vertebra, and functionally as the first vertebra of the sacrum. The left transverse process/rib is incomplete distally. The neural spine is broken, and its height can therefore not be assessed. However, the base of the spine is preserved, revealing that the spine arose along the entire length of the neural arch. The neural arch is co-ossified with the centrum along its entire anteroposterior length. Its lateral surfaces are lateromedially very thin and markedly laterally convex on both sides, extending considerably further laterally at their mid-height than the centrum. The neural arch encloses a neural canal that is quite large relative to the size of the vertebrae (ratio between the height of neural canal and height of anterior articular surface of the centrum of the second sacral: 4.9 mm/4.7 mm = 1.04) and which is virtually circular in cross-section (figure 1b). The centrum is small relative to the overall size of the vertebra and considerably lateromedially wider than dorsoventrally tall. It has a concave posterior articular surface. The transition between the neural arch and centrum is demarcated by a clear anteroposteriorly directed groove in lateral view (figure 2d). The anteroposterior length of the vertebra cannot be measured because the anterior end is covered in matrix. The right prezygapophysis is visible and has a similar morphology to that of the preceding vertebra and that of NHMUK PV 37596.

The centra of the subsequent two vertebrae of NHMUK PV R 37591 are co-ossified, but their margins can be deduced from slight dorsoventral expansions that demarcate the articulations (figure 2b). It is unclear whether the zygapophyses are also co-ossified between the two vertebrae since they are insufficiently preserved. The second sacral centrum is longer (13.3 mm) than the centrum of the third sacral (11.1 mm; table 1), a condition that also occurs in *Lu. bonoi* (PVSJ 906). Both vertebrae preserve the centrum and the left side of the neural arches. The position of the second sacral vertebra matches that of the first primordial sacral of archosauriforms with two sacral vertebrae. The third sacral probably represents an 'inserted' sacral (sensu [71]) because its rib articulates with the ilium anteriorly to the level of the base of the ischiadic peduncle, where the second primordial sacral vertebra articulates in earlier archosauriforms [71]. Only the left prezygapophysis of the second sacral vertebra is partially preserved, and its articular surface faces dorsomedially. The left ribs of both vertebrae are co-ossified with the left ilium, as occurs in skeletally mature individuals of other early neotheropods [72,73]. The right side of the neural arch and the neural spine are completely missing. The vertebral centra are dorsoventrally shorter than in the preceding vertebrae; the anterior articular surface of the centrum of the second sacral is 5.2 mm tall, whereas that of the posteriormost dorsal vertebra is 6.4 mm (table 1), and the dorsal surfaces of the centra are gently concave (figure 2d). The ventral margins of the centra are slightly anteroposteriorly concave, but considerably less so than the posteriormost dorsal vertebra (figure 1b).

The transverse processes of both vertebrae are co-ossified with their corresponding ribs and are somewhat dorsolaterally extended, in the third sacral vertebra more so than in the second (figure 2a). The co-ossified transverse process and rib of the second sacral vertebra taper slightly distally and have a concave posterior margin and a straight anterior margin. The anteroposterior width of the co-ossified process and rib at their distal end is 3.7 mm. Matrix supports the transverse process of the second sacral vertebra ventrally, and it is therefore not possible to discern its thickness, nor the presence of a ventral lamina. The combined transverse process and rib of the third sacral vertebra is anteroposteriorly 5.7 mm wide at its distal end and therefore wider than that of the second sacral (table 1). It broadens distally and has a concave posterior margin and a slightly convex anterior margin in dorsal view. This transverse process has mostly been freed from the surrounding matrix and is dorsoventrally thin. A very thin, ventrally directed lamina is projected from the anterior margin of the transverse process (figure 2d), giving the entire process an L-shape in lateral view. The ventral extent of this lamina cannot be discerned because its ventral section is surrounded by matrix. The attachment sites on the medial surface of the ilium for the second and third sacral ribs are positioned on an anteroposteriorly directed rim that extends further posterior to these attachment sites (figure 1b). This rim terminates posteriorly at a dorsoventrally oriented thickening of the medial surface of the ilium. This thickening connects the rim with the ridge that forms the dorsomedial margin of the brevis fossa. This thickening represents an attachment site scar for the fourth sacral rib, which is not preserved.

A small fragment of the centrum of the fourth sacral vertebra is preserved and co-ossified with the centrum of the third sacral of NHMUK PV R 37591. The second, third and fourth sacral centra form a straight structure in lateral view, resembling *Lu. bonoi* (PVSJ 899), *C. bauri* (CMNH 10971) and *M. rhodesiensis* [60], but contrasting with the dorsally arched sacrum of '*Syntarsus*' *kayentakatae* (TMM 43688-1). There are no visible additional attachment site scars preserved on the medial surface of the ilium. Nevertheless, the presence of another posterior sacral vertebra cannot be ruled out. Therefore, *P. milnerae* possessed at least four sacral vertebrae.

**Table 2.** Measurements of the appendicular skeleton of NHMUK PV R 37591. Measurements were taken with a Sealey electronic vernier calliper. The circumference of the femoral shaft was measured by running a piece of string around the shaft and subsequently measuring the length of the amount of string with the calliper. Values in parentheses represent incomplete values, due to the relevant structure being incompletely preserved.

| | |
|---|---|
| max. length left ilium across iliac blade | 55.8 mm |
| max. length left ilium across peduncles | 26.0 mm |
| max. length left acetabulum | 16.7 mm |
| max. dorsoventral height left acetabulum | 17.6 mm |
| max. length right pubis (excluding imprint) | (63.2 mm) |
| max. length right pubis (including imprint) | (74.8 mm) |
| max. length left ischium | (26.6 mm) |
| max. length left femur | (86.3 mm) |
| estimated length left femur | 102.1 mm |
| max. width proximal head left femur | 15.1 mm |
| min. circumference shaft of left femur | 25.08 mm |

## 3.2. Appendicular skeleton

The pelvic girdle of NHMUK PV R 37591 comprises a complete left hemipelvis excluding the distal ends of the left pubis and ischium, as well as a largely complete right pubis and ischium (figure 1). In addition, a partial left ischium, NHMUK PV R 37597 (figure 5), comprising the iliac peduncle and part of the ischial plate, can be confidently assigned to *P. milnerae* based on the presence and shape of the antitrochanter and the concavity on the acetabular rim, which are in correspondence with the morphology of the left ischium of NHMUK PV R 37591. NHMUK PV R 37597 is slightly larger than the holotype. Finally, a largely complete left femur (figure 3) is referred to the same individual as the articulated pelvic girdle and vertebral series and together comprise the holotype (NHMUK PV R 37591) of *P. milnerae* (see Systematic Palaeontology section). Measurements of the appendicular skeleton of NHMUK PV R 37591 are provided in table 2.

### 3.2.1. Ilium

The left ilium of NHMUK PV R 37591 is virtually complete except for some damage on the posterior end of the postacetabular process, and is preserved in articulation with the left pubis and ischium, and with the second and third sacral ribs (figures 1 and 2). The dorsal margin of the iliac blade is straight to slightly convex on its middle and posterior portions in lateral view, which comprises the section dorsal to the supra-acetabular crest and the elongate postacetabular process (figure 1*a*). Anteriorly, the dorsal margin of the preacetabular process is more distinctly convex and forms a relatively abrupt anteroventral transition to the anterior margin of the preacetabular process, resembling the condition in *S. woodi* [58]. By contrast, the dorsal margin of the preacetabular process is straight or only slightly convex along its entire anteroposterior length in other non-averostran neotheropods (e.g. *C. bauri*: USNM 529376; *L. liliensterni*: HMN MB.R.2175; *Lu. bonoi*: PVSJ 906; *Coelophysis* sp.: PEFO 21373/ UCMP 129618; *Notatesseraeraptor frickensis* [74]; *M. rhodesiensis*: NHMZ QG 1; *D. wetherilli* [64]). The dorsal margin of the iliac blade of *P. milnerae* possesses a somewhat thickened, mostly flat surface that faces slightly laterally. This flat surface extends along most of the bone, with the exception of the anteriormost region of the preacetabular process and starts to taper anteriorly at the mid-length of this process. In the posterior region of the iliac blade, this flat surface extends ventrally as a raised region to occupy the entire dorsoventral height of the lateral surface on the posterior end of the postacetabular process. It has been inferred that the anterior rim of this raised surface probably delimited the attachment site of the *M. iliofemoralis* [73,75]. This same condition occurs in *C. bauri* (USNM 529376), *Lu. bonoi* (PVSJ 899, 906), *Coelophysis* sp. (PEFO 21373/UCMP 129618), '*Syntarsus*' *kayentakatae* [76] and *M. rhodesiensis* (NHMZ QG 1), but not in other early neotheropods [73]. The anterior margin of the preacetabular process is continuously rounded and extends considerably further anterior than the pubic peduncle of the ilium, as occurs in other neotheropods [77]. The

preacetabular process is transversely very thin (i.e. laminar) and slightly medially curved in dorsal view. The ventral margin of the preacetabular process is slightly convex and oriented somewhat anteroventrally to posterodorsally. However, the overall orientation of the preacetabular process is anteriorly facing and a broad gap separates it from the pubic peduncle. This morphology corresponds to that of most theropods, but contrasts with the anteroventrally directed processes of *S. woodi* and some ceratosaurs (e.g. *Ceratosaurus nasicornis*, *Eoabelisaurus mefi*) [58]. At its posterodorsal end, the postacetabular process curves abruptly posteroventrally and the posteroventral end of the process is formed by an acute angle of approximately 65° in lateral view. By contrast, the posteroventral corner of the postacetabular process is approximately right-angled or slightly acute in other non-averostran neotheropods (e.g. *C. bauri*: USNM 529376; *L. liliensterni*: HMN MB.R.2175; *Lu. bonoi*: PVSJ 906; *Coelophysis* sp.: PEFO 21373/UCMP 129618; '*Syntarsus*' *kayentakatae*: [73]; *M. rhodesiensis*: NHMZ QG 1; *D. wetherilli*: [64]; *S. woodi*: NHMUK PV R4840). A notch on the posterior end of the postacetabular process, as has been described for various coelophysoid taxa (e.g. *C. bauri*, *Coelophysis* sp., *M. rhodesiensis*, '*Syntarsus*' *kayentakatae*) [73], is absent. In dorsal view, the ilium is oriented approximately straight anteroposteriorly (figure 2*a*) and the postaccetabular process expands gradually laterally towards its posterior end, resembling the condition in *L. liliensterni* (HMN MB.R.2175), *Lu. bonoi* (PVSJ 906) and *D. wetherilli* (UCMP 37302). By contrast, the postacetabular process is distinctly more laterally expanded, extending beyond the level of the outer rim of the supra-acetabular crest in dorsal view, in *C. bauri* (USNM 529376), *Coelophysis* sp. (PEFO 21373/UCMP 129618) and *M. rhodesiensis* [60].

The lateral surface of the iliac blade is concave along its entire anteroposterior length. A shallow, indistinctly rimmed fossa is present immediately dorsal to the supra-acetabular crest and this region lacks the vertical ridge present in *Lo. airelensis* [70,78]. The ventral margin of the postacetabular process is formed by a distinct and sharp brevis shelf (figure 1*a*). The concave portion of the postacetabular process positioned medioventrally to this shelf is the brevis fossa [75]. This fossa is inferred to have formed the attachment site for the *M. caudofemoralis brevis* and is mediodorsally framed by a distinct ridge (figure 1*b*). The brevis fossa is only visible in lateral view in its anterior portion. The remainder of the fossa faces ventrally or medioventrally and is obscured by the brevis shelf in lateral view, a condition typical of neotheropods [71].

The acetabulum is fully perforated and mostly formed by the ilium (figure 1*a*). On the posterior surface of the acetabulum, a posteriorly well-delimited, crescent-shaped rugosity is present, which represents the antitrochanter. The dorsal portion of the antitrochanter is positioned on the ilium, whereas most of its surface is present on the ischial portion of the acetabular margin. The development of this antitrochanter closely resembles those observed in *M. rhodesiensis* (NHMZ QG 1), *C. bauri* (USNM 529376), *Coelophysis* sp. (PEFO 21373/UCMP 129618), '*Syntarsus*' *kayentakatae* [73] and *Lu. bonoi* (PVSJ 906). Dorsally, the acetabulum is framed by a pronounced supra-acetabular crest, which projects laterally and slightly ventrally (figures 1*a* and 2*c*). The rim of the crest extends close to the connection with the pubis anteriorly and to the origin of the brevis shelf posteriorly. However, the supra-acetabular crest and the brevis shelf do not form the continuous, laterally well-developed ridge present in *M. rhodesiensis* (NHMZ QG 1), *C. bauri* (USNM 529376), *Lo. airelensis* [78], *Pr. triassicus* (SMNS 12591), '*Syntarsus*' *kayentakatae* [73] and *Lu. bonoi* (PVSJ 906). The condition of *P. milnerae* resembles that of *L. liliensterni* (HMN MB.R.2175), *Coelophysis* sp. (PEFO 21373/UCMP 129618), *D. wetherilli* [64] and *S. woodi* [58]. The pubic peduncle is anteroventrally oriented, whereas the ischiadic peduncle is considerably more vertically directed, facing only slightly posteroventrally. The suture between the ilium and pubis is completely unfused. The suture with the ischium is unfused along its posterior portion, but on its anterior portion, which is located across the antitrochanter and part of the acetabulum, the suture is closed and the elements are indistinguishably fused.

### 3.2.2. Pubis

Both the left and the right pubes of NHMUK PV R 37591 are largely complete and in articulation with each other (figure 2*c*). Both elements lack the distal end of the pubic shaft, but the shaft extends further distally in the right element than the left. Overall, the preservation of the left element is superior to that of the right element since the surface of the latter is damaged in several places. Therefore, the description of the pubis is largely based on the left element. The shaft of the pubis is anteroventrally directed and elongate (figure 1*a*). Its extent is considerable but cannot be fully assessed because the distal end is missing on both sides. The longest preserved pubis, the right element, is 63.2 mm long. When including the imprint of the pubic shaft in the matrix, which reaches further distally but likely does not represent the distal terminus of the pubes, the maximum length is 74.8 mm. The shaft is rod-like

with a plate-like medial apron, which is lateromedially wide and anteroposteriorly flat (figure 2b,c). The anterior surface of the shaft is convex and, correspondingly, the posterior surface is concave. The pubic shaft is slightly anteriorly curved in lateral view as in *C. bauri* (AMNH 7223, 7224), *M. rhodesiensis* [60], 'Syntarsus' kayentakatae [79], *Dr. hanigani* (NMW 2015.5G.1–2015.5G.11), *Pr. triassicus* (SMNS 12591), *N. frickensis* (SMF 06-1) and *G. quayi* [61]. A higher degree of curvature occurs in *Eo. murphi* (PVSJ 562). By contrast, the pubic shaft is straight in *L. liliensterni* (HMN MB.R.2175) and *D. wetherilli* [64]. Distally, the shaft gradually narrows mediolaterally. Because the distal end is missing, it cannot be determined whether *P. milnerae* possessed an expanded pubic boot. The pubic shaft meets its antimere distally, but it is unclear whether the shafts were also connected proximally or separated by a pubic foramen [75], since the shafts are covered by the matrix on both sides proximally.

The anterior margin of the proximal portion of the pubis is smooth. The anterior portion of the acetabular contribution of the pubis is slightly rugose and a faint ridge is formed on its lateral margin (figure 1a). The pubis bears two openings on its ventrolateral surface proximally (figures 1a and 2b). The dorsal opening is the obturator foramen. It is oval and approximately four times longer anteroposteriorly than tall dorsoventrally. The obturator foramen occurs widely in archosauromorphs [71,80] but is lost in most averostran theropods, which is attributable to a reduction in the ossification of the puboischiadic plate, resulting in the confluence of the obturator foramen with the puboischiadic fenestra [75]. The considerably larger opening present ventral to the obturator foramen represents the pubic fenestra, which also occurs in *M. rhodesiensis* [60], *Segisaurus halli* [81,82], *C. bauri* [69] and *G. quayi* [61]. The pubic plate of *P. milnerae* is well-ossified as it articulates with its antimere along its entire ventral margin. Among theropods, this ventral median contact is a rarely preserved feature that has previously only been described or figured for *M. rhodesiensis* [60] and *Torvosaurus tanneri* [83]. A suture between the pubis and ilium can be clearly discerned. However, it is unclear whether a suture is also present between the pubis and ischium or whether these elements were fully fused, since several cracks obscure this region. A low tuberosity for the probable origin of the *M. ambiens* is present on the anterolateral surface of the bone in transition between the proximal end and the shaft.

### 3.2.3. Ischium

Ischia are preserved on both sides of the pelvis of NHMUK PV R 37591. The preserved portion of the right ischium comprises the ventral section of the ischiadic plate, including a complete ventral margin, and the base of the ischiadic shaft (figure 1b). The left ischium also only preserves the base of the ischiadic shaft, but the entire proximal part of the element is preserved (figure 1a) and therefore the description of the ischium is mostly based on this element. The maximum length of the left ischium, measured from the distalmost preserved end of the shaft to the connection between the ischium and pubis on the margin of the acetabulum, is 26.6 mm. The base of the shaft of the ischium is posteroventrally directed (figure 1a) and the shaft is considerably lateromedially narrower at its base than the pubis (figure 2b). The ventral margin of the ischiadic plate connects its antimere along its anteroposterior length and is continuous with the ventral margin of the pubic plate. At its posterior end, the ventral margin bears a notch that separates it from the base of the ischiadic shaft (figure 1a). This notch occurs widely among early neotheropods and has previously been considered as a synapomorphy for the group [77]. The notch of *P. milnerae* is shallow as a result of the absence of a posteroventrally oriented projection of the obturator plate. By contrast, this projection is present and forms a deep, V-shaped or U-shaped notch in lateral view in *M. rhodesiensis* (NHMZ QG 1), *Dr. hanigani* [84], *D. wetherilli* (UCMP 37302) and *Tachiraptor admirabilis* [85]. There are no openings on the ischiadic plate, contrasting with the presence of an ischial foramen in *Segisaurus halli* [82]. In contrast with the pubic plates, which are only connected at their distalmost ventral margin, the ischiadic plates form a taller connection that occurs medially for slightly more than half their dorsoventral height. The iliac peduncle of the ischium is clearly separated by a deep concavity from the ventral margin of the acetabulum formed by the ischium. The ventral margin bears a low but distinct ridge (figure 1a). The antitrochanter is largely formed by the ischium. It is a flat rugose surface that is clearly demarcated by a rounded convex ventral ridge.

The isolated partial left ischium NHMUK PV R 37597 comprises the iliac peduncle and part of the ischial plate (figure 5). The medial side of the specimen is fully exposed (figure 5a), whereas the lateral side is largely covered by matrix, only exposing the iliac peduncle. The articular facet with the ilium is anteroposteriorly elongate, oval and concave (figure 5b). Anterolaterally, it bears a clear rim that demarcates the articular surface from the portion of the antitrochanter formed by the ischium. The surface of the antitrochanter is slightly rugose and rounded ventrally. The acetabular

margin of the ischium bears a clear concavity ventral to the antitrochanter (figure 5a). The posterior margin of the ischium is slightly concave and represents the transition from the iliac peduncle to the posterior margin of the shaft of the ischium, which is otherwise not preserved. The ischial shaft is straight as far as it is preserved and rod-like, forming a subtriangular cross-section with its counterpart where they are broken off.

### 3.2.4. Femur

Only the left femur of the holotype NHMUK PV R 37591 is preserved. Originally, this element was displaced and positioned lateroventrally to the right hemipelvis in the same block, but it had been prepared free of matrix by the time of the thesis of Warrener [41]. Based on its association, overall size and morphology, the femur can be confidently identified as belonging to the same individual as the pelvis and associated vertebrae. The femur is very well-preserved and comprises a complete proximal end and most of the shaft, only missing the distal end and the distalmost part of the shaft (figure 3). The preserved length of the femur is 86.3 mm. The proximal femoral head is inturned and would have been oriented anteromedially to articulate with the acetabulum of the pelvis, as in other early dinosaurs [86,87]. The maximum width of the proximal head of the femur is 15.1 mm. On its ventral end, the proximal head bears a clear lip, forming a distinctly concave emargination on the transition with the shaft (figure 3a,b), as is present in other dinosaurs [71]. Directly dorsal to this lip, a small, hook-shaped anteromedial tuber (sensu [71], = posteromedial tuber of [80]) projects from the femoral head medially (figure 3e), as occurs in other early neotheropods (e.g. C. bauri, M. rhodesiensis, D. wetherilli, L. liliensterni) [77]. By contrast, the posteromedial tuber (sensu [71], = posterior tuber of [80,88]), positioned posterolaterally to the anteromedial tuber, is poorly developed, forming a slight convexity on the posteromedial margin of the proximal head. Posterolateral to this, the posteromedial margin of the femoral head is slightly depressed, representing the *facies articularis antitrochanterica*. The anterolateral margin of the femoral head is continuously convex along its length as a result of the presence of an anterolateral tuber (sensu [71], = anteromedial tuber of [80]). The proximal surface of the femur lacks a longitudinal groove, a feature which occurs in morphologically immature specimens of M. rhodesiensis but which is absent in skeletally mature specimens of this taxon [54].

The dorsolateral or 'greater' trochanter is a mound-like tuberosity positioned on the lateral surface of the femur distal to the proximal head (figure 3). Among early neotheropods, a distomedially directed ridge is present directly posteromedial to the dorsolateral trochanter in some, generally relatively mature, specimens of the coelophysoid taxa M. rhodesiensis, C. bauri [54], 'Syntarsus' kayentakatae (MNA V2623) and Segisaurus halli (UCMP 32101). This ridge was identified as an 'obturator ridge' and considered to probably represent an attachment site for *Mm. puboischiofemoralis externi* by Raath [60]. A very distinct ridge positioned and oriented as in these taxa is present on the femur of P. milnerae (figure 3a), and we therefore refer to this structure as the 'obturator ridge'. The 'obturator ridge' is raised considerably, particularly on its posterolateral portion, and its surface is rugose. Another ridge that originates proximally to the 'obturator ridge', and which is oriented proximomedially, occurs in several specimens of M. rhodesiensis and C. bauri [54]. However, as in Segisaurus halli (UCMP 32101) and 'Syntarsus' kayentakatae (MNA V2623), this ridge is absent in NHMUK PV R 37591. Directly distal to the proximal head is a very large, rugose anterior trochanter that is subtriangular in anterior view projects from the anterolateral side of the femur (figure 3). The anterior trochanter is projected proximoanteriorly to proximolaterally from the shaft and gradually merges with the shaft except for its proximomedial margin, which is separated from the shaft by a shallow cleft (figure 3d). This cleft occurs widely among neotheropods (e.g. D. wetherilli [64]; S. woodi [58]; C. bauri and M. rhodesiensis [54]; 'Syntarsus' kayentakatae: MNA V2623; L. liliensterni: HMN MB.R.2175), whereas it is absent in Eo. murphi (PVSJ 562). This concavity opens onto a ridge on the posterolateral side of the anterior trochanter. This ridge, the trochanteric shelf, likely forms an attachment site for the M. iliofemoralis, and is well-developed in many early ornithodirans [55,86,89]. In P. milnerae, the trochanteric shelf is well-developed but only shortly projected, terminating on the posterolateral margin of the shaft. The most posterolateral region of the trochanteric shelf is damaged, thus whether it was connected to the linea intermuscularis caudalis (see below), as in other relatively skeletally mature early neotheropod specimens (e.g. M. rhodesiensis [54]), is unknown. The trochanteric shelf has a marked posterodistal orientation. A straight, mainly lateromedially oriented tuberosity extends along most of the posterior surface of the bone dorsal to the fourth trochanter. It was probably connected to the trochanteric shelf and the linea intermuscularis caudalis but the region of

contact between the structures is damaged. This tuberosity probably represents the insertion scar of the *M. caudifemoralis brevis* and occurs in other skeletally mature individuals of early dinosauriforms [54].

The fourth trochanter is an elongate flange positioned on the posteromedial surface of the shaft (figure 3). It is a thin, 23.1 mm long, crest-like flange with a slightly convex anteromedial face and a concave posterolateral face. Proximally, the flange raises gradually, but on its distal end, which is slightly proximal to the mid-length of the shaft, the flange decreases in height more abruptly (figure 3*c*), but without forming the asymmetric fourth trochanter of most early non-neotheropod dinosaurs (e.g. *Herrerasaurus ischigualastensis* [90]; *Eo. murphi*: PVSJ 562; *Eoraptor lunensis* [91]). The fourth trochanter of *P. milnerae* is relatively well-developed posteriorly, resembling the condition in *L. liliensterni* (HMN MB.R.2175), *Pr. triassicus* (SMNS 12591), *Cr. ellioti* [63] and *D. wetherilli* [64], but contrasting with the very low trochanter of *S. woodi* [58], *M. rhodesiensis* (NHMUK PV R9584, cast of NHMZ QG 1), *Segisaurus halli* (UCMP 32101) and '*Syntarsus*' *kayentakatae* (MCZ 9175, cast of MNA V2623). The fourth trochanter is mainly longitudinally oriented, but it slants slightly from proximomedially to distolaterally in posterior view. On the anteromedial surface of the femur, there is a slightly rugose depression at the base of the fourth trochanter that is demarcated medially by a crescent-shaped ridge. This depression likely represents an insertion area for the *M. caudofemoralis longus*. The femoral shaft is long and slender and anteriorly curved along its entire length. The anterolateral and posteromedial surfaces of the shaft are clearly delineated by two ridges or intermuscular lines extending along the length of the shaft. The anteromedial *linea intermuscularis cranialis* probably demarcated the border between the *M. femorotibialis externus* and *M. femorotibialis internus* (figure 3*c*), whereas the posterolateral *linea intermuscularis caudalis* (figure 3*d*) probably separated the *M. femorotibialis externus* and the *Mm. adductor femoris 1 and 2* [86]. The broken distal end of the femur reveals that the cortex of the shaft was thin (figure 3*f*), 0.8 mm thick, whereas the cross-section of the femur is 8.5 mm long at its greatest extent. The broken distal end of the shaft reveals that it is oval in cross-section, being slightly transversely broader than anteroposteriorly deep, and filled by a crystalline matrix.

# 4. Methods

## 4.1. Phylogenetic analysis

*Pendraig milnerae* was incorporated into the data matrix of Novas *et al.* [92], which represents the most recent iteration of the matrix originally published by Nesbitt *et al.* [89], which has also been modified in various other studies [58,67,84,93–100]. The Nexus and TNT files of the matrix, as well as a list of modifications made to the scoring of various characters for certain taxa and the script to calculate the consistency and retention indices (CI and RI, respectively), are included as electronic supplementary material. Following Novas *et al.* [92], '*Nhandumirim waldsangae*', '*Velociraptor mongoliensis*', '*Powellvenator podocitus* holotype' and '*Lepidus praecisio* combined' were deactivated, whereas '*Powellvenator podocitus*' and '*Lepidus praecisio* holotype' remained active. Recent new information about non-dinosaurian avemetatarsalians is not included in this matrix [101] and the members of this part of the tree should be considered as outgroups without a proper test of their interrelationships here.

The data matrix was analysed under the equally weighted parsimony criterion using TNT 1.5 [102], with *Erythrosuchus africanus* selected as the outgroup. As in Novas *et al.* [92], the following characters were treated as additive: 9, 18, 30, 67, 128–129, 174, 184, 197, 207, 213, 219, 231, 236, 248, 253–254, 273, 329, 343, 345, 347, 349, 354, 366, 371, 374, 377–379, 383–384. Using the Traditional Search algorithm, a heuristic search of 1000 replications of Wagner trees with random addition sequence was performed, followed by TBR branch swapping holding 10 trees per replicate. Homoplasy indices were calculated with a script that does not take into account *a priori* deactivated terminals (see electronic supplementary material, Data—StatsB.run). Bremer and bootstrap support values were calculated, the latter using a 'Traditional search' at 1000 iterations.

## 4.2. Femoral length estimation and ancestral state reconstruction

To test whether *P. milnerae* might represent an insular dwarf as has been suggested for other reptiles of the fissure fills faunas [19–21], ancestral state reconstruction of the femoral length was performed on a sample of early theropods and closely related taxa to compare the relative size of *P. milnerae* in a quantitative phylogenetic framework. Even though femoral circumference is generally considered to represent the best proxy for body size [103], femur length was used since this measurement is

available in a wide range of early theropod specimens and because it is expected to represent a reliable proxy for our relatively narrow phylogenetic sample (*sensu* [100]). The R script for the ancestral state reconstruction analyses and associated data files are included as electronic supplementary material.

The only known femur of *P. milnerae* (NHMUK PV R 37591) is largely complete, but the distal end is missing. We compiled a dataset of femoral measurements from several early theropods, composed of femoral length, maximal longitudinal width of the proximal head, maximal depth of the proximal head and minimal femoral circumference (electronic supplementary material, table S1). We subsequently performed regression analyses in the software environment R v. 4.0.4 [104] with all values having been log-transformed. We used the predict function to estimate the femoral length and calculate confidence intervals for NHMUK PV R 37591. We included the R script for the femoral regression analyses and associated data file as electronic supplementary material.

Ancestral state reconstruction was performed on a dataset of femoral lengths of early theropods and closely related taxa, including the estimated femoral length of *P. milnerae*. The inclusion of taxa represented by immature specimens likely has a considerable effect on the outcome of body size analyses for early theropods, resulting in the underestimation of ancestral femoral lengths [100]. Therefore, in addition to the analysis including all taxa, we performed an alternative analysis in which we excluded theropod taxa that (i) were less mature than *P. milnerae* (a maximum maturity score lower than 19 based on the maturity assessment matrix of Griffin [54]) and (ii) had a femoral length of less than 21 cm (for maturity scores of relevant taxa, see electronic supplementary material, figure S11). Theropod taxa with a femoral length of over 21 cm were included, even when they were represented by immature specimens, because their inclusion was considered not to contribute to an underestimation of ancestral femoral lengths due to their large size and because early large-bodied theropods are exclusively known from immature specimens [57,100]. Following these criteria, the taxa excluded for the alternative analysis are: *Pa. lufengensis*, *Le. praecisio*, *Dr. hanigani* and *Po. podocitus*.

A total of six most parsimonious trees (MPTs) were found for the phylogenetic analysis (see below). Since the focus of our analysis is on *P. milnerae*, we performed the ancestral state reconstruction analyses on three MPTs that represent the different, most parsimonious resolutions for the position of *P. milnerae*, but which otherwise share the same topology. To time-calibrate the trees, the first and last occurrences (which for most theropods and early dinosaurs represent stratigraphic uncertainty rather than true ranges) of all taxa were obtained from the literature, using the International Chronographic Chart [105] to determine the delimitations of the various geological stages (e.g. *P. milnerae*, earliest occurrence late Norian, latest occurrence possibly late Rhaetian; 214.7–201.3 Ma). Branch lengths were calculated using the timepaleophy() function from the paleotree package [106] under three different settings for minimum branch lengths (mbl), 1.0, 0.5 and 0.1 Myr, to test for the influence of different parameter settings on our results. The analyses were performed using the function anc.ml() from the phytools package [107] to calculate ancestral states under a Brownian model of evolution using maximum likelihood.

# 5. Results

## 5.1. Phylogenetic analysis

The phylogenetic analysis found six MPTs of 1360 steps with a CI of 0.346 and a RI of 0.676. *Pendraig milnerae* is found within Coelophysoidea but outside Coelophysidae (i.e. *Coelophysis*, *Procompsognathus*, their most recent common ancestor and all of its descendants) [51] in all MPTs. Within the strict consensus tree (SCT) generated from the MPTs it forms a polytomy together with *Po. podocitus*, *Lu. bonoi*, and a clade composed of '*Syntarsus*' *kayentakatae* and Coelophysidae (figure 6). Among these MPTs, *Lu. bonoi* and *Po. podocitus* alternate as the sister taxon to the '*Syntarsus*' *kayentakatae* + Coelophysidae clade. Three additional steps are required to force *P. milnerae* outside Coelophysoidea, one step for *P. milnerae* to be the sister taxon of the '*Syntarsus*' *kayentakatae* + Coelophysidae clade, and five steps for it to be the sister taxon to or be part of Coelophysidae. The following synapomorphies support the placement of *P. milnerae* within Coelophysoidea: presence of a flat dorsal margin of the ilium dorsal to the supra-acetabular rim (200: $0 \rightarrow 1$) and the presence of a rounded ridge or dorsolateral trochanter on the dorsolateral margin of the proximal portion of the femur (230: $1 \rightarrow 2$). The presence of elongate posterior dorsal vertebrae with a centrum length that is at least twice the height of their anterior articular surface (329: $1 \rightarrow 2$) is present in *P. milnerae* and represents a common synapomorphy for the clade composed of all coelophysoids except *Pa. lufengensis*. The following synapomorphies found for the clade composed of Coelophysidae and '*Syntarsus*' *kayentakatae* are absent in *P. milnerae*: a

royalsocietypublishing.org/journal/rsos　R. Soc. Open Sci. **8**: 210915

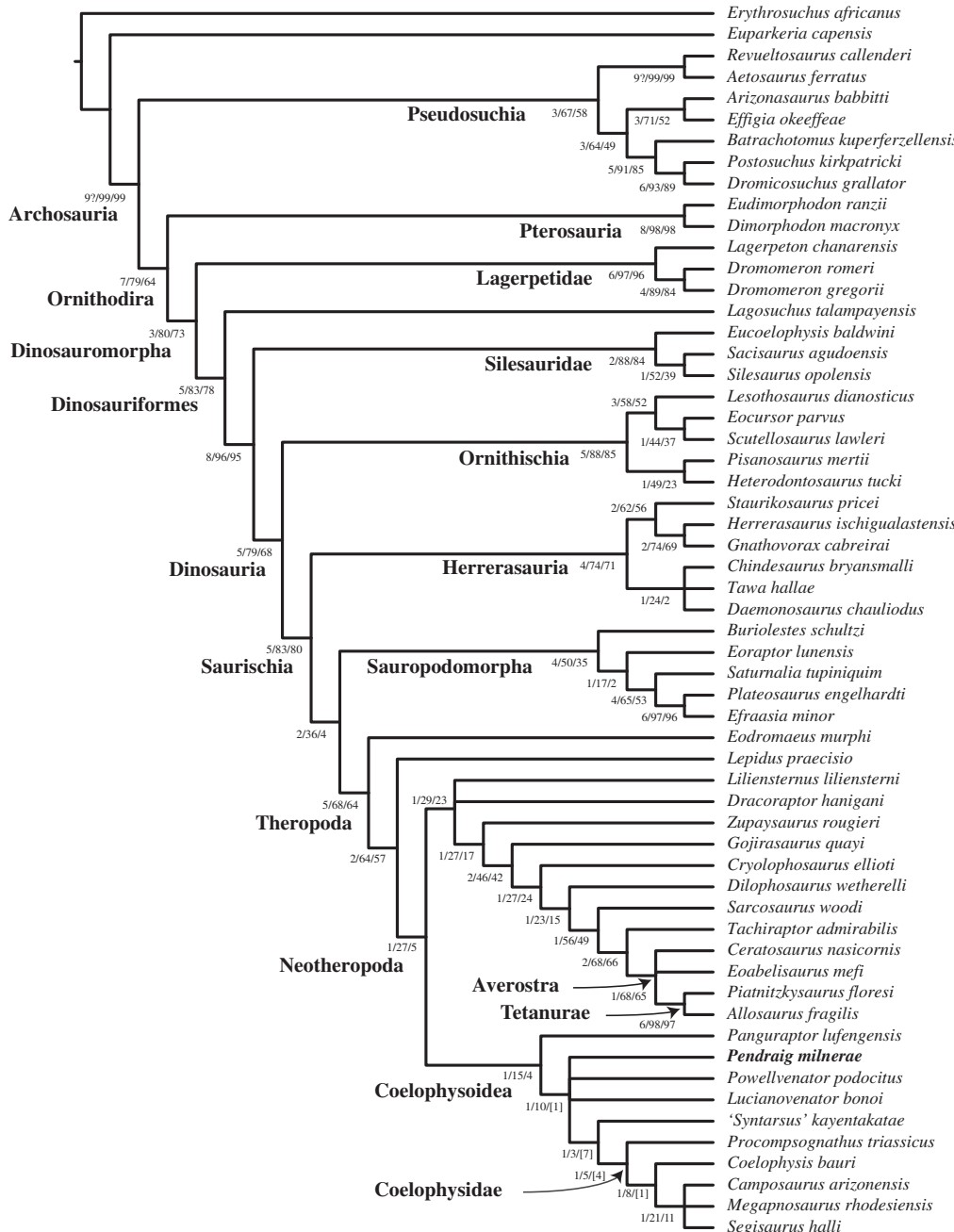

**Figure 6.** Strict consensus of six most parsimonious trees of the phylogenetic analysis. Bremer support, absolute bootstrap frequency and GC bootstrap frequency values are indicated at each branch in that order.

ventrally projected supra-acetabular crest of the ilium (189: $0 \rightarrow 1$; 0 in *P. milnerae* and 0 and 1 in *C. bauri*); a notched or indented posterior margin of the postacetabular process of the ilium in lateral view (194: $0 \rightarrow 1$; 0 in *P. milnerae*); the presence of a laterally well-developed and sharp brevis shelf on the ilium (197: $1 \rightarrow 2$; 1 in *P. milnerae*); a posteriorly poorly developed fourth trochanter that is only raised from the shaft as a low ridge (377: $0 \rightarrow 1$; 0 in *P. milnerae* and 0 or 1 in *C. bauri*). The only unambiguous apomorphy of Coelophysidae based on the phylogenetic analysis, the presence of a diagonal tuberosity on the anterior surface of the distal end of tibia (333: $0 \rightarrow 1$), could not be scored for *P. milnerae*.

## 5.2. Femoral length estimation

We found that the femoral length is significantly correlated with all other variables that we considered (maximal longitudinal width of the proximal head: $p < 2.2 \times 10^{-16}$, $R^2$: 0.9691 (multiple), 0.968 (adjusted); maximal depth of the proximal head: $p = 3.67 \times 10^{-8}$, $R^2$: 0.9424 (multiple), 0.9372 (adjusted); and

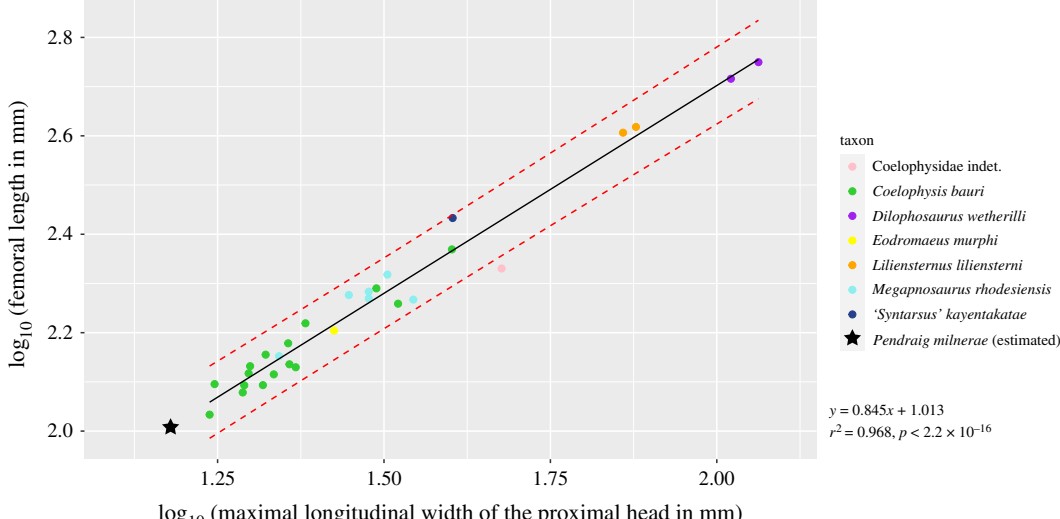

**Figure 7.** Log$_{10}$-transformed bivariate plot of the longitudinal width of proximal head of the femur versus the femoral length of early theropods. The solid black line represents the linear regression described by the formula, and the red dotted lines represent the 95% confidence intervals.

minimal femoral circumference: $p = 7.173 \times 10^{-6}$, $R^2$: 0.9717 (multiple), 0.967 (adjusted)). Since the maximal longitudinal width of the proximal head has the most significant correlation with femoral length and is the most widely available measurement in the studied sample, we used this variable to estimate the femoral length of NHMUK PV R 37591 (figure 7). Based on this, the reconstructed femoral length of NHMUK PV R 37591 is 10.21 cm (lower 95% CI: 8.60 cm; upper 95% CI: 12.11 cm). The results of the ancestral state reconstruction analysis are discussed below.

# 6. Discussion

## 6.1. Phylogenetic implications

The scoring of the ontogenetic characters formulated by Griffin [54] suggests that the holotype NHMUK PV R 37591 of *P. milnerae* is skeletally immature with regard to certain skeletal features (electronic supplementary material, figure S11A). It is therefore important to consider that skeletally immature specimens are often recovered in a different (often less derived) phylogenetic position than mature specimens of the same species (e.g. [100,108,109]). The phylogenetic analysis recovered *P. milnerae* as a non-coelophysid coelophysoid (figure 6), but with low support values for Coelophysoidea and its internal nodes. Nevertheless, the referral of *P. milnerae* to Coelophysoidea is supported by the presence of several typically coelophysoid character states, such as the flat dorsal margin of the ilium, the considerable anteroposterior elongation of the posterior dorsal vertebrae and the distinct posterior rim on the ventral part of the postacetabular process of the ilium exposed in lateral view. Alongside *Pr. triassicus* and possibly *L. liliensterni*, *P. milnerae* represents a third Triassic coelophysoid taxon from Europe and the first unambiguous Triassic theropod from the UK.

The results of our analysis are incongruent with several previous phylogenetic studies of early neotheropods in the placement of *Dr. hanigani* and *Le. praecisio* outside Coelophysoidea and the derived position of '*Syntarsus*' *kayentakatae* within this clade. Previously, *Dr. hanigani* from the Early Jurassic of Wales was recovered as a coelophysoid [58,84], but it was found outside Coelophysoidea by Baron *et al.* [33] in a large polytomy among early Neotheropoda, and as the sister taxon to '*Syntarsus*' *kayentakatae* in Langer *et al.* [38] and Baron *et al.* [39] in analyses that did not recover a monophyletic Coelophysoidea as it is historically considered. Here, *Dr. hanigani* is found as a non-coelophysoid neotheropod in a polytomy with *L. liliensterni* and a clade composed of all remaining averostran-line neotheropods (figure 6). The position of *L. liliensterni* as an early diverging non-coelophysoid neotheropod in our analysis corresponds with several studies [84,95,96,98], although other analyses recovered this species as one of the earliest diverging non-coelophysid coelophysoids

[58,67,100]. *Lepidus praecisio* was previously considered as a coelophysid [58,95,99,100,110], but was found outside Coelophysoidea by Marsh *et al.* [96], in which it was found as the sister taxon to all other non-coelophysoid neotheropods, and by Marsh & Rowe [64], in which it was found as the sister taxon to *L. liliensterni* in a clade that is part of a polytomy at the base of Neotheropoda. Here, the holotype of *Le. praecisio* was recovered outside Neotheropoda as the direct sister taxon to this clade (figure 6). We included only the holotype of *Le. praecisio* in our analysis because of the uncertain taxonomic affinity of the referred material for this species [99]. '*Syntarsus*' *kayentakatae* is unambiguously considered as a coelophysoid theropod and represents one of best-known taxa of the clade [76]. However, the position of '*Syntarsus*' *kayentakatae* within Coelophysoidea is somewhat uncertain and in recent other studies employing the Nesbitt *et al.* [89] matrix, this taxon was recovered as quite distantly related to Coelophysidae (i.e. being more distantly related to *C. bauri* and *Pr. triassicus* than are *Pa. lufengensis*, *Lu. bonoi* and *Po. podocitus*) [58,67,84,95,98,100]. By contrast, in our analysis '*Syntarsus*' *kayentakatae* is recovered as the sister taxon to Coelophysidae (figure 6). The lack of consensus in early neotheropod phylogeny, including Coelophysoidea, has been acknowledged and discussed in recent studies (e.g. [58,100]) and can likely be attributed to large amounts of missing data for many early neotheropod taxa and the inclusion of taxa represented by relatively skeletally immature ontogenetic stages [54,100]. The holotype of *Le. praecisio* includes a particularly large amount of missing data (90.72%) and *Le. praecisio*, *L. lilliensterni* and *Dr. hanigani* are represented by immature specimens (electronic supplementary material, figure S11 and table S2) [57]. The inclusion of taxa based on skeletally immature specimens or specimens of an unclear ontogenetic stage increases the proportion of missing data because ontogenetically variable characters were scored as ambiguous (*sensu* [73]). The more derived position of '*Syntarsus*' *kayentakatae* recovered in our analysis is, in part, a result of the scorings revised in our modified data matrix and the inclusion of the new species *P. milnerae*. Indeed, if the latter species is excluded *a priori* from the analysis, '*Syntarsus*' *kayentakatae* is recovered in multiple positions among non-coelophysid coelophysoids in the resultant MPTs. This result reflects the importance of adding new taxa with a novel combination of character states and the continuous revision of the data matrices in phylogenetic studies.

## 6.2. Body size evolution of *Pendraig milnerae* and other early theropods

Research on early theropod body size evolution has recently been reviewed by Griffin [100] and Griffin & Nesbitt [57]. Recent analyses using ancestral state reconstruction found the femoral length of the last common ancestor of Neotheropoda to be approximately 29–35 cm [100,111,112]. Lee *et al.* [113] found a considerably higher ancestral femoral length of 47.5 cm for Neotheropoda, but the dataset used in that analysis contained a comparatively smaller sample of early theropods. Our analyses reveal that different values used for the minimum branch length parameter (mbl, set at 0.1, 0.5 and 1.0 Myr) have quite large implications for the reconstructed ancestral values. The analysis on the first of the three equally parsimonious trees with mbl set at 1.0 Myr and including all sampled taxa (figure 8) recovered an ancestral femoral length of 24.5 cm for Neotheropoda (upper CI: 33.8 cm; lower CI: 17.8 cm) and 18.7 cm for Coelophysoidea (upper CI: 28.5 cm; lower CI: 12.23 cm) (electronic supplementary material, table S3.1), whereas when an mbl of 0.1 Myr is considered, the ancestral value for Neotheropoda is 39.8 cm (upper CI: 57.4 cm; lower CI: 27.5 cm) and that for Coelophysoidea is 15.0 cm (upper CI: 25.2 cm; lower CI: 8.9 cm) (electronic supplementary material, table S3.3). For an mbl of 0.5 Myr, the ancestral value for Neotheropoda is 29.9 cm (upper CI: 42.9 cm; lower CI: 20.8 cm) and for Coelophysoidea 16.8 cm (upper CI: 27.5 cm; lower CI: 10.3 cm) (electronic supplementary material, table S3.4). When the femoral lengths of small-bodied neotheropod taxa represented by immature specimens are pruned (i.e. taxa with a maximum maturity score of 17 or less: *Pa. lufengensis*, *Le. praecisio*, *Dr. hanigani* and *Po. podocitus*), the ancestral femoral length is 20.16 cm (upper CI: 30.2 cm; lower CI: 13.4 cm) for Neotheropoda and 19.7 cm (upper CI: 31.8 cm; lower CI: 12.2 cm) for Coelophysoidea when considering the first of the three equally parsimonious trees and with mbl set at 1.0 Myr (electronic supplementary material, table S3.2). The three equally parsimonious resolutions of the early coelophysoid relationships result in similar reconstructed femoral lengths for Neotheropoda and Coelophysoidea, with the ancestral estimates for the latter being between 18.5 and 19 cm when mbl is set at 1.0 Myr and the femoral lengths of all sampled taxa are included (electronic supplementary material, tables S3.1, S3.5–S3.6). Overall, the large discrepancy in reconstructed ancestral femoral lengths for Neotheropoda and Coelophysoidea between the different analyses, particularly between the analyses with different minimum branch

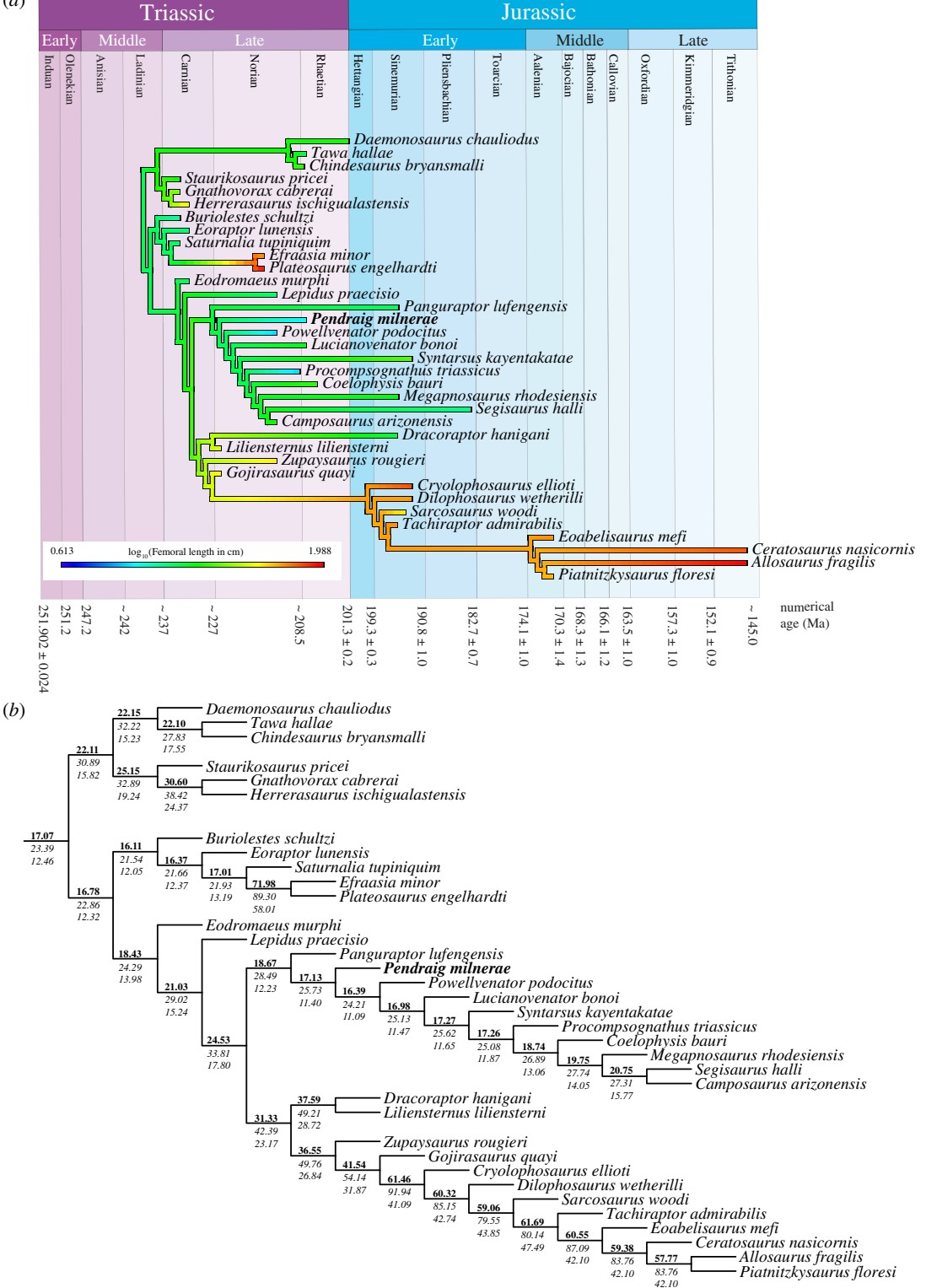

**Figure 8.** Results of the ancestral state reconstruction for Saurischia using the first of the three MPTs used to perform the analyses (the non-saurischian region of the tree was omitted for simplicity). For the analysis figured here, all sampled taxa were included and the minimum branch length was set at 1.0 Myr. (*a*) Time-calibrated heatmap of the log$_{10}$-transformed femoral lengths and (*b*) the same part of the tree with the ancestral estimates for each node plotted above the branch in bold and the upper and lower 95% confidence intervals plotted below it in italics. The complete analysis, as well as the analyses performed under different parameters can be found in the electronic supplementary material.

lengths, indicates that there is much uncertainty in approximating ancestral body sizes in early neotheropods, but our recovered values are broadly similar to the results of Irmis [111], Benson *et al*. [112] and Griffin [100].

Our results indicate that averostran-line neotheropods underwent a size increase during the Triassic (figure 8; electronic supplementary material). By contrast, the body size of coelophysoids is considerably smaller. This corresponds with the results of Griffin [100]. The minor size decreases occurred early in Coelophysoidea and ancestral values gradually increase slightly in consecutive nodes from the clade encompassing *Lu. bonoi*, '*Syntarsus' kayentakatae*, and Coelophysidae onwards regardless of the minimum branch length used in the analyses encompassing all data (electronic supplementary material, tables S3.1, S3.3, S3.4). In the topology in which *Lu. bonoi* is more distantly related to Coelophysidae, ancestral femoral length increases from the clade comprising '*Syntarsus' kayentakatae* and Coelophysidae onwards (electronic supplementary material, table S3.6). By contrast, Griffin [100] found an initial increase in body size in the evolution of Coelophysoidea, which is attributable to the placement of *L. liliensterni* (and, in one of the two analyses, *G. quayi*) at the base of the clade in his phylogenetic analyses and the absence of several coelophysoid taxa in that dataset (*Po. podocitus*, *Pr. triassicus*, *Segisaurus halli*, *Lu. bonoi*, *Camposaurus arizonensis* and the new taxon *P. milnerae*). In the analyses excluding *Pa. lufengensis*, *Le. praecisio*, *Dr. hanigani* and *Po. podocitus*, the ancestral body size of Coelophysoidea is also reduced relative to the ancestral neotheropod condition and decreases somewhat further early in coelophysoid evolution, only to subsequently increase gradually in more apical nodes (electronic supplementary material, table S3.2).

The ontogenetic assessment of NHMUK PV R 37591, the holotype of *P. milnerae*, indicates that this specimen was not skeletally mature (i.e. likely had not reached asymptotic growth [57]) but also that it was probably not at an early ontogenetic stage (electronic supplementary material, figure S11A). Although these cases are rare, some small-sized specimens of *C. bauri* and *M. rhodesiensis* exhibit both a comparable femoral length and a similar range of mature ontogenetic features as NHMUK PV R 37591 [54]. Therefore, while the inferred femoral length of NHMUK PV R 37591 and maturity assessment hint that *P. milnerae* was likely smaller than these taxa, this cannot be stated with certainty. The absence of a hyposphene–hypantrum articulation in the dorsal vertebrae of *P. milnerae* provides an alternative indication of the body size of the taxon. This articulation occurs widely among extinct Archosauria and is correlated with body size, providing biomechanical support to vertebrae in large bodies in the absence of an alternative vertebral bracing mechanism, and it is present in all non-avialan theropods with a femoral length above 170 mm [114]. The clear absence of this feature in *P. milnerae* therefore suggests that its maximum femoral length was beneath this threshold, which implies a considerably smaller body size than better known coelophysoids like *C. bauri*, *M. rhodesiensis* and '*Syntarsus' kayentakatae*.

The ancestral femoral length for the closest node to *P. milnerae* is between 145% (electronic supplementary material, table S3.3) and 190% (electronic supplementary material, table S3.2) larger than the inferred femoral length of *P. milnerae*, thus indicating that the small size of *P. milnerae* is autapomorphic. However, *Pr. triassicus*, *Segisaurus halli* and, in the analyses in which this taxon is considered, *Po. podocitus*, all independently underwent a similar size reduction based on our analyses (figure 8; electronic supplementary material). Because it is not possible to infer the maximum body size of *P. milnerae*, and because its small size is not unique among Coelophysoidea and other coelophysoid taxa that underwent a similar size reduction were not restricted to insular environments, our dataset is ambiguous regarding insular dwarfism as a possible explanation for the reduced body size in *P. milnerae*. However, insular dwarfism in *P. milnerae* also cannot be excluded, and further studies into the palaeohistology and body size evolution of other taxa from Pant-y-ffynnon and related fissure fill deposits are required to investigate the possibility that these faunas were subject to dwarfism or other aspects of the 'Island Rule'.

## 6.3. Palaeoecology of *Pendraig milnerae* and Pant-y-ffynnon

The known dentitions of coelophysoid theropods are characterized by blade-like serrated maxillary and non-mesial dentary teeth, indicating a mostly macrophagous carnivorous diet for these taxa [60,68,76]. It is therefore highly likely that *P. milnerae* had a similar dentition and diet even though no craniodental remains from Pant-y-ffynnon can unequivocally be attributed to this species. *Pendraig milnerae* represents a second macrophagous predator known from Pant-y-ffynnon (figure 9), the other being the non-crocodyliform crocodylomorph *T. gracilis* [18]. Like *P. milnerae*, *T. gracilis* was small-bodied (approx. 76 cm in total body length [18]) and had a gracile body plan. Other likely predators known from the Late Triassic and Early Jurassic fissure fill deposits of southwestern England and southern Wales were either similarly small-bodied: *Terrestrisuchus*-like unidentified crocodylomorphs from Cromhall and Ruthin Quarries [19,32] and '*Agnosphytis cromhallensis*' from Cromhall Quarry [32]; or

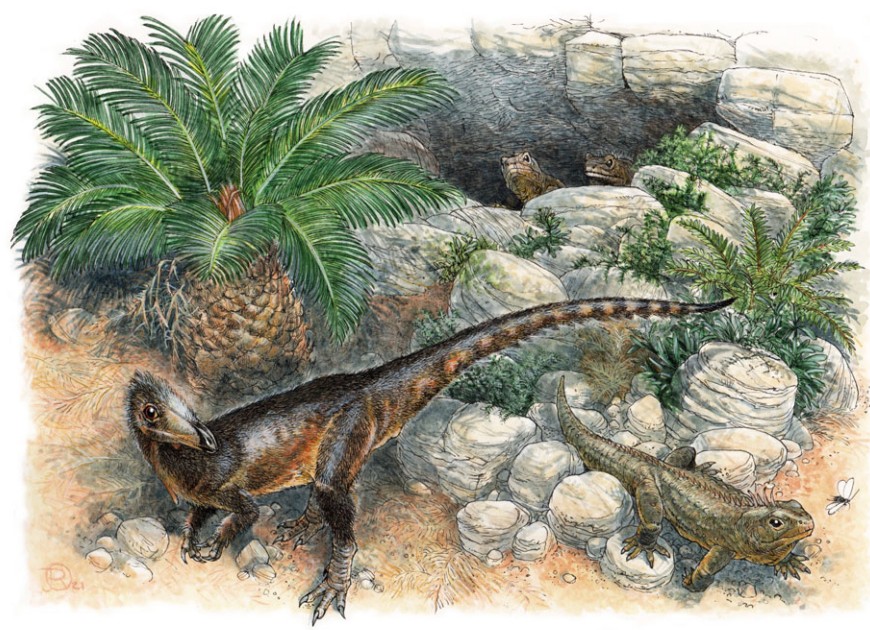

**Figure 9.** Life reconstruction of *P. milnerae* gen. et sp. nov. among the fissures of Pant-y-ffynnon and three individuals of the rhynchocephalian lepidosaur *Clevosaurus cambrica* during the Late Triassic. Artwork by James Robbins.

most likely semi-aquatic (*Paleosaurus platyodon*) [27]. Remains of the considerably larger herbivorous sauropodomorph *Th. antiquus* have been preserved at the Durdham Down and Tytherington deposits [20,27], but only remains of the smaller sauropodomorph *Pa. caducus*, which might represent an immature form of *Th. antiquus* [20], are known from Pant-y-ffynnon [28,29]. Therefore, it is currently unclear whether predators at Pant-y-ffynon simply did not exceed the size of *P. milnerae* and *T. gracilis*, or whether larger-bodied predators in this ecosystem have not yet been discovered or preserved, possibly because taphonomic factors are biased against preservation of large-bodied animals at Pant-y-ffynon and other fissure fill deposits.

Data accessibility. All data are provided as electronic supplementary material and this is indicated in the text. Zip folders are provided for the R code and associated data files for both the regression and ancestral state reconstruction analyses. The initial code for the ancestral state reconstruction is provided for the analysis figured in figure 8. The parameters can easily be changed manually by colleagues to perform the analyses under the various settings as they are presented in the electronic supplementary material. Different MPTs and the femoral lengths used for the conservative analyses are provided in the zip folder and minimum branch length can easily be adjusted from 1.0 to 0.5 and 0.1 in the code.

The data are provided in electronic supplementary material [115].

Authors' contributions. S.N.F.S., M.D.E., R.J.B. and S.C.R.M. designed the study. S.N.F.S. and M.D.E. contributed to data collection. S.N.F.S., M.D.E. and R.J.B. analysed the data. S.N.F.S. made the figures and wrote the majority of the manuscript. All authors contributed to the writing and reviewing of the manuscript.

Competing interests. We declare we have no competing interests.

Funding. S.N.F.S. is funded by a Swiss National Science Foundation Early Postdoc Mobility Fellowship (P2ZHP2_195162). M.D.E. is supported by Agencia Nacional de Promoción Científica y Técnica (PICT 2018–01186) and a Sepkoski Grant 2019 of the Paleontological Society International Research Program.

Acknowledgements. We thank Angela Milner (1947–2021; NHMUK) for relocating the specimen and providing a copy of Warrener's unpublished PhD thesis. We would also like to recognize the support and encouragement provided by Angela over many years to several of the authors as an inspirational colleague, mentor, collaborator and friend, and her substantial contributions to increasing understanding of theropod dinosaurs from the UK, including seminal research on *Baryonyx* and *Proceratosaurus*. We thank Kevin Webb (NHMUK) for providing the high-resolution photographs of the specimens used in the figures, and James Robbins for the life reconstruction used in figure 9. Marc Jones (University College London) and Bruce Griffiths kindly provided advice on the Welsh language in the naming of *Pendraig milnerae*. The following curators, researchers and collection managers are thanked for providing access to specimens under their care for the purpose of this research: Carl Mehling (AMNH), Max Langer (University of São Paulo), Daniela Schwarz (HMN), Jessica Cundiff (MCZ), Eduardo Ruigomez, José Carballido, and Diego Pol (Museo Paleontológico Egidio Feruglio), Sandra Chapman and Lorna Steel (NHMUK), Erich Fitzgerald (NMV), Caroline Buttler (NMW), Sergio Martin, Emilio Vaccari and Gabriela Cisterna (Museo de Ciencias Antropológicas y Naturales at Universidad Nacional de La Rioja), Jaime Powell, Pablo

Ortíz and Rodrigo Gonzalez (PVL), Ricardo Martínez and Diego Abelín (PVSJ), Rainer Schoch (SMNS), Kevin Padian and Pat Holroyd (UCMP) and Michael Brett-Surman and Hans-Dieter Sues (USNM). We are thankful for suggestions by the editor Jennifer Botha and reviewers Chris Griffin and Adam Marsh that helped to improve the manuscript. Access to the free version of TNT 1.5 was possible due to the Willi Henning Society.

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
