## [Peer Review File · Royal Society Open Science]

***Pendraig milnerae*, a new small-sized coelophysoid
theropod from the Late Triassic of Wales**

Stephan N. F. Spiekman, Martín D. Ezcurra, Richard J. Butler, Nicholas C. Fraser and
Susannah C. R. Maidment

Article citation details

R. Soc. open sci. **8**: 210915.
<http://dx.doi.org/10.1098/rsos.210915>

Review timeline

Original submission: 7 June 2021
Revised submission: 25 August 2021
Final acceptance: 7 September 2021

Note: Reports are unedited and appear as
submitted by the referee. The review history
appears in chronological order.

Review History

RSOS-210915.R0 (Original submission)

Review form: Reviewer 1 (Christopher Griffin)

Is the manuscript scientifically sound in its present form?

Yes

Are the interpretations and conclusions justified by the results?

No

Is the language acceptable?

Yes

Do you have any ethical concerns with this paper?

No

Have you any concerns about statistical analyses in this paper?

No

Recommendation?

Major revision is needed (please make suggestions in comments)

Comments to the Author(s)

This manuscript presents a thorough comparative description of a new theropod dinosaur and places it into a phylogenetic hypothesis using a relevant and recent character matrix. The small size of the holotype individual is striking, so the authors also conduct an ancestral state reconstruction for body size among early theropod dinosaurs and also attempt to take the individual's ontogenetic status into account to be sure that the small size does not simply stem from the individual's immaturity.

The manuscript is well-written and thorough, the comparative descriptions sound, the figures are clear and informative, and the analyses all appear to be properly conducted (but see below for problems with the R code). In my opinion the manuscript is largely sound and can be accepted for publication with moderate revision. I have attached a PDF (see Appendix A) with my minor edits and comments.

Major Comments

1) Why are the eleven supplementary figures provided as separate files, with yet another Word file for the captions? It would be much easier to include all figures with associated captions as one PDF file. Also, in Table S2 in the "Juvenile" column, there are several different terms used for maturity assessment, including juvenile, non-juvenile, subadult, immature, etc. Sometimes these seem to be synonymous, but their usage at other times appears to be mutually exclusive. For example, some taxa are listed as "SUBADULT/IMMATURE" and others are just listed as "IMMATURE". Could you provide more explanation for these maturity categories?

2) When running the R code, I experienced an error code that prevented me from continuing to evaluate the rest of the code. The error occurred on line 42, on the `anc.ML()` command, and read: "Error in `optim(c(sig2, a, y, rep(mean(x), length(xx))), fn = likelihood, : non-finite value supplied by optim"`.

All the underlying data looked sound to me; I saw no obvious issues with the way the data was input, the tree file, etc. It may be the issue is with my version of R or R Studio, because I just updated both two days ago, but this affects repeatability and is something the authors should be aware of.

3) I appreciate that the authors take the body size of the type individual into account when evaluating possible small body size. This is done in a clear way using character state transformations that have been useful in other early theropods, particularly *Coelophysis*. I agree with the authors' assessment that this individual does not display the features we might expect of either a very skeletally mature or immature individual, and the character states instead suggest that this individual is in a middling 'gray zone' of ontogeny. Indeed, there is no reasonably complete specimen of *Coelophysis* that has been scored with a consistent combination of character states (data from Griffin & Nesbitt 2016, Griffin 2018), making direct comparison with *Coelophysis* character states and femoral lengths difficult. This, combined with the fact that their scoring of *Panguraptor* ontogenetic character state combinations that contradict any reconstructed ontogenetic sequence of *Coelophysis* (very interesting finding, by the way), strongly suggests that there is even more variation in ontogenetic trajectories among *Coelophysoids* than has presently been reconstructed.

Because of the large amount of ontogenetic variation known from other *coelophysoids*, combined with the fact that most of the size variation in *Coelophysis* is known precisely from this ontogenetic gray zone, I disagree with the authors' statement that this individual was likely near

maximum body size and would not have gotten much larger. Instead, I think that there is not enough evidence to say one way or another.

Although there is no direct comparison between the character states of *Pendraig* and *Coelophysis* or *Megapnosaurus*, some examples from individuals roughly the size of *Pendraig* that also display similarly mature character states may be informative:

--The smallest known individual of *Megapnosaurus* (NHMZ QG 45) possesses fused sacral neural spines (character 1-1; and therefore probably at least some fused sacral centra, although these are not able to be scored) and a small trochanteric shelf (14-1; 15-0), despite its extremely small size (femoral head 1.5 cm; reconstructed femoral length ~11.2 cm).

--*Coelophysis bauri* (TMP 1984.063.0001 #13; reconst. femur length ~10.9 cm) has fused its pubis to the ilium (8-1) and ischium (10-1), but the ilium and ischium remain unfused (9-1).

--*Coelophysis bauri* (SMP 858; femoral head 1.29 cm, reconst. femur length ~10.2 cm) possesses five fused sacrals (2-2), a fused pubis (8-1), a trochanteric shelf (14-1), a mound-like dorsolateral trochanter (16-1), a cranial intermuscular line (17-1), a caudal intermuscular line (18-1), and an 'anterolateral scar' (19-1).

--*Coelophysis bauri* (CMNH 10971 #3; femur length 10.94 cm) possesses five fused sacrals (2-2), a fused pubis (8-1), a trochanteric shelf (14-1), a mound-like dorsolateral trochanter (16-1), a cranial intermuscular line (17-1), a caudal intermuscular line (18-1), and an 'anterolateral scar' (19-1).

There are of course smaller individuals of *Coelophysis* that display more immature character states as well. But my point here is that finding an individual of *Coelophysis* that displays a similar body size and similar ontogenetic character states is not unprecedented. Because we know that *Coelophysis* can reach much larger sizes (~25 cm femoral length), then with a sample size of 1, this is not great evidence that *Pendraig's* maximum size is close to what this individual's is; this taxon could just happen to be represented by one of those more mature-looking but anomalously small individuals also found in other coelophysoids. Therefore, although the data are consistent with *Pendraig* having a small maximum body size, I do not think the data support saying that it did definitively have a small maximum body size.

4) This brings me to my suggestion that the best way to resolve this issue of maturity and body size is by histologically sampling the individual. I understand that the authors probably have qualms about destructively sampling a holotype and only known specimen (barring the referred vertebra) of a taxon, but I do have two ideas that may alleviate some concerns.

I notice in Figure 3 that the femur is already somewhat damaged at midshaft. I suggest that, instead of sampling the entire cross-section of the femoral shaft, you break off a small piece of already-damaged cortex from this midshaft region and histologically sample that (see picture in attached file to see what I mean). I have successfully used this technique when I did not want to damage the full specimen but just sampled a portion that was already damaged, including on a femur of *Dromomeron romeri* (Griffin et al. 2019, PeerJ), and on a femur of *Coelophysis* (CMNH 10971; Barta et al. in prep). Although a full cortical sample is ideal, with a partial cortex you can still see, for example, LAGs, LAG spacing, an EFS if present, etc. I used this method to find an EFS in the *Coelophysis* individual referenced above.

Another option, and slightly more heterodox, would be attempting to sample the distalmost preserved end of the ischium, which is roughly midshaft. Pelvic elements are not often sampled, but in my experience any long, somewhat tubular endochondral bone preserves a record of growth and can be useful for histological maturity assessment when sampled near midshaft. I have seen this work on an immature *Tyrannosaurid* pubis (pers. obs.), metatarsals (McLain et al. 2018, *Palaios*), and hyoid elements (pers. obs., submitted to be a 2021 SVP poster). A 2011 Master's thesis showed that the midshaft of the *Alligator* pubis is skeletochronologically informative, and the ischium did not work only because it is platelike, not elongate, which is not an issue for *Pendraig* (Garcia 2011, "Skeletochronology of the American Alligator (*Alligator Mississippiensis*): Examination of the Utility of Elements for Histological Study", Florida State University). The ischium of *Pendraig* is broken at roughly midshaft, so only ~1 cm of the ischium would need to be removed to make a histological section, causing minimal damage to the specimen. In fact, a

combination of sampling from a femoral fragment and the ischium would be a good multi-elemental way to assess maturity and back up the assessment made by morphology, making the assessment supported by multiple lines of evidence.

I do not hinge my final approval on whether histological sampling is conducted on this specimen, and I think the authors can incorporate the concerns from my comment #3 without including histology. I know that it is not always possible or desirable to destructively sample a holotype. However, I do think that it would make the paper stronger, more convincing, and more citable, and therefore I recommend the authors add this to their study if it is possible.

The authors are free to contact me with any questions, requests for clarification, or concerns.

Chris Griffin
chris.griffin@yale.edu

See attached file (Appendix B) for the modified figures referenced above, showing suggested locations for histological sampling.

Review form: Reviewer 2 (Adam Marsh)

Is the manuscript scientifically sound in its present form?

Yes

Are the interpretations and conclusions justified by the results?

Yes

Is the language acceptable?

Yes

Do you have any ethical concerns with this paper?

No

Have you any concerns about statistical analyses in this paper?

No

Recommendation?

Accept with minor revision (please list in comments)

Comments to the Author(s)

I congratulate the authors on what is a well-researched and well-written manuscript. The description is a good length and provides sufficient comparisons, the assessment and subsequent discussion of skeletal maturity in the new taxon is welcome and justified, and the figures appropriately provide the reader with visual representations of important anatomical characteristics. I conducted the phylogenetic analysis in TNT and have no concerns about that treatment or the subsequent regressions and ancestral state reconstructions (but see below for some comments on interpretation). I have attached my grammar and syntax edits/comments in a marked-up PDF (see Appendix C), but I outline some of the important comments and suggested changes below. Thank you for the opportunity to review this work; it's an important new taxon that says a lot about the diversity of the uppermost western European Triassic and I think will eventually help stabilize this part of the tree.

Page 2, line 47: I would characterize this not necessarily as discrete body size reduction but rather maintaining the plesiomorphic condition of smaller body sizes among dinosauromorphs/pterosauromorphs. See PDF for further comments later in the discussion.

Page 5, line 25: Please add Intuitional Abbreviations section where appropriate.

Page 12, line 31: I personally avoid the word “fused” in these contexts because it can mean different things to different workers in morphological/developmental frameworks. Consider using “coossified”?

Page 14, line 48: Please refer to the ‘Padian theropod’ as PEFO 21373/UCMP 129618 throughout; fossils from federal lands in the US must be first referenced by their federal catalogue number and then by an auxiliary number (if necessary, which in this case I think is since most people know the specimen by its UCMP number).

Page 19, line 46: The holotype of kayentakatae is MNA V2623, is that what you mean here and a few other places?

Page 21, line 29: Did your matrix (or that of Novas et al., 2021 for that matter) include modifications/scorings from Marsh and Rowe, 2020? I’m not asking you to redo analyses, just wanted to clarify. That could be one reason for some different tree topologies between your results and those of Marsh and Rowe, 2020 when it comes to *Lepidus* and/or *kayentakatae*. By the way, at some point we really need to score *Sinosaurus* and *Shuangbaisaurus* into this thing but I think they’ll still fall up-tree from coelophysoids.

Page 21, line 36: I was going to ask about this... I agree it probably doesn't change anything in the part of the tree you're looking at, but at some point those modifications should be incorporated into this matrix for it to be even more useful/up to date.

Page 26, line 49: “Size decreases occurred early in Coelophysoidea and ancestral values gradually increase in consecutive nodes” - I'm not sure you can say this with what is displayed in Fig. 8 (by the way, the next 1.5 pages of text rely heavily on supplemental data). Coelophysoid size was already within the range of most other dinosauriforms, and it only decreases three times (independently?) in *Pendraig*, *Powellvenator*, and *Procompsognathus* with what is shown in this non-supplemental figure. It's surprising to me that *Segisaurus* doesn't show up as markedly smaller in Fig. 8. If there are data that are better displayed in the Supplement that you find yourself citing often, I suggest incorporating them into Fig. 8.

Page 27, line 41: “...our dataset is ambiguous regarding insular dwarfism as a possible explanation of the reduced body size in *Pendraig* orynys. However, insular dwarfism in *Pendraig* orynys cannot be excluded...” - Agreed, but in reality your analyses did not test the hypothesis that insular dwarfism occurred, but rather that body size was changing across the tree in a strictly quantitative way. It can't be excluded because that's not actually what was tested. I think the way you frame your discussion is fine, just be aware of that subtle but important distinction. You didn't test the hypothesis of island dwarfism, you tested the hypothesis that a systematic decrease in body size occurred that may or may not be the result of island dwarfism.

Page 28, line 22: “... which could be attributable to a lack of resources to sustain larger predators as is typical in certain island environments...” - Be very careful here to avoid circular reasoning. You can't use the fact that there are no large-bodied predator fossils as a line of evidence to support island dwarfism, and then also go on to say that island dwarfism caused a lack of large-bodied predators. For what it's worth, I think taphonomic filters are almost certainly at play here.

The fossil record is actually pretty bad and the fact that we can use it to say anything at all is always so incredible to me.

Again, this is overall a very well-written and important contribution and I congratulate the authors on a good piece of work. Please consider my thoughts and let me know if you have any questions or want any clarifications.

Adam Marsh

Decision letter (RSOS-210915.R0)

Dear Dr Spiekman

The Editors assigned to your paper RSOS-210915 "Pendraig orynys, a new small-sized coelophysoid theropod from the Late Triassic of Wales" have now received comments from reviewers and would like you to revise the paper in accordance with the reviewer comments and any comments from the Editors. Please note this decision does not guarantee eventual acceptance.

Please submit your revised manuscript and required files (see below) no later than 21 days from today's (ie 16-Jul-2021) date. Note: the ScholarOne system will 'lock' if submission of the revision is attempted 21 or more days after the deadline. If you do not think you will be able to meet this deadline please contact the editorial office immediately.

Kind regards,
Royal Society Open Science Editorial Office
Royal Society Open Science

on behalf of Dr Jennifer Botha (Associate Editor) and Kevin Padian (Subject Editor)
 openscience@royalsociety.org

Associate Editor Comments to Author (Dr Jennifer Botha):

Associate Editor: 1

Comments to the Author:

This is a good paper, well worth publishing. However, there are some corrections necessary before the paper can be accepted for publication. Two important points worth noting specifically (1) one of the reviewers tested the code and was unable to replicate the results - the code needs to be checked for errors and (2) the interpretation that this individual is a small taxon as opposed to being a juvenile requires further evidence, and reviewer one makes a good case for thin sectioning part of the femur to check for the presence of an EFS. Otherwise there is simply not enough evidence for small size. I encourage the authors to look seriously into this suggestion.

Reviewer comments to Author:

Reviewer: 1

Comments to the Author(s)

This manuscript presents a thorough comparative description of a new theropod dinosaur and places it into a phylogenetic hypothesis using a relevant and recent character matrix. The small size of the holotype individual is striking, so the authors also conduct an ancestral state reconstruction for body size among early theropod dinosaurs and also attempt to take the individual's ontogenetic status into account to be sure that the small size does not simply stem from the individual's immaturity.

The manuscript is well-written and thorough, the comparative descriptions sound, the figures are clear and informative, and the analyses all appear to be properly conducted (but see below for problems with the R code). In my opinion the manuscript is largely sound and can be accepted for publication with moderate revision. I have attached a PDF with my minor edits and comments.

Major Comments

1) Why are the eleven supplementary figures provided as separate files, with yet another Word file for the captions? It would be much easier to include all figures with associated captions as one PDF file. Also, in Table S2 in the "Juvenile" column, there are several different terms used for maturity assessment, including juvenile, non-juvenile, subadult, immature, etc. Sometimes these seem to be synonymous, but their usage at other times appears to be mutually exclusive. For example, some taxa are listed as "SUBADULT/IMMATURE" and others are just listed as "IMMATURE". Could you provide more explanation for these maturity categories?

2) When running the R code, I experienced an error code that prevented me from continuing to evaluate the rest of the code. The error occurred on line 42, on the `anc.ML()` command, and read: "Error in `optim(c(sig2, a, y, rep(mean(x), length(xx))), fn = likelihood, : non-finite value supplied by optim`".

All the underlying data looked sound to me; I saw no obvious issues with the way the data was input, the tree file, etc. It may be the issue is with my version of R or R Studio, because I just updated both two days ago, but this affects repeatability and is something the authors should be aware of.

3) I appreciate that the authors take the body size of the type individual into account when evaluating possible small body size. This is done in a clear way using character state transformations that have been useful in other early theropods, particularly *Coelophysis*. I agree with the authors' assessment that this individual does not display the features we might expect of either a very skeletally mature or immature individual, and the character states instead suggest that this individual is in a middling 'gray zone' of ontogeny. Indeed, there is no reasonably complete specimen of *Coelophysis* that has been scored with a consistent combination of character states (data from Griffin & Nesbitt 2016, Griffin 2018), making direct comparison with *Coelophysis* character states and femoral lengths difficult. This, combined with the fact that their scoring of *Panguraptor* ontogenetic character state combinations that contradict any reconstructed ontogenetic sequence of *Coelophysis* (very interesting finding, by the way), strongly suggests that there is even more variation in ontogenetic trajectories among *Coelophysoids* than has presently been reconstructed.

Because of the large amount of ontogenetic variation known from other *coelophysoids*, combined with the fact that most of the size variation in *Coelophysis* is known precisely from this ontogenetic gray zone, I disagree with the authors' statement that this individual was likely near maximum body size and would not have gotten much larger. Instead, I think that there is not enough evidence to say one way or another.

Although there is no direct comparison between the character states of *Pendraig* and *Coelophysis* or *Megapnosaurus*, some examples from individuals roughly the size of *Pendraig* that also display similarly mature character states may be informative:

--The smallest known individual of *Megapnosaurus* (NHMZ QG 45) possesses fused sacral neural spines (character 1-1; and therefore probably at least some fused sacral centra, although these are not able to be scored) and a small trochanteric shelf (14-1; 15-0), despite its extremely small size (femoral head 1.5 cm; reconstructed femoral length ~11.2 cm).

--*Coelophysis bauri* (TMP 1984.063.0001 #13; reconst. femur length ~10.9 cm) has fused its pubis to the ilium (8-1) and ischium (10-1), but the ilium and ischium remain unfused (9-1).

--*Coelophysis bauri* (SMP 858; femoral head 1.29 cm, reconst. femur length ~10.2 cm) possesses five fused sacrals (2-2), a fused pubis (8-1), a trochanteric shelf (14-1), a mound-like dorsolateral trochanter (16-1), a cranial intermuscular line (17-1), a caudal intermuscular line (18-1), and an 'anterolateral scar' (19-1).

--*Coelophysis bauri* (CMNH 10971 #3; femur length 10.94 cm) possesses five fused sacrals (2-2), a fused pubis (8-1), a trochanteric shelf (14-1), a mound-like dorsolateral trochanter (16-1), a cranial intermuscular line (17-1), a caudal intermuscular line (18-1), and an 'anterolateral scar' (19-1).

There are of course smaller individuals of *Coelophysis* that display more immature character states as well. But my point here is that finding an individual of *Coelophysis* that displays a similar body size and similar ontogenetic character states is not unprecedented. Because we know that *Coelophysis* can reach much larger sizes (~25 cm femoral length), then with a sample size of 1, this is not great evidence that *Pendraig's* maximum size is close to what this individual's is; this taxon could just happen to be represented by one of those more mature-looking but anomalously small individuals also found in other *coelophysoids*. Therefore, although the data are consistent with *Pendraig* having a small maximum body size, I do not think the data support saying that it did definitively have a small maximum body size.

4) This brings me to my suggestion that the best way to resolve this issue of maturity and body size is by histologically sampling the individual. I understand that the authors probably have qualms about destructively sampling a holotype and only known specimen (barring the referred vertebra) of a taxon, but I do have two ideas that may alleviate some concerns.

I notice in Figure 3 that the femur is already somewhat damaged at midshaft. I suggest that, instead of sampling the entire cross-section of the femoral shaft, you break off a small piece of already-damaged cortex from this midshaft region and histologically sample that (see picture in attached file to see what I mean). I have successfully used this technique when I did not want to

damage the full specimen but just sampled a portion that was already damaged, including on a femur of *Dromomeron romeri* (Griffin et al. 2019, PeerJ), and on a femur of *Coelophysis* (CMNH 10971; Barta et al. in prep). Although a full cortical sample is ideal, with a partial cortex you can still see, for example, LAGs, LAG spacing, an EFS if present, etc. I used this method to find an EFS in the *Coelophysis* individual referenced above.

Another option, and slightly more heterodox, would be attempting to sample the distalmost preserved end of the ischium, which is roughly midshaft. Pelvic elements are not often sampled, but in my experience any long, somewhat tubular endochondral bone preserves a record of growth and can be useful for histological maturity assessment when sampled near midshaft. I have seen this work on an immature Tyrannosaurid pubis (pers. obs.), metatarsals (McLain et al. 2018, Palaios), and hyoid elements (pers. obs., submitted to be a 2021 SVP poster). A 2011 Master's thesis showed that the midshaft of the Alligator pubis is skeletochronologically informative, and the ischium did not work only because it is platelike, not elongate, which is not an issue for Pendraig (Garcia 2011, "Skeletochronology of the American Alligator (*Alligator Mississippiensis*): Examination of the Utility of Elements for Histological Study", Florida State University). The ischium of Pendraig is broken at roughly midshaft, so only ~1 cm of the ischium would need to be removed to make a histological section, causing minimal damage to the specimen. In fact, a combination of sampling from a femoral fragment and the ischium would be a good multi-elemental way to assess maturity and back up the assessment made by morphology, making the assessment supported by multiple lines of evidence.

I do not hinge my final approval on whether histological sampling is conducted on this specimen, and I think the authors can incorporate the concerns from my comment #3 without including histology. I know that it is not always possible or desirable to destructively sample a holotype. However, I do think that it would make the paper stronger, more convincing, and more citable, and therefore I recommend the authors add this to their study if it is possible.

The authors are free to contact me with any questions, requests for clarification, or concerns.

Chris Griffin
chris.griffin@yale.edu

See attached file "Spiekman et al_2021_Pendraig orynys_GriffinMajorComments.docx" for the modified figures referenced above, showing suggested locations for histological sampling.

Reviewer: 2

Comments to the Author(s)

I congratulate the authors on what is a well-researched and well-written manuscript. The description is a good length and provides sufficient comparisons, the assessment and subsequent discussion of skeletal maturity in the new taxon is welcome and justified, and the figures appropriately provide the reader with visual representations of important anatomical characteristics. I conducted the phylogenetic analysis in TNT and have no concerns about that treatment or the subsequent regressions and ancestral state reconstructions (but see below for some comments on interpretation). I have attached my grammar and syntax edits/comments in a marked-up PDF, but I outline some of the important comments and suggested changes below. Thank you for the opportunity to review this work; it's an important new taxon that says a lot about the diversity of the uppermost western European Triassic and I think will eventually help stabilize this part of the tree.

Page 2, line 47: I would characterize this not necessarily as discrete body size reduction but rather maintaining the plesiomorphic condition of smaller body sizes among dinosauiromorphs/pterosauiromorphs. See PDF for further comments later in the discussion.

Page 5, line 25: Please add Intuitional Abbreviations section where appropriate.

Page 12, line 31: I personally avoid the word “fused” in these contexts because it can mean different things to different workers in morphological/developmental frameworks. Consider using “coossified”?

Page 14, line 48: Please refer to the ‘Padian theropod’ as PEFO 21373/UCMP 129618 throughout; fossils from federal lands in the US must be first referenced by their federal catalogue number and then by an auxiliary number (if necessary, which in this case I think is since most people know the specimen by its UCMP number).

Page 19, line 46: The holotype of kayentakatae is MNA V2623, is that what you mean here and a few other places?

Page 21, line 29: Did your matrix (or that of Novas et al., 2021 for that matter) include modifications/scorings from Marsh and Rowe, 2020? I’m not asking you to redo analyses, just wanted to clarify. That could be one reason for some different tree topologies between your results and those of Marsh and Rowe, 2020 when it comes to *Lepidus* and/or *kayentakatae*. By the way, at some point we really need to score *Sinosaurus* and *Shuangbaisaurus* into this thing but I think they’ll still fall up-tree from coelophysoids.

Page 21, line 36: I was going to ask about this... I agree it probably doesn't change anything in the part of the tree you're looking at, but at some point those modifications should be incorporated into this matrix for it to be even more useful/up to date.

Page 26, line 49: “Size decreases occurred early in Coelophysoidea and ancestral values gradually increase in consecutive nodes” - I'm not sure you can say this with what is displayed in Fig. 8 (by the way, the next 1.5 pages of text rely heavily on supplemental data). Coelophysoid size was already within the range of most other dinosauriforms, and it only decreases three times (independently?) in *Pendraig*, *Powellvenator*, and *Procompsognathus* with what is shown in this non-supplemental figure. It's surprising to me that *Segisaurus* doesn't show up as markedly smaller in Fig. 8. If there are data that are better displayed in the Supplement that you find yourself citing often, I suggest incorporating them into Fig. 8.

Page 27, line 41: “...our dataset is ambiguous regarding insular dwarfism as a possible explanation of the reduced body size in *Pendraig* ornyns. However, insular dwarfism in *Pendraig* ornyns cannot be excluded...” - Agreed, but in reality your analyses did not test the hypothesis that insular dwarfism occurred, but rather that body size was changing across the tree in a strictly quantitative way. It can't be excluded because that's not actually what was tested. I think the way you frame your discussion is fine, just be aware of that subtle but important distinction. You didn't test the hypothesis of island dwarfism, you tested the hypothesis that a systematic decrease in body size occurred that may or may not be the result of island dwarfism.

Page 28, line 22: “... which could be attributable to a lack of resources to sustain larger predators as is typical in certain island environments...” - Be very careful here to avoid circular reasoning. You can't use the fact that there are no large-bodied predator fossils as a line of evidence to support island dwarfism, and then also go on to say that island dwarfism caused a lack of large-bodied predators. For what it's worth, I think taphonomic filters are almost certainly at play here. The fossil record is actually pretty bad and the fact that we can use it to say anything at all is always so incredible to me.

Again, this is overall a very well-written and important contribution and I congratulate the authors on a good piece of work. Please consider my thoughts and let me know if you have any questions or want any clarifications.

Adam Marsh

===PREPARING YOUR MANUSCRIPT===

===PREPARING YOUR REVISION IN SCHOLARONE===

- 1) One version identifying all the changes that have been made (for instance, in coloured highlight, in bold text, or tracked changes);
 - 2) A 'clean' version of the new manuscript that incorporates the changes made, but does not highlight them.
 - An individual file of each figure (EPS or print-quality PDF preferred [either format should be produced directly from original creation package], or original software format).
 - An editable file of each table (.doc, .docx, .xls, .xlsx, or .csv).
 - An editable file of all figure and table captions.
- Note: you may upload the figure, table, and caption files in a single Zip folder.
- Any electronic supplementary material (ESM).
 - If you are requesting a discretionary waiver for the article processing charge, the waiver form must be included at this step.
 - If you are providing image files for potential cover images, please upload these at this step, and inform the editorial office you have done so. You must hold the copyright to any image provided.
 - A copy of your point-by-point response to referees and Editors. This will expedite the preparation of your proof.

- Ensure that your data access statement meets the requirements at <https://royalsociety.org/journals/authors/author-guidelines/#data>. You should ensure that you cite the dataset in your reference list. If you have deposited data etc in the Dryad repository, please include both the 'For publication' link and 'For review' link at this stage.
- If you are requesting an article processing charge waiver, you must select the relevant waiver option (if requesting a discretionary waiver, the form should have been uploaded at Step 3 'File upload' above).
- If you have uploaded ESM files, please ensure you follow the guidance at <https://royalsociety.org/journals/authors/author-guidelines/#supplementary-material> to include a suitable title and informative caption. An example of appropriate titling and captioning may be found at [https://figshare.com/articles/Table_S2_from_Is_there_a_trade-off_between_peak_performance_and_performance_breadth_across_temperatures_for_aerobic_sc ope_in_teleost_fishes_/3843624](https://figshare.com/articles/Table_S2_from_Is_there_a_trade-off_between_peak_performance_and_performance_breadth_across_temperatures_for_aerobic_scope_in_teleost_fishes_/3843624).

Author's Response to Decision Letter for (RSOS-210915.R0)

See Appendix D.

Decision letter (RSOS-210915.R1)

Dear Dr Spiekman,

It is a pleasure to accept your manuscript entitled "Pendraig milnerae, a new small-sized coelophysoid theropod from the Late Triassic of Wales" in its current form for publication in Royal Society Open Science.

on behalf of Dr Jennifer Botha (Associate Editor) and Kevin Padian (Subject Editor)
openscience@royalsociety.org

Associate Editor Comments to Author (Dr Jennifer Botha):

Associate Editor

Comments to the Author:

The authors have either made all the requested corrections made by the reviewers or offered reasonable explanations for not changing the text, it is now ready to be accepted for publication.

Appendix A**ROYAL SOCIETY
OPEN SCIENCE****Pendraig orynys, a new small-sized coelophysoid theropod
from the Late Triassic of Wales**

Journal:	Royal Society Open Science
Manuscript ID	RSOS-210915
Article Type:	Research
Date Submitted by the Author:	07-Jun-2021
Complete List of Authors:	Spiekman, Stephan; Natural History Museum, Department of Earth Sciences; University of Birmingham, Earth and Environmental Sciences Ezcurra, Martín; CONICET, Sección Paleontología de Vertebrados, Museo Argentino de Ciencias Naturales; University of Birmingham, School of Geography, Earth and Environmental Sciences Butler, Richard; University of Birmingham, School of Geography and Earth Sciences Fraser, Nicholas; National Museums Scotland Maidment, Susannah; Natural History Museum
Subject:	palaeontology < BIOLOGY, evolution < BIOLOGY, taxonomy and systematics < BIOLOGY
Keywords:	Pendraig, Coelophysoidea, Theropoda, Triassic, Body size evolution, Osteology
Subject Category:	Organismal and Evolutionary Biology

Author-supplied statements

Relevant information will appear here if provided.

Ethics

Does your article include research that required ethical approval or permits?:

This article does not present research with ethical considerations

Statement (if applicable):

CUST_IF_YES_ETHICS :No data available.

Data

It is a condition of publication that data, code and materials supporting your paper are made publicly available. Does your paper present new data?:

Yes

Statement (if applicable):

All data are provided for initial review as supplementary material.

Zip folders are provided for the R code and associated data files for both the regression and ancestral state reconstruction analyses. The initial code for the ancestral state reconstruction is provided for the analysis figured in Figure 8. Th parameters can easily be changed manually by reviewers familiar with R to perform the analyses under the various settings provided in the Supplements. Different MPTs and the femoral lengths used for the conservative analyses are provided in the zip folder and minimum branch length can easily be adjusted from 1.0 to 0.5 and 0.1 in the code.

Conflict of interest

I/We declare we have no competing interests

Statement (if applicable):

CUST_STATE_CONFLICT :No data available.

Authors' contributions

This paper has multiple authors and our individual contributions were as below

Statement (if applicable):

SNFS, MDE, RJB, and SCRM designed the study. SNFS and MDE contributed to data collection. SNFS, MDE, and RJB analysed the data. SNFS made the figures and wrote the majority of the manuscript. All authors contributed to the writing and reviewing of the manuscript.

*Pendraig orynys*, a new small-sized coelophysoid theropod from the Late Triassic of Wales

Stephan N. F. Spiekman^{1,2}, Martín D. Ezcurra^{2,3}, Richard J. Butler², Nicholas C. Fraser⁴, Susannah C. R.
Maidment^{1,2}

¹Department of Earth Sciences, Natural History Museum, Cromwell Road, London SW7 5BD, UK

²School of Geography, Earth and Environmental Sciences, University of Birmingham, Edgbaston,
Birmingham B15 2TT, UK

³Sección Paleontología de Vertebrados, CONICET-Museo Argentino de Ciencias Naturales, Ángel
Gallardo 470, C1405DJR, Buenos Aires, Argentina

⁴National Museums Scotland, Chambers St, Edinburgh EH1 1JF, UK

*corresponding author: stephanspiekman@gmail.com

24 **Abstract.**

We describe a new genus and species of small-bodied coelophysoid theropod dinosaur, *Pendraig*

[revised manuscript text omitted]

*lufengensis* ([63], fig. 1: ratio = c. 2.0–2.11 in the last two dorsal vertebrae), and *Megapnosaurus*

*rhodesiensis* ([60]: ratio = 2.10, D13 of NHMB QG 1). By contrast, these vertebrae are considerably
proportionally shorter in *Liliensternus liliensterni* (HMN MB.R.2175 2.22–2.24: ratio = 1.58–1.67,
posterior dorsal vertebrae), *Lucianovenator bonoi* (PVSJ 906: ratio = 1.63, last dorsal vertebra),
*Lophostropheus airelensis* ([64]: ratio = 1.32, last dorsal vertebra), *Dracoraptor hanigani* (NMW
2015.5G.1–2015.5G.11: ratio = 1.63–1.75, middle–posterior dorsal vertebrae), *Cryolophosaurus*
*elliotti* (FMNH PR1821: ratio 1.07, posterior dorsal vertebra), *Dilophosaurus wetherilli* ([61]: ratio =
1.16–1.52, D10, D11 and D13 of UCMP 37302; ratio = 1.19 D14 of UCMP 77270), and *Sarcosaurus*
*woodi* ([57]: ratio = c. 1.9, middle–posterior dorsal vertebra of WARMS G678). The ventral surface of
the centrum is anteroposteriorly concave and lacks a ventral keel (Fig. 1B). The centrum is
amphiplatyan with very slightly concave anterior and posterior articular surfaces. As in the preceding
vertebra, no visible suture is present between the centrum and neural arch. The lateral surface of
the centrum bears an anteroposteriorly elongate but shallow fossa just ventral to the articulation
with the neural arch, which is a common condition in the middle-posterior dorsal vertebrae of early
neotheropods (e.g., *Liliensternus liliensterni*: HMN MB.R.2175; *Procompsognathus triassicus*: SMNS
12591; *Lucianovenator bonoi*: PVSJ 906; *Lophostropheus airelensis* [64]; *Sarcosaurus woodi* [57]). The
last dorsal vertebra possessed only a single articular facet for the rib on each side, located at the end
of a transversely wide, wing-like transverse process (Fig. 2A). In dorsal view, its posterior margin is
concave and its anterior margin appears to be somewhat sinusoidal. There is no distinct fossa on the
dorsal surface of the base of the transverse process. The articular surfaces of the prezygapophyses
face dorsomedially. The articular surface of the left postzygapophysis is poorly preserved. No
hyposphene articular surface is preserved, but this region is poorly preserved. The prezygapophyses
diverge from each other in dorsal view and their tips are well separated from the median line,
contrasting with the sub-parallel prezygapophyses of *Sarcosaurus woodi* [57].

The isolated dorsal vertebra NHMUK PV 37596 is virtually complete, undistorted, and freed from
matrix (Fig. 4). The centrum is 14.6 mm long and 2.59 times longer than the height of its anterior
articular surface (Table 1). The elongation of this centrum matches or closely resembles that of the
middle dorsal vertebrae of the coelophysids *Coelophysis bauri* (e.g., NMV P231382: c. 2.6, middle
dorsal vertebra), *Megapnosaurus rhodesiensis* ([60]: 2.33–2.64, D6 and D7 of NHMB QG 1), and
*Procompsognathus triassicus* (SMNS 12591: 2.70–2.91, D7 and D8), whereas in other early
neotheropods the middle-posterior dorsal vertebrae are proportionally shorter (e.g., *Liliensternus*
*liliensterni*, *Gojirasaurus quayi*; *Dilophosaurus wetherilli*, *Dracoraptor hanigani*, *Cryolophosaurus*
*elliotti*; *Sarcosaurus woodi*) [57, 61, 65–67]. The ventral surface of NHMUK PV 37596 is concave in
lateral view and there is no ventral keel. A single nutrient foramen can be observed close to the
anterior end of the centrum on its right ventrolateral side. The anterior and posterior articular

surfaces of the centrum are both very slightly concave and transversely broader than tall, resembling
the condition in *Gojirasaurus quayi* [65]. By contrast, the centrum is taller than broad in the
posterior dorsal vertebrae of *Megapnosaurus rhodesiensis* ([60]: table 6), *Sarcosaurus woodi* (only
the anterior surface is preserved) [57], *Cryolophosaurus ellioti* [67], and *Dilophosaurus wetherilli*
[61], approximately as broad as tall in *Liliensternus liliensterni* (HMN MB.R.2175), and both
conditions occur in *Eodromaeus murphi* (PVSJ 562). The lateral surfaces of the centrum bear a
shallow fossa directly ventral to the connection to the neural arch, as occurs in NHMUK PV R 37591
and other early neotheropods (see above). The neurocentral suture is closed along most of its
extension, being only visible on the most posterior region of the neural arch peduncle on both sides
of the bone.

The diapophysis is placed on a wide sub-trapezoidal or wing-like transverse process (Fig. 4D). In
dorsal view, the posterior margin of this process is mainly laterally oriented and slightly concave,
whereas the anterior margin is anteromedially to posterolaterally oriented and somewhat
sinusoidal. The anteroposteriorly long base of the transverse process and strong posterolateral
slating of its anterior margin resemble the condition in the middle dorsal vertebrae of *Eodromaeus*
*murphi* (PVSJ 562) and the coelophysids *Coelophysis bauri* (AMNH 7224), *Megapnosaurus*
*rhodesiensis* [60], and *Procompsognathus triassicus* (SMNS 12591). The parapophysis is placed on a
strongly developed, narrow and rod-like stalk, but it is considerably less extended laterally than the
diapophysis, resembling the condition in the middle-posterior dorsal vertebrae of at least some
other early neotheropods (e.g., *Liliensternus liliensterni*: HMN MB.R.2175; *Megapnosaurus*
*rhodesiensis* [60]). Both processes are positioned fully on the neural arch and are connected through
a thin paradiapophyseal lamina (*sensu* [59]). The diapophysis is located slightly dorsal to the
parapophysis (Fig. 4A-B). The articular facet of the diapophysis is oval and anteroposteriorly
elongated, whereas the facet of the parapophysis is subcircular. The paradiapophyseal
centrodiapophyseal fossa ventral to the diapophysis is shallow, whereas the paradiapophyseal
centroprezygapophyseal and postzygapophyseal centrodiapophyseal fossae are very deep and
framed by pronounced and thin laminae (*sensu* [68]). The laminae framing the paradiapophyseal
prezygapophyseal fossa are the prezygaparapophyseal lamina dorsally and the anterior
centroparapophyseal lamina ventrally, whereas the postzygapophyseal centrodiapophyseal fossa is
framed by the postzygodiapophyseal lamina dorsally and the posterior centrodiapophyseal lamina
ventrally (*sensu* [59]). This pattern of laminae and fossae matches that of a posterior dorsal vertebra
of *Liliensternus liliensterni* (HMN MB.R.2175 2.22). There are no pneumatic foramina within the
fossae. The transition between the transverse process of the diapophysis and the neural spine forms
an angle of approximately 90 degrees, and there is no fossa present in this region. The

postzygapophyses are closely placed together and their articulation facets face ventrolaterally. A
hyposphene is absent between the postzygapophyses and therefore there is no accessory
intervertebral articulation (Fig. 4F), contrasting with its presence in the middle-posterior dorsal
vertebrae of *Eodromaeus murphi* (PVSJ 562), *Megapnosaurus rhodesiensis* [60], *Gojirasaurus quayi*
[65], *Cryolophosaurus ellioti* [67], *Sarcosaurus woodi* [57], and *Dilophosaurus wetherilli* [61]. The
articulation facets of the prezygapophyses face dorsomedially (Fig. 4E). Both the
spinoprezygapophyseal and spinopostzygapophyseal fossae are very narrow slit-like openings
between the pre- and postzygapophyses, respectively [68], and they do not extend onto the surface
of the neural spine. The spinopostzygapophyseal fossa is considerably larger than the
spinoprezygapophyseal one.

The neural spine is proportionally low, being 0.4 times taller than anteroposteriorly long at its base,
resembling the condition in the middle and posterior —but not the posteriormost— dorsal
vertebrae of *Coelophysis bauri* (AMNH 7224) and *Megapnosaurus rhodesiensis* [60]. By contrast,
*Eodromaeus murphi* and other early neotheropods (e.g., *Dilophosaurus wetherilli* [61]; *Gojirasaurus*

[revised manuscript text omitted]

*Liliensternus liliensterni*: HMN MB.R.2175; *Lucianovenator bonoi*: PVSJ 906; *Coelophysis* sp.: UCMP
129618; *Notatesseraeraptor frickensis* [73]; *Megapnosaurus rhodesiensis*: NHMB QG 1;
*Dilophosaurus wetherilli* [61]). The dorsal margin of the iliac blade of *Pendraig orynys* possesses a
somewhat thickened, mostly flat surface that faces slightly laterally. This flat surface extends along
most of the bone, with exception of the anteriormost region of the preacetabular process and starts
to taper anteriorly at the mid-length of this process. On the posterior region of the iliac blade, this
flat surface extends ventrally as a raised region to occupy the entire dorsoventral height of the
lateral surface of the posterior end of the postacetabular process. It has been inferred that the
anterior rim of this raised surface probably delimited the attachment site of the *M. iliofemoralis* [72,

74]. This same condition occurs in *Coelophysis bauri* (USNM 529376), *Lucianovenator bonoi* (PVSJ 899, 906), *Coelophysis* sp. (UCMP 129618), '*Syntarsus*' *kayentakatae* [75], and *Megapnosaurus rhodesiensis* (NHMB QG 1), but not in other early neotheropods [72]. The anterior margin of the preacetabular process is continuously rounded and extends considerably further anterior than the pubic peduncle of the ilium, as occurs in other neotheropods [76]. The preacetabular process is transversely very thin (i.e., laminar) and slightly medially curved in dorsal view. The ventral margin of the preacetabular process is slightly convex and oriented somewhat anteroventrally to posterodorsally. However, the overall orientation of the preacetabular process is anteriorly facing and a broad gap separates it from the pubic peduncle. This morphology corresponds to that of most theropods, but contrasts with the anteroventrally directed processes of *Sarcosaurus woodi* and some ceratosaurs (e.g., *Ceratopsaurus nasicornis*, *Eoabelisaurus mefi*) [57]. At its posterodorsal end, the postacetabular process curves abruptly posteroventrally and the posteroventral end of the process is formed by an acute angle of approximately 65 degrees in lateral view. By contrast, the posteroventral corner of the postacetabular process is approximately right-angled or slightly acute in other non-averostran neotheropods (e.g., *Coelophysis bauri*: USNM 529376; *Liliensternus liliensterni*: HMN MB.R.2175; *Lucianovenator bonoi*: PVSJ 906; *Coelophysis* sp.: UCMP 129618; '*Syntarsus*' *kayentakatae*: Tykoski 2005; *Megapnosaurus rhodesiensis*: NHMB QG 1; *Dilophosaurus wetherilli*: Marsh & Rowe 2020; *Sarcosaurus woodi*: NHMUK PV R4840). A notch on the posterior end of the postacetabular process, as has been described for various coelophysoid taxa (e.g., *Coelophysis bauri*, *Coelophysis* sp., *Megapnosaurus rhodesiensis*, '*Syntarsus*' *kayentakatae*) [72], is absent. In dorsal view the ilium is oriented approximately straight anteroposteriorly (Fig. 2A) and the postacetabular process expands gradually laterally towards its posterior end, resembling the condition in *Liliensternus liliensterni* (HMN MB.R.2175), *Lucianovenator bonoi* (PVSJ 906), and *Dilophosaurus wetherilli* (UCMP 37302). By contrast, the the postacetabular process is distinctly more laterally expanded, extending beyond the level of the outer rim of the supra-acetabular crest in dorsal view, in *Coelophysis bauri* (USNM 529376), *Coelophysis* sp. (UCMP 129618), and *Megapnosaurus rhodesiensis* [60].

The lateral surface of the iliac blade is concave along its entire anteroposterior length. A shallow, ~~not well-rimmed~~ fossa is present immediately dorsal to the supra-acetabular crest and this region lacks the vertical ridge present in *Lophostropheus airelensis* [64, 77]. The ventral margin of the postacetabular process is formed by a distinct and sharp brevis shelf (Fig. 1A). The concave portion of the postacetabular process positioned medioventrally to this shelf is the brevis fossa [74]. This fossa is inferred to have formed the attachment site for the *M. caudofemoralis brevis* and is mediodorsally framed by a distinct ridge (Fig. 1B). The brevis fossa is only visible in lateral view in its anterior

portion. The remainder of the fossa faces ventrally or medioventrally and is obscured by the brevis
shelf in lateral view, a condition typical of neotheropods [70].

The acetabulum is fully perforated and mostly formed by the ilium (Fig. 1A). On the posterior surface
of the acetabulum a ~~well posteriorly delimited~~ crescent-shaped rugosity is present, which
represents the antitrochanter. The dorsal portion of the antitrochanter is positioned on the ilium,
whereas most of its surface is present on the ischial portion of the acetabular margin. The
development of this antitrochanter closely resembles those observed in *Megapnosaurus*
*rhodesiensis* (NHMB QG 1), *Coelophysis bauri* (USNM 529376), *Coelophysis* sp. (UCMP 129618),
'Syntarsus' *kayentakatae* [72], and *Lucianovenator bonoi* (PVSJ 906). Dorsally, the acetabulum is
framed by a pronounced supra-acetabular crest, which projects laterally and slightly ventrally (Figs.
1A and 2C). The rim of the crest extends close to the connection with the pubis anteriorly and to the
origin of the brevis shelf posteriorly. However, the supra-acetabular crest and the brevis shelf do not
form the continuous, well-laterally developed ridge present in *Megapnosaurus rhodesiensis* (NHMB
QG 1), *Coelophysis bauri* (USNM 529376), *Lophostropheus airelensis* [77], *Procompsognathus*
*triassicus* (SMNS 12591), 'Syntarsus' *kayentakatae* [72], and *Lucianovenator bonoi* (PVSJ 906). The
condition of *Pendraig orynys* resembles that of *Liliensternus liliensterni* (HMN MB.R.2175),
*Coelophysis* sp. (UCMP 129618), *Dilophosaurus wetherilli* [61], and *Sarcosaurus woodi* [57]. The
pubic peduncle is anteroventrally oriented, whereas the ischiadic peduncle is considerably more
vertically directed, facing only slightly posteroventrally. The suture between the ilium and pubis is
completely unfused. The suture with the ischium is unfused along its posterior portion, but on its
anterior portion, which is located across the antitrochanter and part of the acetabulum, the suture is
closed and the elements are indistinguishably fused.

Pubis.

Both the left and the right pubes of NHMUK PV R 37591 are largely complete and in articulation with
each other (Fig. 2C). Both elements lack the distal end of the pubic shaft, but the shaft extends
further distally in the right element than the left. Overall, the preservation of the left element is
superior to that of the right element since the surface of the latter is damaged in several places.
Therefore, the description of the pubis is largely based on the left element. The shaft of the pubis is
anteroventrally directed and elongate (Fig. 1A). Its extent is considerable but cannot be fully
assessed because the distal end is missing on both sides. The longest preserved pubis, the right
element, is 63.2 mm long. When including the imprint of the pubic shaft, which reaches further
distally but likely does not represent the distal terminus of the pubes, the maximum length is 74.8
52 mm. The shaft is rod-like with a plate-like medial apron, which is lateromedially wide and

anteroposteriorly flat (Fig. 2B-C). The anterior surface of the shaft is convex and, correspondingly,
the posterior surface is concave. As a result, the pubic shaft is slightly anteriorly curved in lateral
view as in *Coelophysis bauri* (AMNH 7223, 7224), *Megapnosaurus rhodesiensis* [60], '*Syntarsus*'
*kayentakatae* [78], *Dracoraptor hanigani* (NMW 2015.5G.1–2015.5G.11), *Procompsognathus*
*triassicus* (SMNS 12591), *Notatesseraeraptor frickensis* (SMF 06-1), and *Gojirasaurus
[revised manuscript text omitted]

*hanigani* are represented by immature specimens (Supplementary Figure 11; Supplementary Table
2) [56]. The presence of taxa based on skeletally immature specimens or with unclear ontogenetic
stage increases the proportion of missing data because ontogenetically variable characters should be
scored as ambiguous. The more derived position of '*Syntarsus*' *kayentakatae* recovered in our
analysis is, in part, a result of the scorings revised in our modified data matrix and the inclusion of
the new species *Pendraig orynys*. Indeed, if the latter species is excluded *a priori* from the analysis,
'*Syntarsus*' *kayentakatae* is recovered in multiple positions among non-coelophysid coelophysoids in
the resultant MPTs. This result reflects the importance of adding new taxa with a novel combination
of character states and the continuous revision of the data matrices in phylogenetic studies.

Body size evolution of *Pendraig orynys* and other early theropods.

Research on early theropod body size evolution has recently been reviewed by Griffin [99] and
Griffin and Nesbitt [56]. Recent analyses using ancestral state reconstruction found the femoral
length of the last common ancestor of Neotheropoda to be approximately 29 to 35 cm [99, 110,
111]. Lee et al. [112] found a considerably higher ancestral femoral length of 47.5 cm for
Neotheropoda, but the dataset used in that analysis contained a comparatively smaller sample of
early theropods. Our analyses reveal that different values used for the minimum branch length

parameter (mbl, set at 0.1, 0.5, and 1.0 million years) have quite large implications for the reconstructed ancestral values. The analysis on the first of the four equally parsimonious trees with mbl set at 1.0 million years and including all sampled taxa (Fig. 8) recovered an ancestral femoral length of 24.2 cm for Neotheropoda (upper CI: 33.7 cm; lower CI: 17.3 cm) and 17.6 cm for Coelophysoidea (upper CI: 27.6 cm; lower CI: 11.2 cm) (Supplementary Table 3.1), whereas when an mbl of 0.1 million years is considered, the ancestral value for Neotheropoda is 39.6 cm (upper CI: 58.0 cm; lower CI: 27.1 cm) and that for Coelophysoidea is 14.9 cm (upper CI: 25.1 cm; lower CI: 8.8 cm) (Supplementary Table 3.3). For an mbl of 0.5 million years, the ancestral value for Neotheropoda is 29.7 cm (upper CI: 43.0 cm; lower CI: 20.5 cm) and for Coelophysoidea 16.4 cm (upper CI: 27.0 cm; lower CI: 9.9 cm) (Supplementary Table 3.4). When the femoral lengths of small-bodied neotheropod taxa represented by immature specimens are pruned (i.e., taxa with a maximum maturity score of 17 or less: *Panguraptor lufengensis*, *Lepidus praecisio*, *Dracoraptor hanigani*, and *Powellvenator podocitus*), the ancestral femoral length is 25.7 cm (upper CI: 36.2 cm; lower CI: 18.3 cm) for Neotheropoda and 21.1 cm (upper CI: 34.5 cm; lower CI: 12.9 cm) for Coelophysoidea when considering the first of the four equally parsimonious trees and with mbl set at 1.0 million years (Supplementary Table 3.2). The four equally parsimonious resolutions of the early coelophysoid relationships result in similar reconstructed femoral lengths for Neotheropoda and Coelophysoidea, with the ancestral estimates for the latter being between 17 and 18 cm when mbl is set at 1.0 million years and the femoral lengths of all sampled taxa are included (Supplementary Tables 3.1, 3.5-3.7). Overall, the large discrepancy in reconstructed ancestral femoral lengths for Neotheropoda and Coelophysoidea between the different analyses, particularly between the analyses with different minimum branch lengths, and the wide confidence intervals ranges for all values indicate that there is much uncertainty in approximating ancestral body sizes in early neotheropods, but our recovered values are broadly similar to the results of Irmis [110], Benson et al. [111], and Griffin [99].

Our results indicate that averostran-line neotheropods underwent a size increase already during the Triassic (Fig. 8; Supplementary Information). In contrast, the body size of coelophysoids is considerably smaller. This corresponds with the results of Griffin [99]. Size decreases occurred early in Coelophysoidea and ancestral values gradually increase in consecutive nodes from the clade encompassing *Lucianovenator bonoi*, '*Syntarsus*' *kayentakatae*, and Coelophysidae onwards regardless of minimum branch length in the analyses encompassing all data (Supplementary Tables 3.1, 3.3, 3.4). In contrast, Griffin [99] found an initial increase in body size in the evolution of Coelophysoidea, which is attributable to the placement of *Liliensternus liliensterni* (and in one of the two analyses *Gojirasaurus quayi*) at the base of the clade in his phylogenetic analyses and the

absence of several coelophysoid taxa in that dataset (*Powellvenator podocitus*, *Procompsognathus*
*triassicus*, *Segisaurus halli*, *Lucianovenator bonoi*, *Camposaurus arizonensis*, and the new taxon
*Pendraig orynys*). In the two equally parsimonious topologies in which *Lucianovenator bonoi* is more
distantly related to Coelophysidae, ancestral femoral length increases from the clade comprising
‘*Syntarsus*’ *kayentakatae*, and Coelophysidae onwards (Supplementary Tables 3.6, 3.7). In the
analyses excluding *Panguraptor lufengensis*, *Lepidus praecisio*, *Dracoraptor hanigani*, and
*Powellvenator podocitus*, the ancestral body size of Coelophysoidea is also reduced relative to the
ancestral neotheropod condition and decreases somewhat further early in coelophysoid evolution
and subsequently remains similar for subsequent nodes, only to increase again at the most apical
node (Supplementary Table 3.2).

The ontogenetic assessment of the holotype of *Pendraig orynys* indicates that this specimen was not
skeletally mature (i.e., likely had not reached asymptotic growth [56]) but that it is likely also not at
an early ontogenetic stage (Supplementary Figure 11A). It therefore seems unlikely that *Pendraig*
*orynys* would have increased much more in size and this species would thus have been considerably
smaller than better known coelophysoids like *Coelophysis bauri*, *Megapnosaurus rhodesiensis*, and
‘*Syntarsus*’ *kayentakatae*. The ancestral femoral length for the closest node for *Pendraig orynys* is
between 144% (Supplementary Table 3.3) and 195% (Supplementary Table 3.2) larger than for
*Pendraig orynys*, thus indicating that the small size of *Pendraig orynys* is autapomorphic. However,
*Procompsognathus triassicus*, *Segisaurus halli*, and, in the analyses in which this taxon is considered,
*Powellvenator podocitus*, all independently underwent a similar size reduction based on our analyses
(Fig. 8; Supplementary Information). Because its small size is not unique among Coelophysoidea and
other coelophysoid taxa that underwent a similar size reduction were not restricted to insular
environments, our dataset is ambiguous regarding insular dwarfism as a possible explanation of the
reduced body size in *Pendraig orynys*. However, insular dwarfism in *Pendraig orynys* cannot be
excluded, and further studies into the palaeohistology and body size evolution of other taxa from
Pant-y-ffynnon and related fissure fill deposits are required to investigate the possibility that these
faunas were subject to dwarfism or other aspects of the ‘Island Rule’.

Palaeoecology of *Pendraig orynys* and Pant-y-ffynnon.

The known dentitions of coelophysoid theropods are characterized by blade-like serrated maxillary
and non-mesial dentary teeth, indicating a mostly macrophagous carnivorous diet for these taxa [60,
69, 75]. It therefore seems highly likely that *Pendraig orynys* had a similar dentition and diet even
though no craniodental remains from Pant-y-ffynnon can unequivocally be attributed to this species.
*Pendraig orynys* represents a second macrophagous predator known from Pant-y-ffynnon (Fig. 9),

the other being the non-crocodyliform crocodylomorph *Terrestrisuchus gracilis* [18]. Like *Pendraig*
*orynys*, *Terrestrisuchus gracilis* was small-bodied (approximately 76 cm in total body length [18]) and
had a gracile body plan. Other likely predators known from the Late Triassic and Early Jurassic fissure
fill deposits of southwestern England and southern Wales were either similarly small-bodied:
*Terrestrisuchus*-like unidentified crocodylomorphs from Cromhall and Ruthin Quarries [19, 32] and
‘*Agnosphytis cromhallensis*’ from Cromhall Quarry [32]; or most likely semi-aquatic (‘*Paleosaurus*
*platyodon*’) [27]. Remains of the considerably larger-sized herbivorous sauropodomorph
*Thecodontosaurus antiquus* have been preserved at the Durdham Down and Tytherington deposits
[20, 27], but only remains of the smaller sauropodomorph *Pantyraco caducus*, which might
represent an immature form of *Thecodontosaurus antiquus* [20], are known from Pant-y-ffynnon
[28, 29]. Therefore, it is currently unclear whether predators at Pant-y-ffynnon simply did not exceed
the size of *Pendraig orynys* and *Terrestrisuchus gracilis*, which could be attributable to a lack of
resources to sustain larger predators as is typical in certain island environments [22], or whether
larger-bodied predators in this ecosystem have not yet been discovered or preserved, possibly
because taphonomic factors are biased against preservation of large-bodied animals at Pant-y-
ffynnon and other fissure fill deposits.

**Acknowledgements.**

We thank Angela Milner (NHMUK) for relocating the specimen and providing a copy of Warrener’s
unpublished PhD thesis, Kevin Webb (NHMUK) for providing the high-resolution photographs of the
specimens used in the figures, and James Robbins for the life reconstruction used in Figure 9. Marc
Jones (UCL) and Bruce Griffiths kindly provided advice on the Welsh language in the naming of
*Pendraig orynys*. The following curators, researchers and collection managers are thanked for
providing access to specimens under their care for the purpose of this research: Carl Mehling
(AMNH), Max Langer (LPRP/USP), Daniela Schwarz (HMN), Jessica Cundiff (MCZ), Eduardo Ruigomez,
José Carballido, and Diego Pol (MPEF), Sandra Chapman and Lorna Steel (NHMUK), Erich Fitzgerald
(NMV), Caroline Buttler (NMW), Sergio Martin, Emilio Vaccari and Gabriela Cisterna (PULR), Jaime
Powell, Pablo Ortíz and Rodrigo Gonzalez (PVL), Ricardo Martínez and Diego Abelín (PVSJ), Rainer
Schoch (SMNS), Kevin Padian and Pat Holroyd (UCMP) and Michael Brett-Surman and Hans-Dieter
Sues (USNM). Access to the free version of TNT 1.1 was possible due to the Willi Henning Society.
SNFS is funded by a Swiss National Science Foundation Early Postdoc Mobility Fellowship
(P2ZHP2_195162). MDE is supported by Agencia Nacional de Promoción Científica y Técnica (PICT
2018–01186) and a Sepkoski Grant 2019 of the Paleontological Society International Research
Program.

Figure captions.

Figure 1. Holotype NHMUK PV R 37591 pelvis and vertebrae of *Pendraig orynys* gen. et sp. nov. in (A) left lateral view, (B) right lateral view. Abbreviations: atr, antitrochanter; bf, brevis fossa; bfr, brevis fossa rim; bs, brevis shelf; dv, dorsal vertebra; iss, ischial shaft; nc, neural canal; no, notch; obf, obturator foramen; poap, postacetabular process; prap, preacetabular process; puf, pubic fenestra; pus, pubic shaft; ras, rib attachment scar; ri, rim; sac, supra-acetabular crest; sv, sacral vertebra.

Figure 2. Holotype NHMUK PV R 37591 pelvis and vertebrae of *Pendraig orynys* gen. et sp. nov. in (A) dorsal view, (B) ventral view, (C) anterior view, and (D) posterior view. Abbreviations: bf, brevis fossa; bfr, brevis fossa rim; diap, diapophysis; dv, dorsal vertebra; gr, groove; il, ilium; ipis, iliac peduncle of ischium; iss, ischiadic shaft; obf, obturator foramen; poap, postacetabular process; ppdl, paradiapophyseal lamina; prap, preacetabular process; puf, pubic fenestra; pus, pubic shaft; sac, supra-acetabular crest; sv, sacral vertebra; tp, transverse process; vl, ventral lamina.

Figure 3. Holotype NHMUK PV R 37591 left femur of *Pendraig orynys* gen. et sp. nov. in (A) posteromedial, (B) anterolateral, (C) anteromedial, (D) posterolateral, (E) proximal, and (F) distal view. Abbreviations: amt, anteromedial tuber; at, anterior trochanter; icfl, depression associated with the insertion of the *M. caudofemoralis longus*; gt, greater trochanter; lica, linea intermuscularis caudalis; lincr, linea intermuscularis cranialis; obr, obturator ridge; pmt, posteromedial tuber; ts, trochanteric shelf; 4th t, fourth trochanter.

Figure 4. Isolated mid to posterior dorsal vertebra NHMUK PV 37596 of *Pendraig orynys* gen. et sp. nov. in (A) right lateral view, (B) left lateral view, (C) ventral view, (D) dorsal view, (E) anterior view, and (F) posterior view. Abbreviations: aas, anterior articular surface; acpl, anterior centroparapophyseal lamina; ce, centrum; diap, diapophysis; nf, nutrient foramen; ns, neural spine; pacdf, parapophyseal centrodiaepophyseal fossa; pacprf, parapophyseal centroprezygapophyseal fossa; pap, parapophysis; pas, posterior articular surface; pcdl, posterior centrodiaepophyseal; pocdf, postzygapophyseal centrodiaepophyseal fossa; podl, postzygodiaepophyseal lamina; poz, postzygapophysis; ppdl, paradiapophyseal lamina; prpl, prezygaparapophyseal lamina; prz, prezygapophysis; spozf, spinopostzygapophyseal fossa; sprzf, spinoprezygapophyseal fossa.

Figure 5. Isolated partial left ischium NHMUK PV R 37597 of *Pendraig orynys* gen. et sp. nov. in (A) medial and (B) dorsal view. Abbreviations: asil, articulation surface with ilium; atr, antitrochanter; ipis, iliac peduncle of ischium.

Figure 6. Strict consensus of six most parsimonious trees of the phylogenetic analysis. Bremer support, absolute bootstrap frequency, and GC bootstrap frequency values are indicated at each branch in that order.

Figure 7. \log_{10} -transformed bivariate plot of the longitudinal width of proximal head of the femur versus the femoral length of early theropods. The solid black line represents the linear regression described by the formula, and the red dotted lines represent the 95% confidence intervals.

Figure 8. Results of the ancestral state reconstruction of the \log_{10} -transformed femoral lengths. For the analysis figured here all sampled taxa were included and minimum branch length was set at 1.0 million years. The polytomy within Coelophysoidea in the strict consensus tree was manually resolved with one of the four equally parsimonious resolutions. The other analyses can be found in the Supplementary Information. The femoral lengths used for the analysis can be found in Supplementary Table 2. Reconstructed values for major nodes can be found in Table 3, and for all nodes in Supplementary Table 3.1. Node numbers used for Supplementary Table 3.1 are indicated for each node in Supplementary Figure 1. 
Figure 9. Life reconstruction of *Pendraig orynys* gen. et sp. nov. amongst the fissures of Pant-y-fynnon and three individuals of the rhynchocephalian lepidosaur *Clevosaurus cambrica* during the Late Triassic. Artwork by James Robbins.

Table 1. Vertebral measurements of NHMUK PV R 37591 and NHMUK PV 37596. Measurements were taken with a Sealey electronic vernier calliper. Values preceded by a tilde (~)  indicate an approximated value because a measurement was hampered, either by poor preservation, or by the relevant structure being partially covered. Abbreviations: dv, dorsal vertebra; sv, sacral vertebra.

							NHMUK PV
	dv1	dv2	sv1	sv2	sv3	sv4	37596
Centrum length	-	14.7 mm	-	13.3 mm	11.1 mm	-	14.6 mm
Neural spine height	-	-	-	-	-	-	5.0 mm
Neural spine length	-	-	-	-	-	-	14.3 mm
Width of diapophysis/transverse processes (+ rib)	-	9.9 mm	~5.39 mm	6.2 mm	6.5 mm	-	7.4 mm
Anterior articular surface centrum height	-	6.4 mm	-	5.2 mm	4.7 mm	-	5.7 mm

Posterior articular surface centrum height	□7.0 mm	6.4 mm	3.94 mm	4.7 mm	□4.2 mm	-	5.8 mm
Anteroposterior width transverse proces + rib distal end	n/a	n/a	n/a	3.7 mm	5.7 mm	n/a	n/a

Table 2. Measurements of the appendicular skeleton of NHMUK PV R 37591. Measurements were taken with a Sealey electronic vernier calliper. The circumference of the femoral shaft was measured by running a piece of string around the shaft and subsequently measuring the length of the amount of string with the calliper. Values in parentheses represent incomplete values, due to the relevant structure being incompletely preserved. Abbreviations: max, maximum

max. length left ilium across iliac blade	55.8 mm
max. length left ilium across peduncles	26.0 mm
max. length left acetabulum	16.7 mm
max. dorsoventral height left acetabulum	17.6 mm
max. length right pubis (excluding imprint)	(63.2 mm)
max. length right pubis (including imprint)	(74.8 mm)
max. length left ischium	(26.6 mm)
max. length left femur	(86.3 mm)
max. width proximal head left femur	15.1 mm
min. circumference shaft of left femur	25.08 mm

Table 3. Ancestral state values (in cm) for major nodes for the analysis shown in Figure 8.

	Ancestral estimate	Variance	Lower 95% Confidence Interval	Upper 95% Confidence Interval
Dinosauria	15.038	1.013	10.787	20.964
Theropoda	17.679	1.008	13.542	23.080
Neotheropoda	24.168	1.013	17.331	33.703
Non-coelophysoid neotheropods	30.899	1.011	22.653	42.146
Averostra	60.474	1.011	44.309	82.536
Coelophysoidea	17.585	1.023	11.191	27.631

Coelophysoidea excluding				
Panguraptor	16.285	1.021	10.658	24.882
Coelophysidae	17.118	1.015	11.891	24.643

References.

- Whiteside D.I., Duffin C.J., Gill P.G., Marshall J.E.A., Benton M.J. 2016 The Late Triassic and Early Jurassic fissure faunas from Bristol and South Wales: stratigraphy and setting. *Palaeontologia Polonica* **67**, 257-287. (doi:10.4202/pp.2016.67_257).
- Robinson P.L. 1957 The Mesozoic fissures of the Bristol Channel area and their vertebrate faunas. *Zoological Journal of the Linnean Society* **43**(291), 260-282.
- Harris T.M. 1957 A Liasso-Rhaetic flora in south Wales. *Proceedings of the Royal Society B: Biological Sciences* **147**(928), 289-308. (doi:10.1098/rspb.1957.0051).
- Lewarne G.C., Pallot J.M. 1957 Mesozoic plants from fissures in the Carboniferous Limestone of South Wales. *Annals and Magazine of Natural History* **10**(109), 72-79. (doi:10.1080/00222935708655930).
- Marshall J.E., Whiteside D.I. 1980 Marine influence in the Triassic 'uplands'. *Nature* **287**(5783), 627-628. (doi:10.1038/287627a0).
- Whiteside D.I., Marshall J.E. 2008 The age, fauna and palaeoenvironment of the Late Triassic fissure deposits of Tytherington, South Gloucestershire, UK. *Geological Magazine* **145**(1), 105-147. (doi:10.1017/S0016756807003925).
- Whiteside D.I., Robinson D. 1983 A glauconitic clay-mineral from a speleological deposit of Late Triassic age. *Palaeogeography, Palaeoclimatology, Palaeoecology* **41**(1-2), 81-85. (doi:10.1016/0031-0182(83)90077-9).
- Mussini G., Whiteside D.I., Hildebrandt C., Benton M.J. 2020 Anatomy of a Late Triassic Bristol fissure: Tytherington fissure 2. *Proceedings of the Geologists' Association* **131**(1), 73-93. (doi:10.1016/j.pgeola.2019.12.001).
- Whiteside D.I., Benton M.J. 2021 Reply to Walkden, Fraser and Simms (2021): The age and formation mechanisms of Late Triassic fissure deposits, Gloucestershire, England: Comments on Mussini, G., Whiteside, D.I., Hildebrandt C. and Benton M.J. *Proceedings of the Geologists' Association* **132**, 138-141. (doi:10.1016/j.pgeola.2020.12.001).
- Morton J.D., Whiteside D.I., Hethke M., Benton M.J. 2017 Biostratigraphy and geometric morphometrics of conchostracans (Crustacea, Branchiopoda) from the Late Triassic fissure deposits of Cromhall Quarry, UK. *Palaeontology* **60**(3), 349-374. (doi:10.1111/pala.12288).
- Walkden G., Fraser N. 1993 Late Triassic fissure sediments and vertebrate faunas: Environmental change and faunal succession at Cromhall, South West Britain. *Modern Geology* **18**, 511-535.
- Walkden G.M., Fraser N.C., Simms M.J. 2021 The age and formation mechanisms of Late Triassic fissure deposits, Gloucestershire, England: Comments on Mussini, G. et al. (2020). Anatomy of a Late Triassic Bristol fissure: Tytherington fissure 2. *Proceedings of the Geologists' Association* **132**(1), 127-137. (doi:10.1016/j.pgeola.2020.10.006).
- Evans S.E. 1980 The skull of a new eosuchian reptile from the Lower Jurassic of South Wales. *Zoological journal of the Linnean Society* **70**(3), 203-264. (doi:10.1111/j.1096-3642.1980.tb00852.x).
- Fraser N. 1982 A new rhynchocephalian from the British Upper Trias. *Palaeontology* **25**(4), 709-725.
- Gill P.G., Purnell M.A., Crumpton N., Brown K.R., Gostling N.J., Stampanoni M., Rayfield E.J. 2014 Dietary specializations and diversity in feeding ecology of the earliest stem mammals. *Nature* **512**(7514), 303. (doi:10.1038/nature13622).

16. Whiteside D.I., Duffin C.J. 2017 Late Triassic terrestrial microvertebrates from Charles Moore's 'Microlestes' quarry, Holwell, Somerset, UK. *Zoological Journal of the Linnean Society* **179**(3), 677-705. (doi:10.1111/zoj.12458).
17. Robinson P.L. 1962 Gliding lizards from the Upper Keuper of Great Britain. *Proceedings of the Geological Society of London* **160**1, 137-146.
18. Crush P.J. 1984 A late Upper Triassic sphenosuchid crocodylian from Wales. *Palaeontology* **27**(1), 131-157.
19. Skinner M., Whiteside D.I., Benton M.J. 2020 Late Triassic island dwarfs? Terrestrial tetrapods of the Ruthin fissure (South Wales, UK) including a new genus of procolophonid. *Proceedings of the Geologists' Association* **131**(5), 535-561. (doi:10.1016/j.pgeola.2020.04.005).
20. Ballell A., Rayfield E.J., Benton M.J. 2020 Osteological redescription of the Late Triassic sauropodomorph dinosaur *Thecodontosaurus antiquus* based on new material from Tytherington, southwestern England. *Journal of Vertebrate Paleontology* **40**(2), e1770774. (doi:10.1080/02724634.2020.1770774).
21. Lovegrove J., Newell A.J., Whiteside D.I., Benton M.J. 2021 Testing the relationship between marine transgression and evolving island palaeogeography using 3D GIS: an example from the Late Triassic of SW England. *Journal of the Geological Society*. (doi:10.1144/jgs2020-158).
22. Lomolino M.V., Brown J.H., Sax D.F. 2010 Island biogeography theory: reticulations and reintegration of "a biogeography of the species". In *The Theory of Island Biogeography Revisited* (eds. Losos J.B., Ricklefs R.E.), pp. 13-51. Princeton, USA, Princeton University Press.
23. Van Valen L. 1973 Pattern and the balance of nature. *Evolutionary Theory* **1**(1), 31-49.
24. Patrick E.L., Whiteside D.I., Benton M.J. 2019 A new crurotarsan archosaur from the Late Triassic of South Wales. *Journal of Vertebrate Paleontology* **39**(3), e1645147. (doi:10.1080/02724634.2019.1645147).
25. Benton M.J. 2012 Naming the Bristol dinosaur, *Thecodontosaurus*: politics and science in the 1830s. *Proceedings of the Geologists' Association* **123**(5), 766-778. (doi:10.1016/j.pgeola.2012.07.012).
26. Riley H., Stutchbury S. 1836 On an additional species of the newly-discovered saurian animals in the Magnesian Conglomerate of Durdham Down, near Bristol. *Annual Report of the British Association for the Advancement of Science, Transactions of the Sections*, 90-94.
27. Benton M.J., Juul L., Storrs G.W., Galton P.M. 2000 Anatomy and systematics of the prosauropod dinosaur *Thecodontosaurus antiquus* from the Upper Triassic of southwest England. *Journal of Vertebrate Paleontology* **20**(1), 77-108. (doi:10.1671/0272-4634(2000)020[0077:AASOTP]2.0.CO;2).
28. Galton P., Kermack D. 2010 The anatomy of *Pantydraco caducus*, a very basal sauropodomorph dinosaur from the Rhaetian (Upper Triassic) of South Wales, UK. *Revue de Paléobiologie* **29**(2), 341-404.
29. Yates A.M. 2003 A new species of the primitive dinosaur *Thecodontosaurus* (Saurischia: Sauropodomorpha) and its implications for the systematics of early dinosaurs. *Journal of Systematic Palaeontology* **1**(1), 1-42. (doi:10.1017/S1477201903001007).
30. Kermack D. 1984 New prosauropod material from South Wales. *Zoological Journal of the Linnean Society* **82**(1-2), 101-117. (doi:10.1111/j.1096-3642.1984.tb00538.x).
31. Galton P.M., Yates A.M., Kermack D. 2007 *Pantydraco* n. gen. for *Thecodontosaurus caducus* Yates, 2003, a basal sauropodomorph dinosaur from the Upper Triassic or Lower Jurassic of South Wales, UK. *Neues Jahrbuch für Geologie und Paläontologie-Abhandlungen* **243**(1), 119-125. (doi:10.1127/0077-7749/2007/0243-0119).
32. Fraser N.C., Padian K., Walkden G.M., Davis A. 2002 Basal dinosauriform remains from Britain and the diagnosis of the Dinosauria. *Palaeontology* **45**(1), 79-95. (doi:10.1111/1475-4983.00228).
33. Baron M.G., Norman D.B., Barrett P.M. 2017 A new hypothesis of dinosaur relationships and early dinosaur evolution. *Nature* **543**(7646), 501-506. (doi:10.1038/nature21700).

34. Bonaparte J., Brea G., Schultz C., Martinelli A. 2007 A new specimen of *Guaibasaurus*
*candelariensis* (basal Saurischia) from the Late Triassic Caturrita Formation of southern Brazil.
*Historical Biology* **19**(1), 73-82. (doi:10.1080/08912960600866862).
35. Ezcurra M.D. 2010 A new early dinosaur (Saurischia: Sauropodomorpha) from the Late
Triassic of Argentina: a reassessment of dinosaur origin and phylogeny. *Journal of Systematic*
*Palaeontology* **8**(3), 371-425. (doi:10.1080/14772019.2010.484650).
36. Langer M.C. 2004 Basal Saurischia. In *The Dinosauria* (eds. Weishampel D.B., Dodson P.,
Osmólska H.), pp. 25-46. Berkeley, University of California Press.
37. Langer M.C., Nesbitt S.J., Bittencourt J.S., Irmis R.B. 2013 Non-dinosaurian
Dinosauromorpha. *Geological Society, London, Special Publications* **379**(1), 157-186.
(doi:10.1144/SP379.9).
38. Langer M.C., Ezcurra M.D., Rauhut O.W., Benton M.J., Knoll F., McPhee B.W., Novas F.E., Pol
D., Brusatte S.L. 2017 Untangling the dinosaur family tree. *Nature* **551**(7678), E1-E3.
(doi:10.1038/nature24011).
39. Baron M.G., Norman D.B., Barrett P.M. 2017 Baron et al. reply. *Nature* **551**(7678), E4-E5.
(doi:10.1038/nature24012).
40. Lucas S.G., Heckert A.B., Fraser N.C., Huber P. 1999 *Aetosaurus* from the Upper Triassic of
Great Britain and its biochronological significance. *Neues Jahrbuch für Geologie und Paläontologie-*
*Monatshefte* **9**, 568-576. (doi:10.1127/njgpm/1999/1999/568).
41. Warrener D. 1983 An archosaurian fauna from a Welsh locality, University College London
(University of London).
42. Rauhut O.W., Hungerbühler A. 2000 A review of European Triassic theropods. *Gaia* **15**, 75-
88.
43. Keeble E., Whiteside D.I., Benton M.J. 2018 The terrestrial fauna of the Late Triassic Pant-y-
ffynnon Quarry fissures, South Wales, UK and a new species of *Clevosaurus* (Lepidosauria:
Rhynchocephalia). *Proceedings of the Geologists' Association* **129**(2), 99-119.
(doi:10.1016/j.pgeola.2017.11.001).
44. Cope E. 1869-1870 Synopsis of the extinct Batrachia, Reptilia and Aves of North America.
*Transactions of the American Philosophical Society* **14**(1). (doi:10.5962/bhl.title.60499).
45. Gauthier J.A., Padian K. 2020 Archosauria. In *Phylonyms: A Companion to the PhyloCode*
(eds. de Queiroz K., Cantino P.D., Gauthier J.A.), pp. 1187-1193. Boca Raton, FL, CRC Press.
46. Owen R. 1842 Report on British fossil reptiles, part II. *Report for the British Association for*
*the Advancement of Science, Plymouth* **11**, 60-294.
47. Langer M.C., Novas F.E., Bittencourt J.S., Ezcurra M.D., Gauthier J.A. 2020 Dinosauria. In
*Phylonyms: A Companion to the PhyloCode* (eds. de Queiroz K., Cantino P.D., Gauthier J.A.), pp.
1209-1217. Boca Raton, FL, CRC Press.
48. Marsh O.C. 1881 Principal characters of American Jurassic dinosaurs. *American Journal of*
*Science (3rd series)* **21**, 417-423.
49. Naish D., Cau A., Holtz T.R., Fabbri M., Gauthier J.A. 2020 Theropoda. In *Phylonyms: A*
*Companion to the PhyloCode* (eds. de Queiroz K., Cantino P.D., Gauthier J.A.), pp. 1235-1246. Boca
Raton, FL, CRC Press.
50. Bakker R.T. 1986 *The Dinosaur Heresies*, William Morrow.
51. Sereno P.C. 1998 A rationale for phylogenetic definitions, with application to the higher-level
taxonomy of Dinosauria. *Neues Jahrbuch für Geologie und Paläontologie-Abhandlungen* **210**(1), 41-
83. (doi:10.1127/njgpa/210/1998/41).
52. Nopcsa F.B. 1928 The genera of reptiles. *Palaeobiologica* **1**, 168-188.
53. Griffin C. 2018 Developmental patterns and variation among early theropods. *Journal of*
*Anatomy* **232**(4), 604-640. (doi:10.1111/joa.12775).
54. Griffin C., Nesbitt S.J. 2016 The femoral ontogeny and long bone histology of the Middle
Triassic (?late Anisian) dinosauriform *Asilisaurus kongwe* and implications for the growth of early

- dinosaurs. *Journal of Vertebrate Paleontology* **36**(3), e1111224. (doi:10.1080/02724634.2016.1111224).
55. Griffin C.T., Nesbitt S.J. 2016 Anomalously high variation in postnatal development is ancestral for dinosaurs but lost in birds. *Proceedings of the National Academy of Sciences* **113**(51), 14757-14762. (doi:10.1073/pnas.1613813113).
56. Griffin C.T., Nesbitt S.J. 2020 Does the maximum body size of theropods increase across the Triassic–Jurassic boundary? Integrating ontogeny, phylogeny, and body size. *The Anatomical Record* **303**(4), 1158-1169. (doi:10.1002/ar.24130).
57. Ezcurra M.D., Butler R.J., Maidment S.C., Sansom I.J., Meade L.E., Radley J.D. 2021 A revision of the early neotheropod genus *Sarcosaurus* from the Early Jurassic (Hettangian–Sinemurian) of central England. *Zoological Journal of the Linnean Society* **191**(1), 113-149. (doi:10.1093/zoolinnean/zlaa054).
58. Martínez R.N., Sereno P.C., Alcober O.A., Colombi C.E., Renne P.R., Montañez I.P., Currie B.S. 2011 A basal dinosaur from the dawn of the dinosaur era in southwestern Pangaea. *Science* **331**(6014), 206-210. (doi:10.1126/science.1198467).
59. Wilson J.A. 1999 A nomenclature for vertebral laminae in sauropods and other saurischian dinosaurs. *Journal of vertebrate Paleontology* **19**(4), 639-653. (doi:10.1080/02724634.1999.10011178).
60. Raath M.A. 1977 The anatomy of the Triassic theropod *Syntarsus rhodesiensis* (Saurischia: Podokesauridae) and a consideration of its biology. Salisbury, Rhodesia, Rhodes University.
61. Marsh A.D., Rowe T.B. 2020 A comprehensive anatomical and phylogenetic evaluation of *Dilophosaurus wetherilli* (Dinosauria, Theropoda) with descriptions of new specimens from the Kayenta Formation of northern Arizona. *Journal of Paleontology* **94**(S78), 1-103. (doi:10.1017/jpa.2020.14).
62. Rinehart L.F., Lucas S.G., Heckert A.B., Spielmann J.A., Celleskey M.D. 2009 The Paleobiology of *Coelophysis bauri* (Cope) from the Upper Triassic (Apachean) Whitaker quarry, New Mexico, with detailed analysis of a single quarry block. *New Mexico Museum of Natural History and Science Bulletin* **45**, 1-260.
63. You H.-L., Azuma Y., Wang T., Wang Y.-M., Dong Z.-M. 2014 The first well-preserved coelophysoid theropod dinosaur from Asia. *Zootaxa* **3873**(3), 233-249. (doi:10.11646/zootaxa.3873.3.3).
64. Cuny G., Galton P.M. 1993 Revision of the Airel theropod dinosaur from the Triassic-Jurassic boundary (Normandy, France). *Neues Jahrbuch für Geologie und Paläontologie Abhandlungen* **187**(3), 261-288.
65. Carpenter K. 1997 A giant coelophysoid (Ceratosauria) theropod from the Upper Triassic of New Mexico, USA. *Neues Jahrbuch für Geologie und Paläontologie-Abhandlungen* **205**(2), 189-208. (doi:10.1127/njgpa/205/1997/189).
66. Sereno P.C. 1999 The evolution of dinosaurs. *Science* **284**(5423), 2137-2147. (doi:10.1126/science.284.5423.2137).
67. Smith N.D., Makovicky P.J., Hammer W.R., Currie P.J. 2007 Osteology of *Cryolophosaurus ellioti* (Dinosauria: Theropoda) from the Early Jurassic of Antarctica and implications for early theropod evolution. *Zoological Journal of the Linnean Society* **151**(2), 377-421. (doi:10.1111/j.1096-3642.2007.00325.x).
68. Wilson J.A., Michael D., Ikejiri T., Moacdieh E.M., Whitlock J.A. 2011 A nomenclature for vertebral fossae in sauropods and other saurischian dinosaurs. *PLoS One* **6**(2), e17114. (doi:10.1371/journal.pone.0017114).
69. Colbert E.H. 1989 The Triassic dinosaur *Coelophysis*. *Museum of Northern Arizona, Bulletin* **57**, 1-160.
70. Nesbitt S.J. 2011 The early evolution of archosaurs: relationships and the origin of major clades. *Bulletin of the American Museum of Natural History* **352**, 1-292. (doi:10.1206/352.1).

- 71. Raath M.A. 1990 Morphological variation in small theropods and its meaning in systematics: evidence from *Syntarsus*. In *Dinosaur Systematics: Perspectives and Approaches* (eds. Carpenter K., Currie P.J.), pp. 91-105. Cambridge, Cambridge University Press.
- 72. Tykoski R.S. 2005 Anatomy, ontogeny, and phylogeny of coelophysoid theropods.
- 73. Zahner M., Brinkmann W. 2019 A Triassic averostran-line theropod from Switzerland and the early evolution of dinosaurs. *Nature ecology & evolution* **3**(8), 1146-1152. (doi:10.1038/s41559-019-0941-z).
- 74. Hutchinson J.R. 2001 The evolution of pelvic osteology and soft tissues on the line to extant birds (Neornithes). *Zoological Journal of the Linnean Society* **131**(2), 123-168. (doi:10.1111/j.1096-3642.2001.tb01313.x).
- 75. Rowe T. 1989 A new species of the theropod dinosaur *Syntarsus* from the Early Jurassic Kayenta Formation of Arizona. *Journal of Vertebrate Paleontology* **9**(2), 125-136. (doi:10.1080/02724634.1989.10011748).
- 76. Rauhut O.W.M. 2003 The interrelationships and evolution of basal theropods (Dinosauria, Saurischia). *Special Papers in Palaeontology* **69**, 1-213.
- 77. Ezcurra M.D., Cuny G. 2007 The coelophysoid *Lophostropheus airelensis*, gen. nov.: a review of the systematics of "*Liliensternus*" *airelensis* from the Triassic–Jurassic outcrops of Normandy (France). *Journal of Vertebrate Paleontology* **27**(1), 73-86. (doi:10.1671/0272-4634(2007)27[73:TCLAGN]2.0.CO;2).
- 78. Tykoski R.S. 1998 The osteology of *Syntarsus kayentakatae* and its implications for ceratosaurid phylogeny. Austin, University of Texas at Austin.
- 79. Ezcurra M.D. 2016 The phylogenetic relationships of basal archosauromorphs, with an emphasis on the systematics of proterosuchian archosauriforms. *PeerJ* **4**, e1778. (doi:10.7717/peerj.1778).
- 80. Camp C.L. 1936 A new type of small bipedal dinosaur from the Navajo Sandstone of Arizona. *Bulletin of the Department of Geological Sciences* **24**, 39-56.
- 81. Carrano M.T., Hutchinson J.R., Sampson S.D. 2005 New information on *Segisaurus halli*, a small theropod dinosaur from the Early Jurassic of Arizona. *Journal of Vertebrate Paleontology* **25**(4), 835-849. (doi:10.1671/0272-4634(2005)025[0835:NIOSHA]2.0.CO;2).
- 82. Galton P., Jensen J.A. 1979 A new large theropod dinosaur from the Upper Jurassic of Colorado. *Brigham Young University Geology Studies* **26**(2), 1-12.
- 83. Martill D.M., Vidovic S.U., Howells C., Nudds J.R. 2016 The oldest Jurassic dinosaur: a basal neotheropod from the Hettangian of Great Britain. *PLoS One* **11**(1), e0145713. (doi:10.1371/journal.pone.0145713).
- 84. Langer M.C., Rincón A.D., Ramezani J., Solórzano A., Rauhut O.W. 2014 New dinosaur (Theropoda, stem-Averostra) from the earliest Jurassic of the La Quinta formation, Venezuelan Andes. *Royal Society Open Science* **1**(2), 140184. (doi:10.1098/rsos.140184).
- 85. Hutchinson J.R. 2001 The evolution of femoral osteology and soft tissues on the line to extant birds (Neornithes). *Zoological Journal of the Linnean Society* **131**(2), 169-197. (doi:10.1111/j.1096-3642.2001.tb01314.x).
- 86. Langer M.C., Ezcurra M.D., Bittencourt J.S., Novas F.E. 2010 The origin and early evolution of dinosaurs. *Biological Reviews* **85**(1), 55-110. (doi:10.1111/j.1469-185X.2009.00094.x).
- 87. Novas F.E. 1996 Dinosaur monophyly. *Journal of Vertebrate Paleontology* **16**(4), 723-741. (doi:10.1080/02724634.1996.10011361).
- 88. Nesbitt S.J., Smith N.D., Irmis R.B., Turner A.H., Downs A., Norell M.A. 2009 A complete skeleton of a Late Triassic saurischian and the early evolution of dinosaurs. *Science* **326**(5959), 1530-1533. (doi:10.1126/science.1180350).
- 89. Novas F.E. 1994 New information on the systematics and postcranial skeleton of *Herrerasaurus ischigualastensis* (Theropoda: Herrerasauridae) from the Ischigualasto Formation (Upper Triassic) of Argentina. *Journal of Vertebrate Paleontology* **13**(4), 400-423. (doi:10.1080/02724634.1994.10011523).

90. Sereno P.C., Martínez R.N., Alcober O.A. 2013 Osteology of *Eoraptor lunensis* (Dinosauria, Sauropodomorpha). *Journal of Vertebrate Paleontology* **32**(sup1), 83-179. (doi:10.1080/02724634.2013.820113).
91. Novas F.E., Agnolin F., Ezcurra M.D., Müller R., Martinelli A., Langer M.C. in press Review of the fossil record of early dinosaurs from South America, and its phylogenetic implications. *Journal of South American Earth Sciences*.
92. Ezcurra M.D., Brusatte S.L. 2011 Taxonomic and phylogenetic reassessment of the early neotheropod dinosaur *Camposaurus arizonensis* from the Late Triassic of North America. *Palaeontology* **54**(4), 763-772. (doi:10.1111/j.1475-4983.2011.01069.x).
93. Sues H.-D., Nesbitt S.J., Berman D.S., Henrici A.C. 2011 A late-surviving basal theropod dinosaur from the latest Triassic of North America. *Proceedings of the Royal Society B: Biological Sciences* **278**(1723), 3459-3464. (doi:10.1098/rspb.2011.0410).
94. Ezcurra M.D. 2017 A new early coelophysoid neotheropod from the Late Triassic of northwestern Argentina. *Ameghiniana* **54**(5), 506-538. (doi:10.5710/AMGH.04.08.2017.3100).
95. Marsh A.D., Parker W.G., Langer M.C., Nesbitt S.J. 2019 Redescription of the holotype specimen of *Chindesaurus bryansmalli* Long and Murry, 1995 (Dinosauria, Theropoda), from Petrified Forest National Park, Arizona. *Journal of Vertebrate Paleontology* **39**(3), e1645682. (doi:10.1080/02724634.2019.1645682).
96. Marsola J.C., Bittencourt J.S., Butler R.J., Da Rosa Á.A., Sayão J.M., Langer M.C. 2018 A new dinosaur with theropod affinities from the Late Triassic Santa Maria Formation, South Brazil. *Journal of Vertebrate Paleontology* **38**(5), e1531878. (doi:10.1080/02724634.2018.1531878).
97. Martínez R.N., Apaldetti C. 2017 A Late Norian—Rhaetian Coelophysid Neotheropod (Dinosauria, Saurischia) from the Quebrada Del Barro Formation, Northwestern Argentina. *Ameghiniana* **54**(5), 488-505. (doi:10.5710/AMGH.09.04.2017.3065).
98. Nesbitt S.J., Ezcurra M.D. 2015 The early fossil record of dinosaurs in North America: a new neotheropod from the base of the Upper Triassic Dockum Group of Texas. *Acta Palaeontologica Polonica* **60**(3), 513-526. (doi:10.4202/app.00143.2014).
99. Griffin C.T. 2019 Large neotheropods from the Upper Triassic of North America and the early evolution of large theropod body sizes. *Journal of Paleontology* **93**(5), 1010-1030. (doi:10.1017/jpa.2019.13).
100. Ezcurra M.D., Nesbitt S.J., Bronzati M., Dalla Vecchia F.M., Agnolin F.L., Benson R.B., Egli F.B., Cabreira S.F., Evers S.W., Gentil A.R. 2020 Enigmatic dinosaur precursors bridge the gap to the origin of Pterosauria. *Nature* **588**, 445-449. (doi:10.1038/s41586-020-3011-4).
101. Goloboff P.A., Catalano S.A. 2016 TNT version 1.5, including a full implementation of phylogenetic morphometrics. *Cladistics* **32**(3), 221-238. (doi:10.1111/cla.12160).
102. Campione N.E., Evans D.C. 2012 A universal scaling relationship between body mass and proximal limb bone dimensions in quadrupedal terrestrial tetrapods. *Bmc Biology* **10**(1), 1-22. (doi:10.1186/1741-7007-10-60).
103. Team R.C. 2020 R: A language and environment for statistical computing. (ed. Computing R.F.f.S.). Vienna, Austria, <https://www.R-project.org/>.
104. Cohen K.M., Harper D.A.T., Gibbard P.L. 2020 ICS International Chronostratigraphic Chart 2020/03. (International Commission on Stratigraphy, IUGS).
105. Bapst D.W. 2012 paleotree: an R package for paleontological and phylogenetic analyses of evolution. *Methods in Ecology and Evolution* **3**(5), 803-807. (doi:10.1111/j.2041-210X.2012.00223.x).
106. Revell L.J. 2012 phytools: an R package for phylogenetic comparative biology (and other things). *Methods in ecology and evolution* **3**(2), 217-223. (doi:10.1111/j.2041-210X.2011.00169.x).
107. Campione N.E., Brink K.S., Freedman E.A., McGarrity C.T., Evans D.C. 2013 '*Glishades ericksoni*', an indeterminate juvenile hadrosaurid from the Two Medicine Formation of Montana: implications for hadrosauroid diversity in the latest Cretaceous (Campanian-Maastrichtian) of western North America. *Palaeobiodiversity and Palaeoenvironments* **93**(1), 65-75. (doi:10.1007/s12549-012-0097-1).

108. Tsuihiji T., Watabe M., Tsogtbaatar K., Tsubamoto T., Barsbold R., Suzuki S., Lee A.H., Ridgely
R.C., Kawahara Y., Witmer L.M. 2011 Cranial osteology of a juvenile specimen of *Tarbosaurus bataar*
(Theropoda, Tyrannosauridae) from the Nemegt Formation (Upper Cretaceous) of Bugin Tsav,
Mongolia. *Journal of Vertebrate Paleontology* **31**(3), 497-517. (doi:10.1080/02724634.2011.557116).
109. Wang S., Stiegler J., Amiot R., Wang X., Du G.-h., Clark J.M., Xu X. 2017 Extreme ontogenetic
changes in a ceratosaurian theropod. *Current Biology* **27**(1), 144-148.
(doi:10.1016/j.cub.2016.10.043).
110. Irmis R.B. 2010 Evaluating hypotheses for the early diversification of dinosaurs. *Earth and*
*Environmental Science Transactions of the Royal Society of Edinburgh* **101**(3-4), 397-426.
(doi:10.1017/S1755691011020068).
111. Benson R.B., Campione N.E., Carrano M.T., Mannion P.D., Sullivan C., Upchurch P., Evans
D.C. 2014 Rates of dinosaur body mass evolution indicate 170 million years of sustained ecological
innovation on the avian stem lineage. *PLoS Biology* **12**(5), e1001853.
(doi:10.1371/journal.pbio.1001853).
112. Lee M.S., Cau A., Naish D., Dyke G.J. 2014 Sustained miniaturization and anatomical
innovation in the dinosaurian ancestors of birds. *Science* **345**(6196), 562-566.
(doi:10.1126/science.1252243).

Figure 1. Holotype NHMUK PV R 37591 pelvis and vertebrae of *Pendraig orynys* gen. et sp. nov. in (A) left lateral view, (B) right lateral view. Abbreviations: atr, antitrochanter; bf, brevis fossa; bfr, brevis fossa rim; bs, brevis shelf; dv, dorsal vertebra; iss, ischial shaft; nc, neural canal; no, notch; obf, obturator foramen; poap, postacetabular process; prap, preacetabular process; puf, pubic fenestra; pus, pubic shaft; ras, rib attachment scar; ri, rib; sac, supra-acetabular crest; sv, sacral vertebra.

Figure 2. Holotype NHMUK PV R 37591 pelvis and vertebrae of *Pendraig orynys* gen. et sp. nov. in (A) dorsal view, (B) ventral view, (C) anterior view, and (D) posterior view. Abbreviations: bf, brevis fossa; bfr, brevis fossa rim; diap, diapophysis; dv, dorsal vertebra; gr, groove; il, ilium; ipis, iliac peduncle of ischium; iss, ischiadic shaft; obf, obturator foramen; poap, postacetabular process; ppdl, paradiapophyseal lamina; prap, preacetabular process; puf, pubic fenestra; pus, pubic shaft; sac, supra-acetabular crest; sv, sacral vertebra; tp, transverse process; vl, ventral lamina.

Figure 3. Holotype NHMUK PV R 37591 left femur of *Pendraig orynys* gen. et sp. nov. in (A) posteromedial, (B) anterolateral, (C) anteromedial, (D) posterolateral, (E) proximal, and (F) distal view. Abbreviations: amt, anteromedial tuber; at, anterior trochanter; icfl, depression associated with the insertion of the *M. caudofemoralis longus*; gt, greater trochanter; lica, linea intermuscularis caudalis; lincr, linea intermuscularis cranialis; obr, oblique anterior ridge; pmt, posteromedial tuber; ts, trochanteric shelf; 4th t, fourth trochanter.

Figure 4. Isolated mid to posterior dorsal vertebra NHMUK PV 37596 of *Pendraig orynys* gen. et sp. nov. in (A) right lateral view, (B) left lateral view, (C) ventral view, (D) dorsal view, (E) anterior view, and (F) posterior view. Abbreviations: aas, anterior articular surface; acpl, anterior centroparapophyseal lamina; ce, centrum; diap, diapophysis; nf, nutrient foramen; ns, neural spine; pacdf, paradiapophyseal centrodiapophyseal fossa; pacprf, parapophyseal centroprezygapophyseal fossa; pap, parapophysis; pas, posterior articular surface; pcpl, posterior centroparapophyseal lamina; pocdf, postzygapophyseal centrodiapophyseal fossa; podl, postzygodiapophyseal lamina; poz, postzygapophysis; ppdl, paradiapophyseal lamina; prpl, prezygaparapophyseal lamina; prz, prezygapophysis; sprzf, spinoprezygapophyseal fossa; sprzf, spinoprezygapophyseal fossa.

Figure 5. Isolated partial left ischium NHMUK PV R 37597 of *Pendraig orynys* gen. et sp. nov. in (A) medial and (B) dorsal view. Abbreviations: asil, articulation surface with ilium; atr, antitrochanter; ipis, iliac peduncle of ischium.

- Taxon**
- *Coelophysidae indet.*
 - *Coelophysis bauri*
 - *Dilophosaurus wetherilli*
 - *Eodromaeus murphi*
 - *Liliensternus liliensterni*
 - *Megapnosaurus rhodesiensis*
 - *'Syntarsus' kayentakatae*
 - *Pendraig orynys (estimated)*

$y = 0.845x + 1.013$
 $r^2 = 0.968, p < 2.2 \times 10^{-16}$

Figure 9. Life reconstruction of *Pendraig orynys* gen. et sp. nov. amongst the fissures of Pant-y-ffynnon and three individuals of the rhynchocephalian lepidosaur *Clevosaurus cambrica* during the Late Triassic. Artwork by James Robbins.

322x239mm (300 x 300 DPI)

Appendix B

This manuscript presents a thorough comparative description of a new theropod dinosaur and places it into a phylogenetic hypothesis using a relevant and recent character matrix. The small size of the holotype individual is striking, so the authors also conduct an ancestral state reconstruction for body size among early theropod dinosaurs and also attempt to take the individual's ontogenetic status into account to be sure that the small size does not simply stem from the individual's immaturity.

The manuscript is well-written and thorough, the comparative descriptions sound, the figures are clear and informative, and the analyses all appear to be properly conducted (but see below for problems with the R code). In my opinion the manuscript is largely sound and can be accepted for publication with moderate revision. I have attached a PDF with my minor edits and comments.

Major Comments

1) Why are the eleven supplementary figures provided as separate files, with yet another Word file for the captions? It would be much easier to include all figures with associated captions as one PDF file. Also, in Table S2 in the "Juvenile" column, there are several different terms used for maturity assessment, including juvenile, non-juvenile, subadult, immature, etc. Sometimes these seem to be synonymous, but their usage at other times appears to be mutually exclusive. For example, some taxa are listed as "SUBADULT/IMMATURE" and others are just listed as "IMMATURE". Could you provide more explanation for these maturity categories?

2) When running the R code, I experienced an error code that prevented me from continuing to evaluate the rest of the code. The error occurred on line 42, on the `anc.ML()` command, and read: "Error in `optim(c(sig2, a, y, rep(mean(x), length(xx))), fn = likelihood, : non-finite value supplied by optim`".

All the underlying data looked sound to me; I saw no obvious issues with the way the data was input, the tree file, etc. It may be the issue is with my version of R or R Studio, because I just updated both two days ago, but this affects repeatability and is something the authors should be aware of.

3) I appreciate that the authors take the body size of the type individual into account when evaluating possible small body size. This is done in a clear way using character state transformations that have been useful in other early theropods, particularly *Coelophysis*. I agree with the authors' assessment that this individual does not display the features we might expect of either a very skeletally mature *or* immature individual, and the character states instead suggest that this individual is in a middling 'gray zone' of ontogeny. Indeed, there is no reasonably complete specimen of *Coelophysis* that has been scored with a consistent combination of character states (data from Griffin & Nesbitt 2016, Griffin 2018), making direct comparison with *Coelophysis* character states and femoral lengths difficult. This, combined with the fact that their scoring of *Panguraptor* ontogenetic character state combinations that contradict any reconstructed ontogenetic sequence of *Coelophysis* (very interesting finding, by

the way), strongly suggests that there is even more variation in ontogenetic trajectories among Coelophysoids than has presently been reconstructed.

Because of the large amount of ontogenetic variation known from other coelophysoids, combined with the fact that most of the size variation in Coelophysis is known precisely from this ontogenetic gray zone, I disagree with the authors' statement that this individual was likely near maximum body size and would not have gotten much larger. Instead, I think that there is not enough evidence to say one way or another.

Although there is no direct comparison between the character states of Pendraig and Coelophysis or Megapnosaurus, some examples from individuals roughly the size of Pendraig that also display similarly mature character states may be informative:

--The smallest known individual of Megapnosaurus (NHMZ QG 45) possesses fused sacral neural spines (character 1-1; and therefore probably at least some fused sacral centra, although these are not able to be scored) and a small trochanteric shelf (14-1; 15-0), despite its extremely small size (femoral head 1.5 cm; reconstructed femoral length ~11.2 cm).

--Coelophysis bauri (TMP 1984.063.0001 #13; reconst. femur length ~10.9 cm) has fused its pubis to the ilium (8-1) and ischium (10-1), but the ilium and ischium remain unfused (9-1).

-- Coelophysis bauri (SMP 858; femoral head 1.29 cm, reconst. femur length ~10.2 cm) possesses five fused sacrals (2-2), a fused pubis (8-1), a trochanteric shelf (14-1), a mound-like dorsolateral trochanter (16-1), a cranial intermuscular line (17-1), a caudal intermuscular line (18-1), and an 'anterolateral scar' (19-1).

-- Coelophysis bauri (CMNH 10971 #3; femur length 10.94 cm) possesses five fused sacrals (2-2), a fused pubis (8-1), a trochanteric shelf (14-1), a mound-like dorsolateral trochanter (16-1), a cranial intermuscular line (17-1), a caudal intermuscular line (18-1), and an 'anterolateral scar' (19-1).

There are of course smaller individuals of Coelophysis that display more immature character states as well. But my point here is that finding an individual of Coelophysis that displays a similar body size and similar ontogenetic character states is not unprecedented. Because we know that Coelophysis can reach much larger sizes (~25 cm femoral length), then with a sample size of 1, this is not great evidence that Pendraig's maximum size is close to what this individual's is; this taxon could just happen to be represented by one of those more mature-looking but anomalously small individuals also found in other coelophysoids. Therefore, although the data are *consistent* with Pendraig having a small maximum body size, I do not think the data support saying that it *did* have a small maximum body size.

4) This brings me to my suggestion that the best way to resolve this issue of maturity and body size is by histologically sampling the individual. I understand that the authors probably have qualms about destructively sampling a holotype and only known specimen (barring the referred vertebra) of a taxon, but I do have two ideas that may alleviate some concerns.

I notice in Figure 3 that the femur is already somewhat damaged at midshaft. I suggest that, instead of sampling the entire cross-section of the femoral shaft, you break off a small piece of already-damaged cortex from this midshaft region and histologically sample that (see picture below to see what I mean). I have successfully used this technique when I did not want to damage the full specimen but just sampled a portion that was already damaged, including on

a femur of *Dromomeron romeri* (Griffin et al. 2019, PeerJ), and on a femur of *Coelophysis* (CMNH 10971; Barta et al. in prep). Although a full cortical sample is ideal, with a partial cortex you can still see, for example, LAGs, LAG spacing, an EFS if present, etc. I used this method to find an EFS in the *Coelophysis* individual referenced above.

Another option, and slightly more heterodox, would be attempting to sample the distalmost preserved end of the ischium, which is roughly midshaft. Pelvic elements are not often sampled, but in my experience any long, somewhat tubular endochondral bone preserves a record of growth and can be useful for histological maturity assessment when sampled near midshaft. I have seen this work on an immature Tyrannosaurid pubis (pers. obs.), metatarsals (McLain et al. 2018, Palaios), and hyoid elements (pers. obs., submitted to be a 2021 SVP poster). A 2011 Master's thesis showed that the midshaft of the Alligator pubis is skeletochronologically informative, and the ischium did not work only because it is platelike, not elongate, which is not an issue for Pendraig (Garcia 2011, "Skeletochronology of the American Alligator (*Alligator Mississippiensis*): Examination of the Utility of Elements for Histological Study", Florida State University). The ischium of Pendraig is broken at roughly midshaft, so only ~1 cm of the ischium would need to be removed to make a histological section, causing minimal damage to the specimen. In fact, a combination of sampling from a femoral fragment and the ischium would be a good multi-elemental way to assess maturity and back up the assessment made by morphology, making the assessment supported by multiple lines of evidence.

I do not hinge my final approval on whether histological sampling is conducted on this specimen, and I think the authors can incorporate the concerns from my comment #3 without including histology. I know that it is not always possible or desirable to destructively sample a holotype. However, I do think that it would make the paper stronger, more convincing, and more citable, and therefore I recommend the authors add this to their study, if it is possible.

The authors are free to contact me with any questions, requests for clarification, or concerns.

Chris Griffin
chris.griffin@yale.edu

Modified figures showing suggested sampling locations on next two pages:

Appendix C**ROYAL SOCIETY
OPEN SCIENCE****Pendraig orynys, a new small-sized coelophysoid theropod
from the Late Triassic of Wales**

Journal:	Royal Society Open Science
Manuscript ID	RSOS-210915
Article Type:	Research
Date Submitted by the Author:	07-Jun-2021
Complete List of Authors:	Spiekman, Stephan; Natural History Museum, Department of Earth Sciences; University of Birmingham, Earth and Environmental Sciences Ezcurra, Martín; CONICET, Sección Paleontología de Vertebrados, Museo Argentino de Ciencias Naturales; University of Birmingham, School of Geography, Earth and Environmental Sciences Butler, Richard; University of Birmingham, School of Geography and Earth Sciences Fraser, Nicholas; National Museums Scotland Maidment, Susannah; Natural History Museum
Subject:	palaeontology < BIOLOGY, evolution < BIOLOGY, taxonomy and systematics < BIOLOGY
Keywords:	Pendraig, Coelophysoidea, Theropoda, Triassic, Body size evolution, Osteology
Subject Category:	Organismal and Evolutionary Biology

Author-supplied statements

Relevant information will appear here if provided.

Ethics

Does your article include research that required ethical approval or permits?:

This article does not present research with ethical considerations

Statement (if applicable):

CUST_IF_YES_ETHICS :No data available.

Data

It is a condition of publication that data, code and materials supporting your paper are made publicly available. Does your paper present new data?:

Yes

Statement (if applicable):

All data are provided for initial review as supplementary material.

Zip folders are provided for the R code and associated data files for both the regression and ancestral state reconstruction analyses. The initial code for the ancestral state reconstruction is provided for the analysis figured in Figure 8. Th parameters can easily be changed manually by reviewers familiar with R to perform the analyses under the various settings provided in the Supplements. Different MPTs and the femoral lengths used for the conservative analyses are provided in the zip folder and minimum branch length can easily be adjusted from 1.0 to 0.5 and 0.1 in the code.

Conflict of interest

I/We declare we have no competing interests

Statement (if applicable):

CUST_STATE_CONFLICT :No data available.

Authors' contributions

This paper has multiple authors and our individual contributions were as below

Statement (if applicable):

SNFS, MDE, RJB, and SCRM designed the study. SNFS and MDE contributed to data collection. SNFS, MDE, and RJB analysed the data. SNFS made the figures and wrote the majority of the manuscript. All authors contributed to the writing and reviewing of the manuscript.

Pendraig orynys, a new small-sized coelophysoid theropod from the Late Triassic of Wales

Stephan N. F. Spiekman^{1,2}, Martín D. Ezcurra^{2,3}, Richard J. Butler², Nicholas C. Fraser⁴, Susannah C. R. Maidment^{1,2}

¹Department of Earth Sciences, Natural History Museum, Cromwell Road, London SW7 5BD, UK

²School of Geography, Earth and Environmental Sciences, University of Birmingham, Edgbaston, Birmingham B15 2TT, UK

³Sección Paleontología de Vertebrados, CONICET-Museo Argentino de Ciencias Naturales, Ángel Gallardo 470, C1405DJR, Buenos Aires, Argentina

⁴National Museums Scotland, Chambers St, Edinburgh EH1 1JF, UK

*corresponding author: stephanspiekman@gmail.com

Abstract.

We describe a new genus and species of small-bodied coelophysoid theropod dinosaur, *Pendraig orynys* 
[revised manuscript text omitted]

*lufengensis* ([63], fig. 1: ratio = c. 2.0–2.11 in the last two dorsal vertebrae), and *Megapnosaurus*

*rhodesiensis* ([60]: ratio = 2.10, D13 of NHMB QG 1). By contrast, these vertebrae are considerably
proportionally shorter in *Liliensternus liliensterni* (HMN MB.R.2175 2.22–2.24: ratio = 1.58–1.67,
posterior dorsal vertebrae), *Lucianovenator bonoi* (PVSJ 906: ratio = 1.63, last dorsal vertebra),
*Lophostropheus airelensis* ([64]: ratio = 1.32, last dorsal vertebra), *Dracoraptor hanigani* (NMW
2015.5G.1–2015.5G.11: ratio = 1.63–1.75, middle–posterior dorsal vertebrae), *Cryolophosaurus*
*elliotti* (FMNH PR1821: ratio 1.07, posterior dorsal vertebra), *Dilophosaurus wetherilli* ([61]: ratio =
1.16–1.52, D10, D11 and D13 of UCMP 37302; ratio = 1.19 D14 of UCMP 77270), and *Sarcosaurus*
*woodi* ([57]: ratio = c. 1.9, middle–posterior dorsal vertebra of WARMS G678). The ventral surface of
the centrum is anteroposteriorly concave and lacks a ventral keel (Fig. 1B). The centrum is
amphiplatyan with very slightly concave anterior and posterior articular surfaces. As in the preceding
vertebra, no visible suture is present between the centrum and neural arch. The lateral surface of
the centrum bears an anteroposteriorly elongate but shallow fossa just ventral to the articulation
with the neural arch, which is a common condition in the middle-posterior dorsal vertebrae of early
neotheropods (e.g., *Liliensternus liliensterni*: HMN MB.R.2175; *Procompsognathus triassicus*: SMNS
12591; *Lucianovenator bonoi*: PVSJ 906; *Lophostropheus airelensis* [64]; *Sarcosaurus woodi* [57]). The
last dorsal vertebra possessed only a single articular facet for the rib on each side, located at the end
of a transversely wide, wing-like transverse process (Fig. 2A). In dorsal view, its posterior margin is
concave and its anterior margin appears to be somewhat sinusoidal. There is no distinct fossa on the
dorsal surface of the base of the transverse process. The articular surfaces of the prezygapophyses
face dorsomedially. The articular surface of the left postzygapophysis is poorly preserved. No
hyposphene articular surface is preserved, but this region is poorly preserved. The prezygapophyses
diverge from each other in dorsal view and their tips are well separated from the median line,
contrasting with the sub-parallel prezygapophyses of *Sarcosaurus woodi* [57].

The isolated dorsal vertebra NHMUK PV 37596 is virtually complete, undistorted, and freed from
matrix (Fig. 4). The centrum is 14.6 mm long and 2.59 times longer than the height of its anterior
articular surface (Table 1). The elongation of this centrum matches or closely resembles that of the
middle dorsal vertebrae of the coelophysids *Coelophysis bauri* (e.g., NMV P231382: c. 2.6, middle
dorsal vertebra), *Megapnosaurus rhodesiensis* ([60]: 2.33–2.64, D6 and D7 of NHMB QG 1), and
*Procompsognathus triassicus* (SMNS 12591: 2.70–2.91, D7 and D8), whereas in other early
neotheropods the middle-posterior dorsal vertebrae are proportionally shorter (e.g., *Liliensternus*
*liliensterni*, *Gojirasaurus quayi*; *Dilophosaurus wetherilli*, *Dracoraptor hanigani*, *Cryolophosaurus*
*elliotti*; *Sarcosaurus woodi*) [57, 61, 65–67]. The ventral surface of NHMUK PV 37596 is concave in
lateral view and there is no ventral keel. A single nutrient foramen can be observed close to the
anterior end of the centrum on its right ventrolateral side. The anterior and posterior articular

surfaces of the centrum are both very slightly concave and transversely broader than tall, resembling
the condition in *Gojirasaurus quayi* [65]. By contrast, the centrum is taller than broad in the
posterior dorsal vertebrae of *Megapnosaurus rhodesiensis* ([60]: table 6), *Sarcosaurus woodi* (only
the anterior surface is preserved) [57], *Cryolophosaurus ellioti* [67], and *Dilophosaurus wetherilli*
[61], approximately as broad as tall in *Liliensternus liliensterni* (HMN MB.R.2175), and both
conditions occur in *Eodromaeus murphi* (PVSJ 562). The lateral surfaces of the centrum bear a
shallow fossa directly ventral to the connection to the neural arch, as occurs in NHMUK PV R 37591
and other early neotheropods (see above). The neurocentral suture is closed along most of its
extension, being only visible on the most posterior region of the neural arch peduncle on both sides
of the bone.

The diapophysis is placed on a wide sub-trapezoidal or wing-like transverse process (Fig. 4D). In
dorsal view, the posterior margin of this process is mainly laterally oriented and slightly concave,
whereas the anterior margin is anteromedially to posterolaterally oriented and somewhat
sinusoidal. The anteroposteriorly long base of the transverse process and strong posterolateral
slating of its anterior margin resemble the condition in the middle dorsal vertebrae of *Eodromaeus*
*murphi* (PVSJ 562) and the coelophysids *Coelophysis bauri* (AMNH 7224), *Megapnosaurus*
*rhodesiensis* [60], and *Procompsognathus triassicus* (SMNS 12591). The parapophysis is placed on a
strongly developed, narrow and rod-like stalk, but it is considerably less extended laterally than the
diapophysis, resembling the condition in the middle-posterior dorsal vertebrae of at least some
other early neotheropods (e.g., *Liliensternus liliensterni*: HMN MB.R.2175; *Megapnosaurus*
*rhodesiensis* [60]). Both processes are positioned fully on the neural arch and are connected through
a thin paradiapophyseal lamina (*sensu* [59]). The diapophysis is located slightly dorsal to the
parapophysis (Fig. 4A-B). The articular facet of the diapophysis is oval and anteroposteriorly
elongated, whereas the facet of the parapophysis is subcircular. The parapophyseal
centrodiapophyseal fossa ventral to the diapophysis is shallow, whereas the parapophyseal
centroprezygapophyseal and postzygapophyseal centrodiapophyseal fossae are very deep and
framed by pronounced and thin laminae (*sensu* [68]). The laminae framing the parapophyseal
prezygapophyseal fossa are the prezygaparapophyseal lamina dorsally and the anterior
centroparapophyseal lamina ventrally, whereas the postzygapophyseal centrodiapophyseal fossa is
framed by the postzygodiapophyseal lamina dorsally and the posterior centrodiapophyseal lamina
ventrally (*sensu* [59]). This pattern of laminae and fossae matches that of a posterior dorsal vertebra
of *Liliensternus liliensterni* (HMN MB.R.2175 2.22). There are no pneumatic foramina within the
fossae. The transition between the transverse process of the diapophysis and the neural spine forms
an angle of approximately 90 degrees, and there is no fossa present in this region. The

postzygapophyses are closely placed together and their articulation facets face ventrolaterally. A
hyposphene is absent between the postzygapophyses and therefore there is no accessory
intervertebral articulation (Fig. 4F), contrasting with its presence in the middle-posterior dorsal
vertebrae of *Eodromaeus murphi* (PVSJ 562), *Megapnosaurus rhodesiensis* [60], *Gojirasaurus quayi*
[65], *Cryolophosaurus ellioti* [67], *Sarcosaurus woodi* [57], and *Dilophosaurus wetherilli* [61]. The
articulation facets of the prezygapophyses face dorsomedially (Fig. 4E). Both the
spinoprezygapophyseal and spinopostzygapophyseal fossae are very narrow slit-like openings
between the pre -and postzygapophyses, respectively [68], and they do not extend onto the surface
of the neural spine. The spinopostzygapophyseal fossa is considerably larger than the
spinoprezygapophyseal one.

The neural spine is proportionally low, being 0.4 times taller than anteroposteriorly long at its base,
resembling the condition in the middle and posterior —but not the posteriormost— dorsal
vertebrae of *Coelophysis bauri* (AMNH 7224) and *Megapnosaurus rhodesiensis* [60]. By contrast,
*Eodromaeus murphi* and other early neotheropods (e.g., *Dilophosaurus wetherilli* [61]; *Gojirasaurus*

[revised manuscript text omitted]

*Liliensternus liliensterni*: HMN MB.R.2175; *Lucianovenator bonoi*: PVSJ 906; *Coelophysis* sp.: UCMP
129618; *Notatesseraeraptor frickensis* [73]; *Megapnosaurus rhodesiensis*: NHMB QG 1;
*Dilophosaurus wetherilli* [61]). The dorsal margin of the iliac blade of *Pendraig orynys* possesses a
somewhat thickened, mostly flat surface that faces slightly laterally. This flat surface extends along
most of the bone, with exception of the anteriormost region of the preacetabular process and starts
to taper anteriorly at the mid-length of this process. On the posterior region of the iliac blade, this
flat surface extends ventrally as a raised region to occupy the entire dorsoventral height of the
lateral surface of the posterior end of the postacetabular process. It has been inferred that the
anterior rim of this raised surface probably delimited the attachment site of the *M. iliofemoralis* [72,

74]. This same condition occurs in *Coelophysis bauri* (USNM 529376), *Lucianovenator bonoi* (PVSJ 899, 906), *Coelophysis* sp. (~~UCMP 129618~~), '*Syntarsus*' *kayentakatae* [75], and *Megapnosaurus rhodesiensis* (NHMB QG 1), but not in other early neotheropods [72]. The anterior margin of the preacetabular process is continuously rounded and extends considerably further anterior than the pubic peduncle of the ilium, as occurs in other neotheropods [76]. The preacetabular process is transversely very thin (i.e., laminar) and slightly medially curved in dorsal view. The ventral margin of the preacetabular process is slightly convex and oriented somewhat anteroventrally to posterodorsally. However, the overall orientation of the preacetabular process is anteriorly facing and a broad gap separates it from the pubic peduncle. This morphology corresponds to that of most theropods, but contrasts with the anteroventrally directed processes of *Sarcosaurus woodi* and some ceratosaurs (e.g., *Ceratopsaurus nasicornis*, *Eoabelisaurus mefi*) [57]. At its posterodorsal end, the postacetabular process curves abruptly posteroventrally and the posteroventral end of the process is formed by an acute angle of approximately 65 degrees in lateral view. By contrast, the posteroventral corner of the postacetabular process is approximately right-angled or slightly acute in other non-averostran neotheropods (e.g., *Coelophysis bauri*: USNM 529376; *Liliensternus liliensterni*: HMN MB.R.2175; *Lucianovenator bonoi*: PVSJ 906; *Coelophysis* sp.: ~~UCMP 129618~~; '*Syntarsus*' *kayentakatae*: Tykoski 2005; *Megapnosaurus rhodesiensis*: NHMB QG 1; *Dilophosaurus wetherilli*: Marsh & Rowe 2020; *Sarcosaurus woodi*: NHMUK PV R4840). A notch on the posterior end of the postacetabular process, as has been described for various coelophysoid taxa (e.g., *Coelophysis bauri*, *Coelophysis* sp., *Megapnosaurus rhodesiensis*, '*Syntarsus*' *kayentakatae*) [72], is absent. In dorsal view the ilium is oriented approximately straight anteroposteriorly (Fig. 2A) and the postacetabular process expands gradually laterally towards its posterior end, resembling the condition in *Liliensternus liliensterni* (HMN MB.R.2175), *Lucianovenator bonoi* (PVSJ 906), and *Dilophosaurus wetherilli* (UCMP 37302). By contrast, the the postacetabular process is distinctly more laterally expanded, extending beyond the level of the outer rim of the supra-acetabular crest in dorsal view, in *Coelophysis bauri* (USNM 529376), *Coelophysis* sp. (~~UCMP 129618~~), and *Megapnosaurus rhodesiensis* [60].

The lateral surface of the iliac blade is concave along its entire anteroposterior length. A shallow, not well-rimmed fossa is present immediately dorsal to the supra-acetabular crest and this region lacks the vertical ridge present in *Lophostropheus airelensis* [64, 77]. The ventral margin of the postacetabular process is formed by a distinct and sharp brevis shelf (Fig. 1A). The concave portion of the postacetabular process positioned medioventrally to this shelf is the brevis fossa [74]. This fossa is inferred to have formed the attachment site for the *M. caudofemoralis brevis* and is mediodorsally framed by a distinct ridge (Fig. 1B). The brevis fossa is only visible in lateral view in its anterior

portion. The remainder of the fossa faces ventrally or medioventrally and is obscured by the brevis
shelf in lateral view, a condition typical of neotheropods [70].

The acetabulum is fully perforated and mostly formed by the ilium (Fig. 1A). On the posterior surface
of the acetabulum a well posteriorly delimited, crescent-shaped rugosity is present, which
represents the antitrochanter. The dorsal portion of the antitrochanter is positioned on the ilium,
whereas most of its surface is present on the ischial portion of the acetabular margin. The
development of this antitrochanter closely resembles those observed in *Megapnosaurus*
*rhodesiensis* (NHMB QG 1), *Coelophysis bauri* (USNM 529376), *Coelophysis* sp. (UCMP 129618),
*'Syntarsus' kayentakatae* [72], and *Lucianovenator bonoi* (PVSJ 906). Dorsally, the acetabulum is
framed by a pronounced supra-acetabular crest, which projects laterally and slightly ventrally (Figs.
1A and 2C). The rim of the crest extends close to the connection with the pubis anteriorly and to the
origin of the brevis shelf posteriorly. However, the supra-acetabular crest and the brevis shelf do not
form the continuous, well-laterally developed ridge present in *Megapnosaurus rhodesiensis* (NHMB
QG 1), *Coelophysis bauri* (USNM 529376), *Lophostropheus airelensis* [77], *Procompsognathus*
*triassicus* (SMNS 12591), *'Syntarsus' kayentakatae* [72], and *Lucianovenator bonoi* (PVSJ 906). The
condition of *Pendraig orynys* resembles that of *Liliensternus liliensterni* (HMN MB.R.2175),
*Coelophysis* sp. (UCMP 129618), *Dilophosaurus wetherilli* [61], and *Sarcosaurus woodi* [57]. The
pubic peduncle is anteroventrally oriented, whereas the ischiadic peduncle is considerably more
vertically directed, facing only slightly posteroventrally. The suture between the ilium and pubis is
completely unfused. The suture with the ischium is unfused along its posterior portion, but on its
anterior portion, which is located across the antitrochanter and part of the acetabulum, the suture is
closed and the elements are indistinguishably fused.

Pubis.

Both the left and the right pubes of NHMUK PV R 37591 are largely complete and in articulation with
each other (Fig. 2C). Both elements lack the distal end of the pubic shaft, but the shaft extends
further distally in the right element than the left. Overall, the preservation of the left element is
superior to that of the right element since the surface of the latter is damaged in several places.
Therefore, the description of the pubis is largely based on the left element. The shaft of the pubis is
anteroventrally directed and elongate (Fig. 1A). Its extent is considerable but cannot be fully
assessed because the distal end is missing on both sides. The longest preserved pubis, the right
element, is 63.2 mm long. When including the imprint of the pubic shaft, which reaches further
distally but likely does not represent the distal terminus of the pubes, the maximum length is 74.8
52 mm. The shaft is rod-like with a plate-like medial apron, which is lateromedially wide and

anteroposteriorly flat (Fig. 2B-C). The anterior surface of the shaft is convex and, correspondingly,
the posterior surface is concave. As a result, the pubic shaft is slightly anteriorly curved in lateral
view as in *Coelophysis bauri* (AMNH 7223, 7224), *Megapnosaurus rhodesiensis* [60], '*Syntarsus*'
*kayentakatae* [78], *Dracoraptor hanigani* (NMW 2015.5G.1–2015.5G.11), *Procompsognathus*
*triassicus* (SMNS 12591), *Notatesseraeraptor frickensis* (SMF 06-1), and *Gojirasaurus
[revised manuscript text omitted]

*hanigani* are represented by immature specimens (Supplementary Figure 11; Supplementary Table
2) [56]. The presence of taxa based on skeletally immature specimens or with unclear ontogenetic
stage increases the proportion of missing data because ontogenetically variable characters should be
scored as ambiguous. The more derived position of '*Syntarsus*' *kayentakatae* recovered in our
analysis is, in part, a result of the scorings revised in our modified data matrix and the inclusion of
the new species *Pendraig orynys*. Indeed, if the latter species is excluded *a priori* from the analysis,
'*Syntarsus*' *kayentakatae* is recovered in multiple positions among non-coelophysid coelophysoids in
the resultant MPTs. This result reflects the importance of adding new taxa with a novel combination
of character states and the continuous revision of the data matrices in phylogenetic studies.

Body size evolution of *Pendraig orynys* and other early theropods.

Research on early theropod body size evolution has recently been reviewed by Griffin [99] and
Griffin and Nesbitt [56]. Recent analyses using ancestral state reconstruction found the femoral
length of the last common ancestor of Neotheropoda to be approximately 29 to 35 cm [99, 110,
111]. Lee et al. [112] found a considerably higher ancestral femoral length of 47.5 cm for
Neotheropoda, but the dataset used in that analysis contained a comparatively smaller sample of
early theropods. Our analyses reveal that different values used for the minimum branch length

parameter (mbl, set at 0.1, 0.5, and 1.0 million years) have quite large implications for the
reconstructed ancestral values. The analysis on the first of the four equally parsimonious trees with
mbl set at 1.0 million years and including all sampled taxa (Fig. 8) recovered an ancestral femoral
length of 24.2 cm for Neotheropoda (upper CI: 33.7 cm; lower CI: 17.3 cm) and 17.6 cm for
Coelophysoidea (upper CI: 27.6 cm; lower CI: 11.2 cm) (Supplementary Table 3.1), whereas when an
mbl of 0.1 million years is considered, the ancestral value for Neotheropoda is 39.6 cm (upper CI:
58.0 cm; lower CI: 27.1 cm) and that for Coelophysoidea is 14.9 cm (upper CI: 25.1 cm; lower CI: 8.8
10 cm) (Supplementary Table 3.3). For an mbl of 0.5 million years, the ancestral value for
Neotheropoda is 29.7 cm (upper CI: 43.0 cm; lower CI: 20.5 cm) and for Coelophysoidea 16.4 cm
(upper CI: 27.0 cm; lower CI: 9.9 cm) (Supplementary Table 3.4). When the femoral lengths of small-
bodied neotheropod taxa represented by immature specimens are pruned (i.e., taxa with a
maximum maturity score of 17 or less: *Panguraptor lufengensis*, *Lepidus praecisio*, *Dracoraptor*
*hanigani*, and *Powellvenator podocitus*), the ancestral femoral length is 25.7 cm (upper CI: 36.2 cm;
lower CI: 18.3 cm) for Neotheropoda and 21.1 cm (upper CI: 34.5 cm; lower CI: 12.9 cm) for
Coelophysoidea when considering the first of the four equally parsimonious trees and with mbl set
at 1.0 million years (Supplementary Table 3.2). The four equally parsimonious resolutions of the
early coelophysoid relationships result in similar reconstructed femoral lengths for Neotheropoda
and Coelophysoidea, with the ancestral estimates for the latter being between 17 and 18 cm when
mbl is set at 1.0 million years and the femoral lengths of all sampled taxa are included
(Supplementary Tables 3.1, 3.5-3.7). Overall, the large discrepancy in reconstructed ancestral
femoral lengths for Neotheropoda and Coelophysoidea between the different analyses, particularly
between the analyses with different minimum branch lengths, and the wide confidence interval
ranges for all values indicate that there is much uncertainty in approximating ancestral body sizes in
early neotheropods, but our recovered values are broadly similar to the results of Irmis [110],
Benson et al. [111], and Griffin [99].

Our results indicate that averostran-line neotheropods underwent a size increase already during the
Triassic (Fig. 8; Supplementary Information). In contrast, the body size of coelophysoids is
considerably smaller. This corresponds with the results of Griffin [99]. Size decreases occurred early
in Coelophysoidea and ancestral values gradually increase in consecutive nodes from the clade
encompassing *Lucianovenator bonoi*, *'Syntarsus' kayentakatae*, and Coelophysidae onwards
regardless of minimum branch length in the analyses encompassing all data (Supplementary Tables
3.1, 3.3, 3.4). In contrast, Griffin [99] found an initial increase in body size in the evolution of
Coelophysoidea, which is attributable to the placement of *Liliensternus liliensterni* (and in one of the
two analyses *Gojirasaurus quayi*) at the base of the clade in his phylogenetic analyses and the

absence of several coelophysoid taxa in that dataset (*Powellvenator podocitus*, *Procompsognathus*
*triassicus*, *Segisaurus halli*, *Lucianovenator bonoi*, *Camposaurus arizonensis*, and the new taxon
*Pendraig orynys*). In the two equally parsimonious topologies in which *Lucianovenator bonoi* is more
distantly related to Coelophysidae, ancestral femoral length increases from the clade comprising
‘*Syntarsus*’ *kayentakatae*, and Coelophysidae onwards (Supplementary Tables 3.6, 3.7). In the
analyses excluding *Panguraptor lufengensis*, *Lepidus praecisio*, *Dracoraptor hanigani*, and
*Powellvenator podocitus*, the ancestral body size of Coelophysoidea is also reduced relative to the
ancestral neotheropod condition and decreases somewhat further early in coelophysoid evolution
and subsequently remains similar for subsequent nodes, only to increase again at the most apical
node (Supplementary Table 3.2).

The ontogenetic assessment of the holotype of *Pendraig orynys* indicates that this specimen was not
skeletally mature (i.e., likely had not reached asymptotic growth [56]) but that it is likely also not at
an early ontogenetic stage (Supplementary Figure 11A). It therefore seems unlikely that *Pendraig*
*orynys* would have increased much more in size and this species would thus have been considerably
smaller than better known coelophysoids like *Coelophysis bauri*, *Megapnosaurus rhodesiensis*, and
‘*Syntarsus*’ *kayentakatae*. The ancestral femoral length for the closest node for *Pendraig orynys* is
between 144% (Supplementary Table 3.3) and 195% (Supplementary Table 3.2) larger than for
*Pendraig orynys*, thus indicating that the small size of *Pendraig orynys* is autapomorphic. However,
*Procompsognathus triassicus*, *Segisaurus halli*, and, in the analyses in which this taxon is considered,
*Powellvenator podocitus*, all independently underwent a similar size reduction based on our analyses
(Fig. 8; Supplementary Information). Because its small size is not unique among Coelophysoidea and
other coelophysoid taxa that underwent a similar size reduction were not restricted to insular
environments, our dataset is ambiguous regarding insular dwarfism as a possible explanation of the
reduced body size in *Pendraig orynys*. However, insular dwarfism in *Pendraig orynys* cannot be
excluded, and further studies into the palaeohistology and body size evolution of other taxa from
Pant-y-ffynnon and related fissure fill deposits are required to investigate the possibility that these
faunas were subject to dwarfism or other aspects of the ‘Island Rule’.

Palaeoecology of *Pendraig orynys* and Pant-y-ffynnon.

The known dentitions of coelophysoid theropods are characterized by blade-like serrated maxillary
and non-mesial dentary teeth, indicating a mostly macrophagous carnivorous diet for these taxa [60,
69, 75]. It therefore seems highly likely that *Pendraig orynys* had a similar dentition and diet even
though no craniodental remains from Pant-y-ffynnon can unequivocally be attributed to this species.
*Pendraig orynys* represents a second macrophagous predator known from Pant-y-ffynnon (Fig. 9),

the other being the non-crocodyliform crocodylomorph *Terrestrisuchus gracilis* [18]. Like *Pendraig*
*orynys*, *Terrestrisuchus gracilis* was small-bodied (approximately 76 cm in total body length [18]) and
had a gracile body plan. Other likely predators known from the Late Triassic and Early Jurassic fissure
fill deposits of southwestern England and southern Wales were either similarly small-bodied:
*Terrestrisuchus*-like unidentified crocodylomorphs from Cromhall and Ruthin Quarries [19, 32] and
‘*Agnosphytis cromhallensis*’ from Cromhall Quarry [32]; or most likely semi-aquatic (‘*Paleosaurus*
*platyodon*’) [27]. Remains of the considerably larger-sized herbivorous sauropodomorph
*Thecodontosaurus antiquus* have been preserved at the Durdham Down and Tytherington deposits
[20, 27], but only remains of the smaller sauropodomorph *Pantydraco caducus*, which might
represent an immature form of *Thecodontosaurus antiquus* [20], are known from Pant-y-ffynnon
[28, 29]. Therefore, it is currently unclear whether predators at Pant-y-ffynnon simply did not exceed
the size of *Pendraig orynys* and *Terrestrisuchus gracilis*, which could be attributable to a lack of
resources to sustain larger predators as is typical in certain island environments [22], or whether
larger-bodied predators in this ecosystem have not yet been discovered or preserved, possibly
because taphonomic factors are biased against preservation of large-bodied animals at Pant-y-
ffynnon and other fissure fill deposits.

Acknowledgements.

We thank Angela Milner (NHMUK) for relocating the specimen and providing a copy of Warrener’s
unpublished PhD thesis, Kevin Webb (NHMUK) for providing the high-resolution photographs of the
specimens used in the figures, and James Robbins for the life reconstruction used in Figure 9. Marc
Jones (UCL) and Bruce Griffiths kindly provided advice on the Welsh language in the naming of
*Pendraig orynys*. The following curators, researchers and collection managers are thanked for
providing access to specimens under their care for the purpose of this research: Carl Mehling
(AMNH), Max Langer (LPRP/USP), Daniela Schwarz (HMN), Jessica Cundiff (MCZ), Eduardo Ruigomez,
José Carballido, and Diego Pol (MPEF), Sandra Chapman and Lorna Steel (NHMUK), Erich Fitzgerald
(NMV), Caroline Buttler (NMW), Sergio Martin, Emilio Vaccari and Gabriela Cisterna (PULR), Jaime
Powell, Pablo Ortíz and Rodrigo Gonzalez (PVL), Ricardo Martínez and Diego Abelín (PVSJ), Rainer
Schoch (SMNS), Kevin Padian and Pat Holroyd (UCMP) and Michael Brett-Surman and Hans-Dieter
Sues (USNM). Access to the free version of TNT 1.1 was possible due to the Willi Henning Society.
SNFS is funded by a Swiss National Science Foundation Early Postdoc Mobility Fellowship
(P2ZHP2_195162). MDE is supported by Agencia Nacional de Promoción Científica y Técnica (PICT
2018–01186) and a Sepkoski Grant 2019 of the Paleontological Society International Research
Program.

Figure captions.

Figure 1. Holotype NHMUK PV R 37591 pelvis and vertebrae of *Pendraig orynys* gen. et sp. nov. in (A) left lateral view, (B) right lateral view. Abbreviations: atr, antitrochanter; bf, brevis fossa; bfr, brevis fossa rim; bs, brevis shelf; dv, dorsal vertebra; iss, ischial shaft; nc, neural canal; no, notch; obf, obturator foramen; poap, postacetabular process; prap, preacetabular process; puf, pubic fenestra; pus, pubic shaft; ras, rib attachment scar; ri, rim; sac, supra-acetabular crest; sv, sacral vertebra.

Figure 2. Holotype NHMUK PV R 37591 pelvis and vertebrae of *Pendraig orynys* gen. et sp. nov. in (A) dorsal view, (B) ventral view, (C) anterior view, and (D) posterior view. Abbreviations: bf, brevis fossa; bfr, brevis fossa rim; diap, diapophysis; dv, dorsal vertebra; gr, groove; il, ilium; ipis, iliac peduncle of ischium; iss, ischiadic shaft; obf, obturator foramen; poap, postacetabular process; ppdl, paradiapophyseal lamina; prap, preacetabular process; puf, pubic fenestra; pus, pubic shaft; sac, supra-acetabular crest; sv, sacral vertebra; tp, transverse process; vl, ventral lamina.

Figure 3. Holotype NHMUK PV R 37591 left femur of *Pendraig orynys* gen. et sp. nov. in (A) posteromedial, (B) anterolateral, (C) anteromedial, (D) posterolateral, (E) proximal, and (F) distal view. Abbreviations: amt, anteromedial tuber; at, anterior trochanter; icfl, depression associated with the insertion of the *M. caudofemoralis longus*; gt, greater trochanter; lica, linea intermuscularis caudalis; lincr, linea intermuscularis cranialis; obr, obturator ridge; pmt, posteromedial tuber; ts, trochanteric shelf; 4th t, fourth trochanter.

Figure 4. Isolated mid to posterior dorsal vertebra NHMUK PV 37596 of *Pendraig orynys* gen. et sp. nov. in (A) right lateral view, (B) left lateral view, (C) ventral view, (D) dorsal view, (E) anterior view, and (F) posterior view. Abbreviations: aas, anterior articular surface; acpl, anterior centroparapophyseal lamina; ce, centrum; diap, diapophysis; nf, nutrient foramen; ns, neural spine; pacdf, parapophyseal centrodiaepophyseal fossa; pacprf, parapophyseal centroprezygapophyseal fossa; pap, parapophysis; pas, posterior articular surface; pcdl, posterior centrodiaepophyseal; pocdf, postzygapophyseal centrodiaepophyseal fossa; podl, postzygodiaepophyseal lamina; poz, postzygapophysis; ppdl, paradiapophyseal lamina; prpl, prezygaparapophyseal lamina; prz, prezygapophysis; spozf, spinopostzygapophyseal fossa; sprzf, spinoprezygapophyseal fossa.

Figure 5. Isolated partial left ischium NHMUK PV R 37597 of *Pendraig orynys* gen. et sp. nov. in (A) medial and (B) dorsal view. Abbreviations: asil, articulation surface with ilium; atr, antitrochanter; ipis, iliac peduncle of ischium.

Figure 6. Strict consensus of six most parsimonious trees of the phylogenetic analysis. Bremer support, absolute bootstrap frequency, and GC bootstrap frequency values are indicated at each branch in that order.

Figure 7. \log_{10} -transformed bivariate plot of the longitudinal width of proximal head of the femur versus the femoral length of early theropods. The solid black line represents the linear regression described by the formula, and the red dotted lines represent the 95% confidence intervals.

Figure 8. Results of the ancestral state reconstruction of the \log_{10} -transformed femoral lengths. For the analysis figured here all sampled taxa were included and minimum branch length was set at 1.0 million years. The polytomy within Coelophysoidea in the strict consensus tree was manually resolved with one of the four equally parsimonious resolutions. The other analyses can be found in the Supplementary Information. The femoral lengths used for the analysis can be found in Supplementary Table 2. Reconstructed values for major nodes can be found in Table 3, and for all nodes in Supplementary Table 3.1. Node numbers used for Supplementary Table 3.1 are indicated for each node in Supplementary Figure 1.

Figure 9. Life reconstruction of *Pendraig orynys* gen. et sp. nov. amongst the fissures of Pant-y-fynnon and three individuals of the rhynchocephalian lepidosaur *Clevosaurus cambrica* during the Late Triassic. Artwork by James Robbins.

Table 1. Vertebral measurements of NHMUK PV R 37591 and NHMUK PV 37596. Measurements were taken with a Sealey electronic vernier calliper. Values preceded by a tilde (~) indicate an approximated value because a measurement was hampered, either by poor preservation, or by the relevant structure being partially covered. Abbreviations: dv, dorsal vertebra; sv, sacral vertebra.

							NHMUK PV
	dv1	dv2	sv1	sv2	sv3	sv4	37596
Centrum length	-	14.7 mm	-	13.3 mm	11.1 mm	-	14.6 mm
Neural spine height	-	-	-	-	-	-	5.0 mm
Neural spine length	-	-	-	-	-	-	14.3 mm
Width of diapophysis/transverse processes (+ rib)	-	9.9 mm	~5.39 mm	6.2 mm	6.5 mm	-	7.4 mm
Anterior articular surface centrum height	-	6.4 mm	-	5.2 mm	4.7 mm	-	5.7 mm

Posterior articular surface centrum height	□7.0 mm	6.4 mm	3.94 mm	4.7 mm	□4.2 mm	-	5.8 mm
Anteroposterior width transverse proces + rib distal end	n/a	n/a	n/a	3.7 mm	5.7 mm	n/a	n/a

Table 2. Measurements of the appendicular skeleton of NHMUK PV R 37591. Measurements were taken with a Sealey electronic vernier calliper. The circumference of the femoral shaft was measured by running a piece of string around the shaft and subsequently measuring the length of the amount of string with the calliper. Values in parentheses represent incomplete values, due to the relevant structure being incompletely preserved. Abbreviations: max, maximum.

max. length left ilium across iliac blade	55.8 mm
max. length left ilium across peduncles	26.0 mm
max. length left acetabulum	16.7 mm
max. dorsoventral height left acetabulum	17.6 mm
max. length right pubis (excluding imprint)	(63.2 mm)
max. length right pubis (including imprint)	(74.8 mm)
max. length left ischium	(26.6 mm)
max. length left femur	(86.3 mm)
max. width proximal head left femur	15.1 mm
min. circumference shaft of left femur	25.08 mm

Table 3. Ancestral state values (in cm) for major nodes for the analysis shown in Figure 8.

	Ancestral estimate	Variance	Lower 95% Confidence Interval	Upper 95% Confidence Interval
Dinosauria	15.038	1.013	10.787	20.964
Theropoda	17.679	1.008	13.542	23.080
Neotheropoda	24.168	1.013	17.331	33.703
Non-coelophysoid neotheropods	30.899	1.011	22.653	42.146
Averostra	60.474	1.011	44.309	82.536
Coelophysoidea	17.585	1.023	11.191	27.631

Coelophysoidea excluding				
Panguraptor	16.285	1.021	10.658	24.882
Coelophysidae	17.118	1.015	11.891	24.643

References.

- Whiteside D.I., Duffin C.J., Gill P.G., Marshall J.E.A., Benton M.J. 2016 The Late Triassic and Early Jurassic fissure faunas from Bristol and South Wales: stratigraphy and setting. *Palaeontologia Polonica* **67**, 257-287. (doi:10.4202/pp.2016.67_257).
- Robinson P.L. 1957 The Mesozoic fissures of the Bristol Channel area and their vertebrate faunas. *Zoological Journal of the Linnean Society* **43**(291), 260-282.
- Harris T.M. 1957 A Liasso-Rhaetic flora in south Wales. *Proceedings of the Royal Society B: Biological Sciences* **147**(928), 289-308. (doi:10.1098/rspb.1957.0051).
- Lewarne G.C., Pallot J.M. 1957 Mesozoic plants from fissures in the Carboniferous Limestone of South Wales. *Annals and Magazine of Natural History* **10**(109), 72-79. (doi:10.1080/00222935708655930).
- Marshall J.E., Whiteside D.I. 1980 Marine influence in the Triassic 'uplands'. *Nature* **287**(5783), 627-628. (doi:10.1038/287627a0).
- Whiteside D.I., Marshall J.E. 2008 The age, fauna and palaeoenvironment of the Late Triassic fissure deposits of Tytherington, South Gloucestershire, UK. *Geological Magazine* **145**(1), 105-147. (doi:10.1017/S0016756807003925).
- Whiteside D.I., Robinson D. 1983 A glauconitic clay-mineral from a speleological deposit of Late Triassic age. *Palaeogeography, Palaeoclimatology, Palaeoecology* **41**(1-2), 81-85. (doi:10.1016/0031-0182(83)90077-9).
- Mussini G., Whiteside D.I., Hildebrandt C., Benton M.J. 2020 Anatomy of a Late Triassic Bristol fissure: Tytherington fissure 2. *Proceedings of the Geologists' Association* **131**(1), 73-93. (doi:10.1016/j.pgeola.2019.12.001).
- Whiteside D.I., Benton M.J. 2021 Reply to Walkden, Fraser and Simms (2021): The age and formation mechanisms of Late Triassic fissure deposits, Gloucestershire, England: Comments on Mussini, G., Whiteside, DI, Hildebrandt C. and Benton MJ. *Proceedings of the Geologists' Association* **132**, 138-141. (doi:10.1016/j.pgeola.2020.12.001).
- Morton J.D., Whiteside D.I., Hethke M., Benton M.J. 2017 Biostratigraphy and geometric morphometrics of conchostracans (Crustacea, Branchiopoda) from the Late Triassic fissure deposits of Cromhall Quarry, UK. *Palaeontology* **60**(3), 349-374. (doi:10.1111/pala.12288).
- Walkden G., Fraser N. 1993 Late Triassic fissure sediments and vertebrate faunas: Environmental change and faunal succession at Cromhall, South West Britain. *Modern Geology* **18**, 511-535.
- Walkden G.M., Fraser N.C., Simms M.J. 2021 The age and formation mechanisms of Late Triassic fissure deposits, Gloucestershire, England: Comments on Mussini, G. et al. (2020). Anatomy of a Late Triassic Bristol fissure: Tytherington fissure 2. *Proceedings of the Geologists' Association* **132**(1), 127-137. (doi:10.1016/j.pgeola.2020.10.006).
- Evans S.E. 1980 The skull of a new eosuchian reptile from the Lower Jurassic of South Wales. *Zoological journal of the Linnean Society* **70**(3), 203-264. (doi:10.1111/j.1096-3642.1980.tb00852.x).
- Fraser N. 1982 A new rhynchocephalian from the British Upper Trias. *Palaeontology* **25**(4), 709-725.
- Gill P.G., Purnell M.A., Crumpton N., Brown K.R., Gostling N.J., Stampanoni M., Rayfield E.J. 2014 Dietary specializations and diversity in feeding ecology of the earliest stem mammals. *Nature* **512**(7514), 303. (doi:10.1038/nature13622).

16. Whiteside D.I., Duffin C.J. 2017 Late Triassic terrestrial microvertebrates from Charles Moore's 'Microlestes' quarry, Holwell, Somerset, UK. *Zoological Journal of the Linnean Society* **179**(3), 677-705. (doi:10.1111/zoj.12458).
17. Robinson P.L. 1962 Gliding lizards from the Upper Keuper of Great Britain. *Proceedings of the Geological Society of London* **160**1, 137-146.
18. Crush P.J. 1984 A late Upper Triassic sphenosuchid crocodylian from Wales. *Palaeontology* **27**(1), 131-157.
19. Skinner M., Whiteside D.I., Benton M.J. 2020 Late Triassic island dwarfs? Terrestrial tetrapods of the Ruthin fissure (South Wales, UK) including a new genus of procolophonid. *Proceedings of the Geologists' Association* **131**(5), 535-561. (doi:10.1016/j.pgeola.2020.04.005).
20. Ballell A., Rayfield E.J., Benton M.J. 2020 Osteological redescription of the Late Triassic sauropodomorph dinosaur *Thecodontosaurus antiquus* based on new material from Tytherington, southwestern England. *Journal of Vertebrate Paleontology* **40**(2), e1770774. (doi:10.1080/02724634.2020.1770774).
21. Lovegrove J., Newell A.J., Whiteside D.I., Benton M.J. 2021 Testing the relationship between marine transgression and evolving island palaeogeography using 3D GIS: an example from the Late Triassic of SW England. *Journal of the Geological Society*. (doi:10.1144/jgs2020-158).
22. Lomolino M.V., Brown J.H., Sax D.F. 2010 Island biogeography theory: reticulations and reintegration of "a biogeography of the species". In *The Theory of Island Biogeography Revisited* (eds. Losos J.B., Ricklefs R.E.), pp. 13-51. Princeton, USA, Princeton University Press.
23. Van Valen L. 1973 Pattern and the balance of nature. *Evolutionary Theory* **1**(1), 31-49.
24. Patrick E.L., Whiteside D.I., Benton M.J. 2019 A new crurotarsan archosaur from the Late Triassic of South Wales. *Journal of Vertebrate Paleontology* **39**(3), e1645147. (doi:10.1080/02724634.2019.1645147).
25. Benton M.J. 2012 Naming the Bristol dinosaur, *Thecodontosaurus*: politics and science in the 1830s. *Proceedings of the Geologists' Association* **123**(5), 766-778. (doi:10.1016/j.pgeola.2012.07.012).
26. Riley H., Stutchbury S. 1836 On an additional species of the newly-discovered saurian animals in the Magnesian Conglomerate of Durdham Down, near Bristol. *Annual Report of the British Association for the Advancement of Science, Transactions of the Sections*, 90-94.
27. Benton M.J., Juul L., Storrs G.W., Galton P.M. 2000 Anatomy and systematics of the prosauropod dinosaur *Thecodontosaurus antiquus* from the Upper Triassic of southwest England. *Journal of Vertebrate Paleontology* **20**(1), 77-108. (doi:10.1671/0272-4634(2000)020[0077:AASOTP]2.0.CO;2).
28. Galton P., Kermack D. 2010 The anatomy of *Pantydraco caducus*, a very basal sauropodomorph dinosaur from the Rhaetian (Upper Triassic) of South Wales, UK. *Revue de Paléobiologie* **29**(2), 341-404.
29. Yates A.M. 2003 A new species of the primitive dinosaur *Thecodontosaurus* (Saurischia: Sauropodomorpha) and its implications for the systematics of early dinosaurs. *Journal of Systematic Palaeontology* **1**(1), 1-42. (doi:10.1017/S1477201903001007).
30. Kermack D. 1984 New prosauropod material from South Wales. *Zoological Journal of the Linnean Society* **82**(1-2), 101-117. (doi:10.1111/j.1096-3642.1984.tb00538.x).
31. Galton P.M., Yates A.M., Kermack D. 2007 *Pantydraco* n. gen. for *Thecodontosaurus caducus* Yates, 2003, a basal sauropodomorph dinosaur from the Upper Triassic or Lower Jurassic of South Wales, UK. *Neues Jahrbuch für Geologie und Paläontologie-Abhandlungen* **243**(1), 119-125. (doi:10.1127/0077-7749/2007/0243-0119).
32. Fraser N.C., Padian K., Walkden G.M., Davis A. 2002 Basal dinosauriform remains from Britain and the diagnosis of the Dinosauria. *Palaeontology* **45**(1), 79-95. (doi:10.1111/1475-4983.00228).
33. Baron M.G., Norman D.B., Barrett P.M. 2017 A new hypothesis of dinosaur relationships and early dinosaur evolution. *Nature* **543**(7646), 501-506. (doi:10.1038/nature21700).

34. Bonaparte J., Brea G., Schultz C., Martinelli A. 2007 A new specimen of *Guaibasaurus candelariensis* (basal Saurischia) from the Late Triassic Caturrita Formation of southern Brazil. *Historical Biology* **19**(1), 73-82. (doi:10.1080/08912960600866862).
35. Ezcurra M.D. 2010 A new early dinosaur (Saurischia: Sauropodomorpha) from the Late Triassic of Argentina: a reassessment of dinosaur origin and phylogeny. *Journal of Systematic Palaeontology* **8**(3), 371-425. (doi:10.1080/14772019.2010.484650).
36. Langer M.C. 2004 Basal Saurischia. In *The Dinosauria* (eds. Weishampel D.B., Dodson P., Osmólska H.), pp. 25-46. Berkeley, University of California Press.
37. Langer M.C., Nesbitt S.J., Bittencourt J.S., Irmis R.B. 2013 Non-dinosaurian Dinosauromorpha. *Geological Society, London, Special Publications* **379**(1), 157-186. (doi:10.1144/SP379.9).
38. Langer M.C., Ezcurra M.D., Rauhut O.W., Benton M.J., Knoll F., McPhee B.W., Novas F.E., Pol D., Brusatte S.L. 2017 Untangling the dinosaur family tree. *Nature* **551**(7678), E1-E3. (doi:10.1038/nature24011).
39. Baron M.G., Norman D.B., Barrett P.M. 2017 Baron et al. reply. *Nature* **551**(7678), E4-E5. (doi:10.1038/nature24012).
40. Lucas S.G., Heckert A.B., Fraser N.C., Huber P. 1999 *Aetosaurus* from the Upper Triassic of Great Britain and its biochronological significance. *Neues Jahrbuch für Geologie und Paläontologie-Monatshefte* **9**, 568-576. (doi:10.1127/njgpm/1999/1999/568).
41. Warrener D. 1983 An archosaurian fauna from a Welsh locality, University College London (University of London).
42. Rauhut O.W., Hungerbühler A. 2000 A review of European Triassic theropods. *Gaia* **15**, 75-88.
43. Keeble E., Whiteside D.I., Benton M.J. 2018 The terrestrial fauna of the Late Triassic Pant-y-ffynnon Quarry fissures, South Wales, UK and a new species of *Clevosaurus* (Lepidosauria: Rhynchocephalia). *Proceedings of the Geologists' Association* **129**(2), 99-119. (doi:10.1016/j.pgeola.2017.11.001).
44. Cope E. 1869-1870 Synopsis of the extinct Batrachia, Reptilia and Aves of North America. *Transactions of the American Philosophical Society* **14**(1). (doi:10.5962/bhl.title.60499).
45. Gauthier J.A., Padian K. 2020 Archosauria. In *Phylonyms: A Companion to the PhyloCode* (eds. de Queiroz K., Cantino P.D., Gauthier J.A.), pp. 1187-1193. Boca Raton, FL, CRC Press.
46. Owen R. 1842 Report on British fossil reptiles, part II. *Report for the British Association for the Advancement of Science, Plymouth* **11**, 60-294.
47. Langer M.C., Novas F.E., Bittencourt J.S., Ezcurra M.D., Gauthier J.A. 2020 Dinosauria. In *Phylonyms: A Companion to the PhyloCode* (eds. de Queiroz K., Cantino P.D., Gauthier J.A.), pp. 1209-1217. Boca Raton, FL, CRC Press.
48. Marsh O.C. 1881 Principal characters of American Jurassic dinosaurs. *American Journal of Science (3rd series)* **21**, 417-423.
49. Naish D., Cau A., Holtz T.R., Fabbri M., Gauthier J.A. 2020 Theropoda. In *Phylonyms: A Companion to the PhyloCode* (eds. de Queiroz K., Cantino P.D., Gauthier J.A.), pp. 1235-1246. Boca Raton, FL, CRC Press.
50. Bakker R.T. 1986 *The Dinosaur Heresies*, William Morrow.
51. Sereno P.C. 1998 A rationale for phylogenetic definitions, with application to the higher-level taxonomy of Dinosauria. *Neues Jahrbuch für Geologie und Paläontologie-Abhandlungen* **210**(1), 41-83. (doi:10.1127/njgpa/210/1998/41).
52. Nopcsa F.B. 1928 The genera of reptiles. *Palaeobiologica* **1**, 168-188.
53. Griffin C. 2018 Developmental patterns and variation among early theropods. *Journal of Anatomy* **232**(4), 604-640. (doi:10.1111/joa.12775).
54. Griffin C., Nesbitt S.J. 2016 The femoral ontogeny and long bone histology of the Middle Triassic (?late Anisian) dinosauriform *Asilisaurus kongwe* and implications for the growth of early

- dinosaurs. *Journal of Vertebrate Paleontology* **36**(3), e1111224. (doi:10.1080/02724634.2016.1111224).
55. Griffin C.T., Nesbitt S.J. 2016 Anomalously high variation in postnatal development is ancestral for dinosaurs but lost in birds. *Proceedings of the National Academy of Sciences* **113**(51), 14757-14762. (doi:10.1073/pnas.1613813113).
56. Griffin C.T., Nesbitt S.J. 2020 Does the maximum body size of theropods increase across the Triassic–Jurassic boundary? Integrating ontogeny, phylogeny, and body size. *The Anatomical Record* **303**(4), 1158-1169. (doi:10.1002/ar.24130).
57. Ezcurra M.D., Butler R.J., Maidment S.C., Sansom I.J., Meade L.E., Radley J.D. 2021 A revision of the early neotheropod genus *Sarcosaurus* from the Early Jurassic (Hettangian–Sinemurian) of central England. *Zoological Journal of the Linnean Society* **191**(1), 113-149. (doi:10.1093/zoolinnean/zlaa054).
58. Martínez R.N., Sereno P.C., Alcober O.A., Colombi C.E., Renne P.R., Montañez I.P., Currie B.S. 2011 A basal dinosaur from the dawn of the dinosaur era in southwestern Pangaea. *Science* **331**(6014), 206-210. (doi:10.1126/science.1198467).
59. Wilson J.A. 1999 A nomenclature for vertebral laminae in sauropods and other saurischian dinosaurs. *Journal of vertebrate Paleontology* **19**(4), 639-653. (doi:10.1080/02724634.1999.10011178).
60. Raath M.A. 1977 The anatomy of the Triassic theropod *Syntarsus rhodesiensis* (Saurischia: Podokesauridae) and a consideration of its biology. Salisbury, Rhodesia, Rhodes University.
61. Marsh A.D., Rowe T.B. 2020 A comprehensive anatomical and phylogenetic evaluation of *Dilophosaurus wetherilli* (Dinosauria, Theropoda) with descriptions of new specimens from the Kayenta Formation of northern Arizona. *Journal of Paleontology* **94**(S78), 1-103. (doi:10.1017/jpa.2020.14).
62. Rinehart L.F., Lucas S.G., Heckert A.B., Spielmann J.A., Celleskey M.D. 2009 The Paleobiology of *Coelophysis bauri* (Cope) from the Upper Triassic (Apachean) Whitaker quarry, New Mexico, with detailed analysis of a single quarry block. *New Mexico Museum of Natural History and Science Bulletin* **45**, 1-260.
63. You H.-L., Azuma Y., Wang T., Wang Y.-M., Dong Z.-M. 2014 The first well-preserved coelophysoid theropod dinosaur from Asia. *Zootaxa* **3873**(3), 233-249. (doi:10.11646/zootaxa.3873.3.3).
64. Cuny G., Galton P.M. 1993 Revision of the Airel theropod dinosaur from the Triassic-Jurassic boundary (Normandy, France). *Neues Jahrbuch für Geologie und Paläontologie Abhandlungen* **187**(3), 261-288.
65. Carpenter K. 1997 A giant coelophysoid (Ceratosauria) theropod from the Upper Triassic of New Mexico, USA. *Neues Jahrbuch für Geologie und Paläontologie-Abhandlungen* **205**(2), 189-208. (doi:10.1127/njgpa/205/1997/189).
66. Sereno P.C. 1999 The evolution of dinosaurs. *Science* **284**(5423), 2137-2147. (doi:10.1126/science.284.5423.2137).
67. Smith N.D., Makovicky P.J., Hammer W.R., Currie P.J. 2007 Osteology of *Cryolophosaurus ellioti* (Dinosauria: Theropoda) from the Early Jurassic of Antarctica and implications for early theropod evolution. *Zoological Journal of the Linnean Society* **151**(2), 377-421. (doi:10.1111/j.1096-3642.2007.00325.x).
68. Wilson J.A., Michael D., Ikejiri T., Moacdieh E.M., Whitlock J.A. 2011 A nomenclature for vertebral fossae in sauropods and other saurischian dinosaurs. *PLoS One* **6**(2), e17114. (doi:10.1371/journal.pone.0017114).
69. Colbert E.H. 1989 The Triassic dinosaur *Coelophysis*. *Museum of Northern Arizona, Bulletin* **57**, 1-160.
70. Nesbitt S.J. 2011 The early evolution of archosaurs: relationships and the origin of major clades. *Bulletin of the American Museum of Natural History* **352**, 1-292. (doi:10.1206/352.1).

- 71. Raath M.A. 1990 Morphological variation in small theropods and its meaning in systematics: evidence from *Syntarsus*. In *Dinosaur Systematics: Perspectives and Approaches* (eds. Carpenter K., Currie P.J.), pp. 91-105. Cambridge, Cambridge University Press.
- 72. Tykoski R.S. 2005 Anatomy, ontogeny, and phylogeny of coelophysoid theropods.
- 73. Zahner M., Brinkmann W. 2019 A Triassic averostran-line theropod from Switzerland and the early evolution of dinosaurs. *Nature ecology & evolution* **3**(8), 1146-1152. (doi:10.1038/s41559-019-0941-z).
- 74. Hutchinson J.R. 2001 The evolution of pelvic osteology and soft tissues on the line to extant birds (Neornithes). *Zoological Journal of the Linnean Society* **131**(2), 123-168. (doi:10.1111/j.1096-3642.2001.tb01313.x).
- 75. Rowe T. 1989 A new species of the theropod dinosaur *Syntarsus* from the Early Jurassic Kayenta Formation of Arizona. *Journal of Vertebrate Paleontology* **9**(2), 125-136. (doi:10.1080/02724634.1989.10011748).
- 76. Rauhut O.W.M. 2003 The interrelationships and evolution of basal theropods (Dinosauria, Saurischia). *Special Papers in Palaeontology* **69**, 1-213.
- 77. Ezcurra M.D., Cuny G. 2007 The coelophysoid *Lophostropheus airelensis*, gen. nov.: a review of the systematics of "*Liliensternus*" *airelensis* from the Triassic–Jurassic outcrops of Normandy (France). *Journal of Vertebrate Paleontology* **27**(1), 73-86. (doi:10.1671/0272-4634(2007)27[73:TCLAGN]2.0.CO;2).
- 78. Tykoski R.S. 1998 The osteology of *Syntarsus kayentakatae* and its implications for ceratosaurid phylogeny. Austin, University of Texas at Austin.
- 79. Ezcurra M.D. 2016 The phylogenetic relationships of basal archosauromorphs, with an emphasis on the systematics of proterosuchian archosauriforms. *PeerJ* **4**, e1778. (doi:10.7717/peerj.1778).
- 80. Camp C.L. 1936 A new type of small bipedal dinosaur from the Navajo Sandstone of Arizona. *Bulletin of the Department of Geological Sciences* **24**, 39-56.
- 81. Carrano M.T., Hutchinson J.R., Sampson S.D. 2005 New information on *Segisaurus halli*, a small theropod dinosaur from the Early Jurassic of Arizona. *Journal of Vertebrate Paleontology* **25**(4), 835-849. (doi:10.1671/0272-4634(2005)025[0835:NIOSHA]2.0.CO;2).
- 82. Galton P., Jensen J.A. 1979 A new large theropod dinosaur from the Upper Jurassic of Colorado. *Brigham Young University Geology Studies* **26**(2), 1-12.
- 83. Martill D.M., Vidovic S.U., Howells C., Nudds J.R. 2016 The oldest Jurassic dinosaur: a basal neotheropod from the Hettangian of Great Britain. *PLoS One* **11**(1), e0145713. (doi:10.1371/journal.pone.0145713).
- 84. Langer M.C., Rincón A.D., Ramezani J., Solórzano A., Rauhut O.W. 2014 New dinosaur (Theropoda, stem-Averostra) from the earliest Jurassic of the La Quinta formation, Venezuelan Andes. *Royal Society Open Science* **1**(2), 140184. (doi:10.1098/rsos.140184).
- 85. Hutchinson J.R. 2001 The evolution of femoral osteology and soft tissues on the line to extant birds (Neornithes). *Zoological Journal of the Linnean Society* **131**(2), 169-197. (doi:10.1111/j.1096-3642.2001.tb01314.x).
- 86. Langer M.C., Ezcurra M.D., Bittencourt J.S., Novas F.E. 2010 The origin and early evolution of dinosaurs. *Biological Reviews* **85**(1), 55-110. (doi:10.1111/j.1469-185X.2009.00094.x).
- 87. Novas F.E. 1996 Dinosaur monophyly. *Journal of Vertebrate Paleontology* **16**(4), 723-741. (doi:10.1080/02724634.1996.10011361).
- 88. Nesbitt S.J., Smith N.D., Irmis R.B., Turner A.H., Downs A., Norell M.A. 2009 A complete skeleton of a Late Triassic saurischian and the early evolution of dinosaurs. *Science* **326**(5959), 1530-1533. (doi:10.1126/science.1180350).
- 89. Novas F.E. 1994 New information on the systematics and postcranial skeleton of *Herrerasaurus ischigualastensis* (Theropoda: Herrerasauridae) from the Ischigualasto Formation (Upper Triassic) of Argentina. *Journal of Vertebrate Paleontology* **13**(4), 400-423. (doi:10.1080/02724634.1994.10011523).

90. Sereno P.C., Martínez R.N., Alcober O.A. 2013 Osteology of *Eoraptor lunensis* (Dinosauria, Sauropodomorpha). *Journal of Vertebrate Paleontology* **32**(sup1), 83-179. (doi:10.1080/02724634.2013.820113).
91. Novas F.E., Agnolin F., Ezcurra M.D., Müller R., Martinelli A., Langer M.C. in press Review of the fossil record of early dinosaurs from South America, and its phylogenetic implications. *Journal of South American Earth Sciences*.
92. Ezcurra M.D., Brusatte S.L. 2011 Taxonomic and phylogenetic reassessment of the early neotheropod dinosaur *Camposaurus arizonensis* from the Late Triassic of North America. *Palaeontology* **54**(4), 763-772. (doi:10.1111/j.1475-4983.2011.01069.x).
93. Sues H.-D., Nesbitt S.J., Berman D.S., Henrici A.C. 2011 A late-surviving basal theropod dinosaur from the latest Triassic of North America. *Proceedings of the Royal Society B: Biological Sciences* **278**(1723), 3459-3464. (doi:10.1098/rspb.2011.0410).
94. Ezcurra M.D. 2017 A new early coelophysoid neotheropod from the Late Triassic of northwestern Argentina. *Ameghiniana* **54**(5), 506-538. (doi:10.5710/AMGH.04.08.2017.3100).
95. Marsh A.D., Parker W.G., Langer M.C., Nesbitt S.J. 2019 Redescription of the holotype specimen of *Chindesaurus bryansmalli* Long and Murry, 1995 (Dinosauria, Theropoda), from Petrified Forest National Park, Arizona. *Journal of Vertebrate Paleontology* **39**(3), e1645682. (doi:10.1080/02724634.2019.1645682).
96. Marsola J.C., Bittencourt J.S., Butler R.J., Da Rosa Á.A., Sayão J.M., Langer M.C. 2018 A new dinosaur with theropod affinities from the Late Triassic Santa Maria Formation, South Brazil. *Journal of Vertebrate Paleontology* **38**(5), e1531878. (doi:10.1080/02724634.2018.1531878).
97. Martínez R.N., Apaldetti C. 2017 A Late Norian—Rhaetian Coelophysid Neotheropod (Dinosauria, Saurischia) from the Quebrada Del Barro Formation, Northwestern Argentina. *Ameghiniana* **54**(5), 488-505. (doi:10.5710/AMGH.09.04.2017.3065).
98. Nesbitt S.J., Ezcurra M.D. 2015 The early fossil record of dinosaurs in North America: a new neotheropod from the base of the Upper Triassic Dockum Group of Texas. *Acta Palaeontologica Polonica* **60**(3), 513-526. (doi:10.4202/app.00143.2014).
99. Griffin C.T. 2019 Large neotheropods from the Upper Triassic of North America and the early evolution of large theropod body sizes. *Journal of Paleontology* **93**(5), 1010-1030. (doi:10.1017/jpa.2019.13).
100. Ezcurra M.D., Nesbitt S.J., Bronzati M., Dalla Vecchia F.M., Agnolin F.L., Benson R.B., Egli F.B., Cabreira S.F., Evers S.W., Gentil A.R. 2020 Enigmatic dinosaur precursors bridge the gap to the origin of Pterosauria. *Nature* **588**, 445-449. (doi:10.1038/s41586-020-3011-4).
101. Goloboff P.A., Catalano S.A. 2016 TNT version 1.5, including a full implementation of phylogenetic morphometrics. *Cladistics* **32**(3), 221-238. (doi:10.1111/cla.12160).
102. Campione N.E., Evans D.C. 2012 A universal scaling relationship between body mass and proximal limb bone dimensions in quadrupedal terrestrial tetrapods. *Bmc Biology* **10**(1), 1-22. (doi:10.1186/1741-7007-10-60).
103. Team R.C. 2020 R: A language and environment for statistical computing. (ed. Computing R.F.f.S.). Vienna, Austria, <https://www.R-project.org/>.
104. Cohen K.M., Harper D.A.T., Gibbard P.L. 2020 ICS International Chronostratigraphic Chart 2020/03. (International Commission on Stratigraphy, IUGS).
105. Bapst D.W. 2012 paleotree: an R package for paleontological and phylogenetic analyses of evolution. *Methods in Ecology and Evolution* **3**(5), 803-807. (doi:10.1111/j.2041-210X.2012.00223.x).
106. Revell L.J. 2012 phytools: an R package for phylogenetic comparative biology (and other things). *Methods in ecology and evolution* **3**(2), 217-223. (doi:10.1111/j.2041-210X.2011.00169.x).
107. Campione N.E., Brink K.S., Freedman E.A., McGarrity C.T., Evans D.C. 2013 '*Glishades ericksoni*', an indeterminate juvenile hadrosaurid from the Two Medicine Formation of Montana: implications for hadrosauroid diversity in the latest Cretaceous (Campanian-Maastrichtian) of western North America. *Palaeobiodiversity and Palaeoenvironments* **93**(1), 65-75. (doi:10.1007/s12549-012-0097-1).

108. Tsuihiji T., Watabe M., Tsogtbaatar K., Tsubamoto T., Barsbold R., Suzuki S., Lee A.H., Ridgely
R.C., Kawahara Y., Witmer L.M. 2011 Cranial osteology of a juvenile specimen of *Tarbosaurus bataar*
(Theropoda, Tyrannosauridae) from the Nemegt Formation (Upper Cretaceous) of Bugin Tsav,
Mongolia. *Journal of Vertebrate Paleontology* **31**(3), 497-517. (doi:10.1080/02724634.2011.557116).
109. Wang S., Stiegler J., Amiot R., Wang X., Du G.-h., Clark J.M., Xu X. 2017 Extreme ontogenetic
changes in a ceratosaurian theropod. *Current Biology* **27**(1), 144-148.
(doi:10.1016/j.cub.2016.10.043).
110. Irmis R.B. 2010 Evaluating hypotheses for the early diversification of dinosaurs. *Earth and*
*Environmental Science Transactions of the Royal Society of Edinburgh* **101**(3-4), 397-426.
(doi:10.1017/S1755691011020068).
111. Benson R.B., Campione N.E., Carrano M.T., Mannion P.D., Sullivan C., Upchurch P., Evans
D.C. 2014 Rates of dinosaur body mass evolution indicate 170 million years of sustained ecological
innovation on the avian stem lineage. *PLoS Biology* **12**(5), e1001853.
(doi:10.1371/journal.pbio.1001853).
112. Lee M.S., Cau A., Naish D., Dyke G.J. 2014 Sustained miniaturization and anatomical
innovation in the dinosaurian ancestors of birds. *Science* **345**(6196), 562-566.
(doi:10.1126/science.1252243).

Figure 1. Holotype NHMUK PV R 37591 pelvis and vertebrae of *Pendraig orynys* gen. et sp. nov. in (A) left lateral view, (B) right lateral view. Abbreviations: atr, antitrochanter; bf, brevis fossa; bfr, brevis fossa rim; bs, brevis shelf; dv, dorsal vertebra; iss, ischial shaft; nc, neural canal; no, notch; obf, obturator foramen; poap, postacetabular process; prap, preacetabular process; puf, pubic fenestra; pus, pubic shaft; ras, rib attachment scar; ri, rib; sac, supra-acetabular crest; sv, sacral vertebra.

Figure 2. Holotype NHMUK PV R 37591 pelvis and vertebrae of *Pendraig orynys* gen. et sp. nov. in (A) dorsal view, (B) ventral view, (C) anterior view, and (D) posterior view. Abbreviations: bf, brevis fossa; bfr, brevis fossa rim; diap, diapophysis; dv, dorsal vertebra; gr, groove; il, ilium; ipis, iliac peduncle of ischium; iss, ischiadic shaft; obf, obturator foramen; poap, postacetabular process; ppdl, paradiapophyseal lamina; prap, preacetabular process; puf, pubic fenestra; pus, pubic shaft; sac, supra-acetabular crest; sv, sacral vertebra; tp, transverse process; vl, ventral lamina.

Figure 3. Holotype NHMUK PV R 37591 left femur of *Pendraig orynys* gen. et sp. nov. in (A) posteromedial, (B) anterolateral, (C) anteromedial, (D) posterolateral, (E) proximal, and (F) distal view. Abbreviations: amt, anteromedial tuber; at, anterior trochanter; icfl, depression associated with the insertion of the *M. caudofemoralis longus*; gt, greater trochanter; lica, linea intermuscularis caudalis; lincr, linea intermuscularis cranialis; obr, obturator ridge; pmt, posteromedial tuber; ts, trochanteric shelf; 4th t, fourth trochanter.

Figure 4. Isolated mid to posterior dorsal vertebra NHMUK PV 37596 of *Pendraig orynys* gen. et sp. nov. in (A) right lateral view, (B) left lateral view, (C) ventral view, (D) dorsal view, (E) anterior view, and (F) posterior view. Abbreviations: aas, anterior articular surface; acpl, anterior centroparapophyseal lamina; ce, centrum; diap, diapophysis; nf, nutrient foramen; ns, neural spine; pacdf, paradiapophyseal centrodiapophyseal fossa; pacprf, parapophyseal centroprezygapophyseal fossa; pap, parapophysis; pas, posterior articular surface; pcdl, posterior centrodiapophyseal; pocdf, postzygapophyseal centrodiapophyseal fossa; podl, postzygodiapophyseal lamina; poz, postzygapophysis; ppdl, paradiapophyseal lamina; prpl, prezygaparapophyseal lamina; prz, prezygapophysis; spozf, spinopostzygapophyseal fossa; sprzf, spinoprezygapophyseal fossa.

Figure 5. Isolated partial left ischium NHMUK PV R 37597 of *Pendraig orynys* gen. et sp. nov. in (A) medial and (B) dorsal view. Abbreviations: asil, articulation surface with ilium; atr, antitrochanter; ipis, iliac peduncle of ischium.

- Taxon**
- *Coelophysidae indet.*
 - *Coelophysis bauri*
 - *Dilophosaurus wetherilli*
 - *Eodromaeus murphi*
 - *Liliensternus liliensterni*
 - *Megapnosaurus rhodesiensis*
 - *'Syntarsus' kayentakatae*
 - *Pendraig orynys (estimated)*

$y = 0.845x + 1.013$
 $r^2 = 0.968, p < 2.2 \times 10^{-16}$

<http://manuscriptcentral.com/rso>

Figure 9. Life reconstruction of *Pendraig orynys* gen. et sp. nov. amongst the fissures of Pant-y-ffynnon and three individuals of the rhynchocephalian lepidosaur *Clevosaurus cambrica* during the Late Triassic. Artwork by James Robbins.

322x239mm (300 x 300 DPI)

Appendix D

Dear Drs. Botha and Padian,

Please find our response to your recommendations and those of the reviewers below indicated in bold and in the annotated PDFs, which we have uploaded separately.

Kind regards, also on behalf of my co-authors,

Dr. Stephan Spiekman

Dear Dr Spiekman

The Editors assigned to your paper RSOS-210915 "Pendraig orynys, a new small-sized coelophysoid theropod from the Late Triassic of Wales" have now received comments from reviewers and would like you to revise the paper in accordance with the reviewer comments and any comments from the Editors. Please note this decision does not guarantee eventual acceptance.

Please submit your revised manuscript and required files (see below) no later than 21 days from today's (ie 16-Jul-2021) date. Note: the ScholarOne system will 'lock' if submission of the revision is attempted 21 or more days after the deadline. If you do not think you will be able to meet this deadline please contact the editorial office immediately.

on behalf of Dr Jennifer Botha (Associate Editor) and Kevin Padian (Subject Editor)

Associate Editor Comments to Author (Dr Jennifer Botha):

Associate Editor: 1

Comments to the Author:

This is a good paper, well worth publishing. However, there are some corrections necessary before the paper can be accepted for publication. Two important points worth noting specifically (1) one of the reviewers tested the code and was unable to replicate the results - the code needs to be checked for errors and (2) the interpretation that this individual is a small taxon as opposed to being a juvenile requires further evidence, and reviewer one makes a good case for thin sectioning part of the femur to check for the presence of an EFS. Otherwise there is simply not enough evidence for small size. I encourage the authors to look seriously into this suggestion.

We have responded to these issues below where they were raised by the reviewers.

Reviewer comments to Author:

Reviewer: 1

Comments to the Author(s)

This manuscript presents a thorough comparative description of a new theropod dinosaur and places it into a phylogenetic hypothesis using a relevant and recent character matrix. The small size of the holotype individual is striking, so the authors also conduct an ancestral state reconstruction for body size among early theropod dinosaurs and also attempt to take the individual's ontogenetic status into account to be sure that the small size does not simply stem from the individual's immaturity.

The manuscript is well-written and thorough, the comparative descriptions sound, the figures are clear and informative, and the analyses all appear to be properly conducted (but see below for problems with the R code). In my opinion the manuscript is largely sound and can be accepted for publication with moderate revision. I have attached a PDF with my minor edits and comments.

Major Comments

1) Why are the eleven supplementary figures provided as separate files, with yet another Word file for the captions? It would be much easier to include all figures with associated captions as one PDF file. Also, in Table S2 in the "Juvenile" column, there are several different terms used for maturity assessment, including juvenile, non-juvenile, subadult, immature, etc. Sometimes these seem to be synonymous, but their usage at other times appears to be mutually exclusive. For example, some taxa are listed as "SUBADULT/IMMATURE" and others are just listed as "IMMATURE". Could you provide more explanation for these maturity categories?

We now provide a single PDF file that includes all Supplementary Figures and their corresponding captions. The dataset provided in Table S2 is part of a large dataset compiled for a separate study.

For the purposes of this study, we have now modified all identifiers of a non-mature state to “immature”.

2) When running the R code, I experienced an error code that prevented me from continuing to evaluate the rest of the code. The error occurred on line 42, on the `anc.ML()` command, and read: “Error in `optim(c(sig2, a, y, rep(mean(x), length(xx))), fn = likelihood, : non-finite value supplied by optim”.`

All the underlying data looked sound to me; I saw no obvious issues with the way the data was input, the tree file, etc. It may be the issue is with my version of R or R Studio, because I just updated both two days ago, but this affects repeatability and is something the authors should be aware of.

We ran the code on three different machines and in all cases we did not encounter any error. As the reviewer stated, the errors he experienced are most likely attributable to having an older version of R and Rstudio installed. Under the latest versions of R (4.1.0) and RStudio (1.4.1717) the code should work fine.

3) I appreciate that the authors take the body size of the type individual into account when evaluating possible small body size. This is done in a clear way using character state transformations that have been useful in other early theropods, particularly *Coelophysis*. I agree with the authors' assessment that this individual does not display the features we might expect of either a very skeletally mature or immature individual, and the character states instead suggest that this individual is in a middling 'gray zone' of ontogeny. Indeed, there is no reasonably complete specimen of *Coelophysis* that has been scored with a consistent combination of character states (data from Griffin & Nesbitt 2016, Griffin 2018), making direct comparison with *Coelophysis* character states and femoral lengths difficult. This, combined with the fact that their scoring of *Panguraptor* ontogenetic character state combinations that contradict any reconstructed ontogenetic sequence of *Coelophysis* (very interesting finding, by the way), strongly suggests that there is even more variation in ontogenetic trajectories among *Coelophysoids* than has presently been reconstructed. Because of the large amount of ontogenetic variation known from other *coelophysoids*, combined with the fact that most of the size variation in *Coelophysis* is known precisely from this ontogenetic gray zone, I disagree with the authors' statement that this individual was likely near maximum body size and would not have gotten much larger. Instead, I think that there is not enough evidence to say one way or another.

Although there is no direct comparison between the character states of *Pendraig* and *Coelophysis* or *Megapnosaurus*, some examples from individuals roughly the size of *Pendraig* that also display similarly mature character states may be informative:

--The smallest known individual of *Megapnosaurus* (NHMZ QG 45) possesses fused sacral neural spines (character 1-1; and therefore probably at least some fused sacral centra, although these are not able to be scored) and a small trochanteric shelf (14-1; 15-0), despite its extremely small size (femoral head 1.5 cm; reconstructed femoral length ~11.2 cm).
--*Coelophysis bauri* (TMP 1984.063.0001 #13; reconst. femur length ~10.9 cm) has fused its pubis to the ilium (8-1) and ischium (10-1), but the ilium and ischium remain unfused (9-1).
-- *Coelophysis bauri* (SMP 858; femoral head 1.29 cm, reconst. femur length ~10.2 cm) possesses five fused sacrals (2-2), a fused pubis (8-1), a trochanteric shelf (14-1), a mound-like dorsolateral trochanter (16-1), a cranial intermuscular line (17-1), a caudal intermuscular

line (18-1), and an 'anterolateral scar' (19-1).

-- *Coelophysis bauri* (CMNH 10971 #3; femur length 10.94 cm) possesses five fused sacrals (2-2), a fused pubis (8-1), a trochanteric shelf (14-1), a mound-like dorsolateral trochanter (16-1), a cranial intermuscular line (17-1), a caudal intermuscular line (18-1), and an 'anterolateral scar' (19-1).

There are of course smaller individuals of *Coelophysis* that display more immature character states as well. But my point here is that finding an individual of *Coelophysis* that displays a similar body size and similar ontogenetic character states is not unprecedented. Because we know that *Coelophysis* can reach much larger sizes (~25 cm femoral length), then with a sample size of 1, this is not great evidence that *Pendraig's* maximum size is close to what this individual's is; this taxon could just happen to be represented by one of those more mature-looking but anomalously small individuals also found in other coelophysoids.

Therefore, although the data are consistent with *Pendraig* having a small maximum body size, I do not think the data support saying that it did definitively have a small maximum body size.

We appreciate the reviewer's comments on this subject as we based our assessment of the ontogenetic stage of *Pendraig* on the maturity matrix he developed. We have modified this part of the discussion according to these comments. We now no longer claim that the analysis based on the maturity matrix shows that *Pendraig* was smaller than *Coelophysis* and *Megapnosaurus*. But since the reviewer also indicates that specimens of *Coelophysis* and *Megapnosaurus* with a similar size and degree of maturity as the holotype of *Pendraig* represent exceptions rather than the rule (i.e., anomalies), we do point out that the analysis at least hints at a smaller body size for *Pendraig*. We additionally point out, based on the reviewer's comments, that the absence of a hyposphene-hypantrum articulation in the dorsal vertebrae provides an alternative line of evidence that suggests that *Pendraig* was smaller than the aforementioned taxa. This articulation occurs in all known theropod taxa with a femoral length >170 mm and its clear absence in *Pendraig* thus suggests that this taxon very likely had a smaller femur length than this threshold value and was therefore considerably smaller than the largest known specimens of *Coelophysis* and *Megapnosaurus*.

4) This brings me to my suggestion that the best way to resolve this issue of maturity and body size is by histologically sampling the individual. I understand that the authors probably have qualms about destructively sampling a holotype and only known specimen (barring the referred vertebra) of a taxon, but I do have two ideas that may alleviate some concerns. I notice in Figure 3 that the femur is already somewhat damaged at midshaft. I suggest that, instead of sampling the entire cross-section of the femoral shaft, you break off a small piece of already-damaged cortex from this midshaft region and histologically sample that (see picture in attached file to see what I mean). I have successfully used this technique when I did not want to damage the full specimen but just sampled a portion that was already damaged, including on a femur of *Dromomeron romeri* (Griffin et al. 2019, PeerJ), and on a femur of *Coelophysis* (CMNH 10971; Barta et al. in prep). Although a full cortical sample is ideal, with a partial cortex you can still see, for example, LAGs, LAG spacing, an EFS if present, etc. I used this method to find an EFS in the *Coelophysis* individual referenced above.

Another option, and slightly more heterodox, would be attempting to sample the distalmost preserved end of the ischium, which is roughly midshaft. Pelvic elements are not often sampled, but in my experience any long, somewhat tubular endochondral bone preserves a record of growth and can be useful for histological maturity assessment when sampled near

midshaft. I have seen this work on an immature Tyrannosaurid pubis (pers. obs.), metatarsals (McLain et al. 2018, Palaios), and hyoid elements (pers. obs., submitted to be a 2021 SVP poster). A 2011 Master's thesis showed that the midshaft of the Alligator pubis is skeletochronologically informative, and the ischium did not work only because it is platelike, not elongate, which is not an issue for Pendraig (Garcia 2011, "Skeletochronology of the American Alligator (*Alligator Mississippiensis*): Examination of the Utility of Elements for Histological Study", Florida State University). The ischium of Pendraig is broken at roughly midshaft, so only ~1 cm of the ischium would need to be removed to make a histological section, causing minimal damage to the specimen. In fact, a combination of sampling from a femoral fragment and the ischium would be a good multi-elemental way to assess maturity and back up the assessment made by morphology, making the assessment supported by multiple lines of evidence.

I do not hinge my final approval on whether histological sampling is conducted on this specimen, and I think the authors can incorporate the concerns from my comment #3 without including histology. I know that it is not always possible or desirable to destructively sample a holotype. However, I do think that it would make the paper stronger, more convincing, and more citable, and therefore I recommend the authors add this to their study if it is possible.

Destructive sampling of this specimen (whether taking a full section or breaking parts off) has been declined by the NHM. The specimen is the holotype and only known specimen of this genus and species. The NHM's destructive sampling procedures indicate that, in general, destructive sampling of holotypes should not be carried out unless there is a very realistic likelihood that the results will significantly improve our knowledge and understanding of the specimen, and we do not think that is likely here. Previous attempts have been made to histologically sample several specimens from Pant-y-ffynnon quarry with poor results which have precluded publication of these studies. It is not unlikely, therefore, that we could cut a thin section from this holotype and it would reveal that taphonomic effects have destroyed or overprinted the bone fabric, precluding us from obtaining the ontogenetic information we seek. Furthermore, histological sectioning is unlikely to actually tell us more than we have already determined from a skeletal assessment of maturity. The histological characteristic usually used to indicate maturity in dinosaurs, an external fundamental system, is rarely found, suggesting that the vast majority of dinosaur specimens that have ever been sectioned either die before they attain maximum body size, or that not all dinosaurs developed an EFS. It is more likely that we would conclude from any histological study that the specimen was not a juvenile, but not fully grown, which is what we have determined from our skeletal investigation anyway: histology will not shed light on the final body size of the taxon. If there were many specimens of this genus (as there are for *Coelophysis*, for example) we would definitely have attempted histological sectioning, but as it is, we can't justify destructively sampling this specimen given the likelihood of little advance in our knowledge.

The authors are free to contact me with any questions, requests for clarification, or concerns.

Chris Griffin
chris.griffin@yale.edu

See attached file "Spiekman et al_2021_Pendraig orynys_GriffinMajorComments.docx" for the modified figures referenced above, showing suggested locations for histological sampling.

Reviewer: 2

Comments to the Author(s)

I congratulate the authors on what is a well-researched and well-written manuscript. The description is a good length and provides sufficient comparisons, the assessment and subsequent discussion of skeletal maturity in the new taxon is welcome and justified, and the figures appropriately provide the reader with visual representations of important anatomical characteristics. I conducted the phylogenetic analysis in TNT and have no concerns about that treatment or the subsequent regressions and ancestral state reconstructions (but see below for some comments on interpretation). I have attached my grammar and syntax edits/comments in a marked-up PDF, but I outline some of the important comments and suggested changes below. Thank you for the opportunity to review this work; it's an important new taxon that says a lot about the diversity of the uppermost western European Triassic and I think will eventually help stabilize this part of the tree.

Page 2, line 47: I would characterize this not necessarily as discrete body size reduction but rather maintaining the plesiomorphic condition of smaller body sizes among dinosauromorphs/pterosauromorphs. See PDF for further comments later in the discussion.

Early small-sized ornithodirans (e.g., pterosauriforms, *Lagosuchus*) have been also included in our ancestral body size analyses. Our reconstructions found an increase of body size among dracohors/dinosaurs from the condition present in early ornithodirans, and a body size reduction among coelophysids from the ancestral condition of Dinosauria and from that of Neotheropoda (the closest ancestral node). So, we think that the interpretation of coelophysids as maintaining a plesiomorphic small size is not supported by the results of the analyses. However, because observed changes are relatively small, we have nuanced our statement in the text by writing that "coelophysoids underwent a small body size decrease early in their evolution".

Page 5, line 25: Please add Intuitional Abbreviations section where appropriate.

We have now added an institutional abbreviations section. Thank you for pointing this out!

Page 12, line 31: I personally avoid the word "fused" in these contexts because it can mean different things to different workers in morphological/developmental frameworks. Consider using "coossified"?

We have replaced fused with co-ossified throughout the text when two or more bony structures have grown or ossified together. When specifically talking about the absence or presence of visible sutures, we have maintained the term fused (e.g., whether or not there is a visible suture between the pubis and ischium).

Page 14, line 48: Please refer to the 'Padian theropod' as PEFO 21373/UCMP 129618 throughout; fossils from federal lands in the US must be first referenced by their federal catalogue number and then by an auxiliary number (if necessary, which in this case I think is since most people know the specimen by its UCMP number).

We have now refer to both numbers for this specimen.

Page 19, line 46: The holotype of *kayentakatae* is MNA V2623, is that what you mean here and a few other places?

We have corrected this typing error throughout the manuscript.

Page 21, line 29: Did your matrix (or that of Novas et al., 2021 for that matter) include modifications/scorings from Marsh and Rowe, 2020? I'm not asking you to redo analyses, just wanted to clarify. That could be one reason for some different tree topologies between your results and those of Marsh and Rowe, 2020 when it comes to *Lepidus* and/or *kayentakatae*. By the way, at some point we really need to score *Sinosaurus* and *Shuangbaisaurus* into this thing but I think they'll still fall up-tree from coelophysoids.

No, it didn't. The modifications from Marsh and Rowe (2020) were not included in the matrix of Novas et al. (2021), nor here. However, this version of the data matrix and the changes of Marsh and Rowe (2020) will be integrated together in a project that it is currently being lead by one of the authors of this manuscript (see preliminary results in Ezcurra et al. 2021. A REVISION OF COELOPHYSOID THEROPOD SPECIMENS FROM PETRIFIED FOREST NATIONAL PARK, ARIZONA (USA), REVEALS A NEW SPECIES FROM THE UPPER TRIASSIC CHINLE FORMATION. Libro de Resúmenes de las 34 Jornadas Argentinas de Paleontología de Vertebrados: 39-40).

***Sinosaurus* and its posible synonym *Shuangbaisaurus* are very up-tree in comparison to coelophysoids, they could even be averostrans. We need a single dataset that includes all these taxa and one of the authors is working on that (MDE).**

Page 21, line 36: I was going to ask about this... I agree it probably doesn't change anything in the part of the tree you're looking at, but at some point those modifications should be incorporated into this matrix for it to be even more useful/up to date.

Yes, we agree, one of us (MDE) is working to incorporatate all these new information into a single dataset.

Page 26, line 49: "Size decreases occurred early in Coelophysoidea and ancestral values gradually increase in consecutive nodes" - I'm not sure you can say this with what is displayed in Fig. 8 (by the way, the next 1.5 pages of text rely heavily on supplemental data). Coelophysoid size was already within the range of most other dinosauriforms, and it only decreases three times (independently?) in *Pendraig*, *Powellvenator*, and *Procompsognathus* with what is shown in this non-supplemental figure. It's surprising to me that *Segisaurus* doesn't show up as markedly smaller in Fig. 8. If there are data that are better displayed in the Supplement that you find yourself citing often, I suggest incorporating them into Fig. 8.

We agree with the reviewer that the size fluctuations within Coelophysoidea are not remarkably strong when compared to broader dinosauriform evolution. However, we still think it is worthwhile to point out that the earliest nodes within Coelophysoidea indicate a slight size decrease relative to the ancestral neotheropod size (a decrease that is statistically supported since it is larger than the corresponding variances for these nodes) and that these values gradually increase slightly in subsequent nodes. We have modified this sentence to state that "Minor size

decreases occurred early in Coelophysoidea and ancestral values gradually increase slightly in consecutive nodes...".

We agree with the reviewer that this section of the text refers a lot to the Supplementary Tables. We have tried to negate this by presenting an additional image in Figure 8 showing the nodal values and CIs for the region of interest in the tree. However, we nevertheless still refer to the Supplementary Tables in the text. We think this is relevant, because it indicates that, regardless of which parameters are used, our analyses show the same general patterns in body size evolution. We consider the tables to be too extensive to completely list them in the main text of our manuscript.

Although the femur length of *Segisaurus* is quite small (14.29 cm), it is still quite a bit larger than that of *Pendraig* (10.21 cm), *Powellvenator* (9.39 cm), and *Procompsognathus* (9.3 cm). This explains why it is found as less small (i.e. less cyan colored) than these other taxa. Also note that the branch length for this taxon specifically is quite long, which has an influence on length of the closest ancestral node, which will be longer than if the branch would be shorter.

Page 27, line 41: "...our dataset is ambiguous regarding insular dwarfism as a possible explanation of the reduced body size in *Pendraig* orynys. However, insular dwarfism in *Pendraig* orynys cannot be excluded..." - Agreed, but in reality your analyses did not test the hypothesis that insular dwarfism occurred, but rather that body size was changing across the tree in a strictly quantitative way. It can't be excluded because that's not actually what was tested. I think the way you frame your discussion is fine, just be aware of that subtle but important distinction. You didn't test the hypothesis of island dwarfism, you tested the hypothesis that a systematic decrease in body size occurred that may or may not be the result of island dwarfism.

We state in the sentence above the one quoted by the reviewer that "Because its small size is not unique among Coelophysoidea and other coelophysoid taxa that underwent a similar size reduction were not restricted to insular environments, our dataset is ambiguous regarding insular dwarfism as a possible explanation of the reduced body size in *Pendraig milnerae*." We wanted to test how body sizes change throughout early neotheropod evolution to see whether *Pendraig* represents an anomalously small taxon. If this were to be the case, insular dwarfism could be a potential explanation for this anomaly. If *Pendraig* was not found to be small at all, then this would suggest against the possibility of insular dwarfism for this taxon. This was our approach to investigating this issue. However, the results are ambiguous when it comes to the question of whether *Pendraig* was anomalously small. Since the reviewer sees no real fault with the way we framed this discussion, and we had already stated that insular dwarfism was one possible explanation of reduced body size, we have maintained it.

Page 28, line 22: "... which could be attributable to a lack of resources to sustain larger predators as is typical in certain island environments..." - Be very careful here to avoid circular reasoning. You can't use the fact that there are no large-bodied predator fossils as a line of evidence to support island dwarfism, and then also go on to say that island dwarfism caused a lack of large-bodied predators. For what it's worth, I think taphonomic filters are almost certainly at play here. The fossil record is actually pretty bad and the fact that we can use it to say anything at all is always so incredible to me.

We agree that the statement “which could be attributable to a lack of resources to sustain larger predators as is typical in certain island environments” represents a form of circular reasoning, so we have removed it from the text.

Again, this is overall a very well-written and important contribution and I congratulate the authors on a good piece of work. Please consider my thoughts and let me know if you have any questions or want any clarifications.

Adam Marsh